# Localized Randomized Smoothing for Collective Robustness Certification

**Jan Schuchardt**[1]*, **Tom Wollschläger**[1]*, **Aleksandar Bojchevski**[2], **Stephan Günnemann**[1]
{j.schuchardt,t.wollschlaeger,s.guennemann}@tum.de
{bojchevski}@cispa.de
[1]Technical University of Munich
[2]CISPA Helmholtz Center for Information Security

## Abstract

Models for image segmentation, node classification and many other tasks map a single input to multiple labels. By perturbing this single shared input (e.g. the image) an adversary can manipulate several predictions (e.g. misclassify several pixels). Collective robustness certification is the task of provably bounding the number of robust predictions under this threat model. The only dedicated method that goes beyond certifying each output independently is limited to *strictly local* models, where each prediction is associated with a small receptive field. We propose a more general collective robustness certificate for all types of models. We further show that this approach is beneficial for the larger class of *softly local* models, where each output is dependent on the entire input but assigns different levels of importance to different input regions (e.g. based on their proximity in the image). The certificate is based on our novel localized randomized smoothing approach, where the random perturbation strength for different input regions is proportional to their importance for the outputs. Localized smoothing Pareto-dominates existing certificates on both image segmentation and node classification tasks, simultaneously offering higher accuracy and stronger certificates.

## 1 Introduction

There is a wide range of tasks that require models making multiple predictions based on a single input. For example, semantic segmentation requires assigning a label to each pixel in an image. When deploying such *multi-output* classifiers in practice, their robustness should be a key concern. After all – just like simple classifiers (Szegedy et al., 2014) – they can fall victim to adversarial attacks (Xie et al., 2017; Zügner & Günnemann, 2019; Belinkov & Bisk, 2018). Even without an adversary, random noise or measuring errors can cause predictions to unexpectedly change.

We propose a novel method providing provable guarantees on *how many* predictions can be changed by an adversary. As all outputs operate on the same input, they have to be attacked simultaneously by choosing a single perturbed input, which can be more challenging for an adversary than attacking them independently. We must account for this to obtain a proper *collective robustness certificate*.

The only dedicated collective certificate that goes beyond certifying each output independently (Schuchardt et al., 2021) is only beneficial for models we call *strictly local*, where each output depends on a small, pre-defined subset of the input. Multi-output classifiers , however, are often only *softly local*. While all their predictions are in principle dependent on the entire input, each output may assign different importance to different subsets. For example, convolutional networks for image segmentation can have small effective receptive fields (Luo et al., 2016; Liu et al., 2018), i.e. primarily use a small region of the image in labeling each pixel. Many models for node classification are based on the homophily assumption that connected nodes are mostly of the same class. Thus, they primarily use features from neighboring nodes. Transformers, which can in principle attend to arbitrary parts of the input, may in practice learn "sparse" attention maps, with the prediction for each token being mostly determined by a few (not necessarily nearby) tokens (Shi et al., 2021).

---

*equal contribution

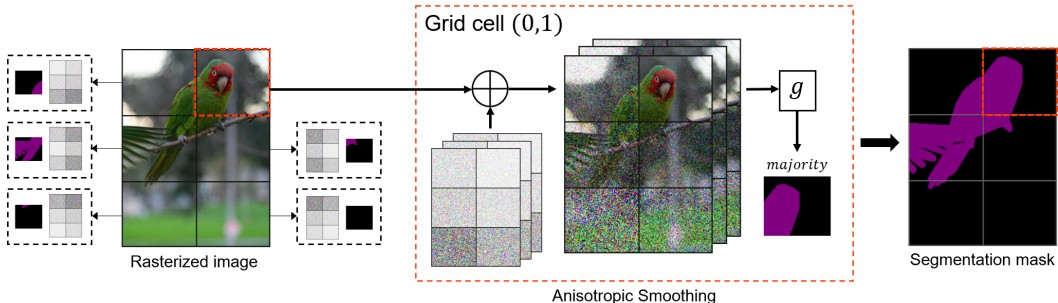

Figure 1: Localized randomized smoothing applied to semantic segmentation. We assume that the most relevant information for labeling a pixel is contained in other nearby pixels. We partition the input image into multiple grid cells. For each grid cell, we sample noisy images from a *different* anisotropic distribution that applies more noise to far-away, less relevant cells. Segmenting all noisy images, cropping the result and computing the majority vote yields a local segmentation mask. These per-cell segmentation masks can then be combined into a complete segmentation mask.

Softly local models pose a *budget allocation problem* for an adversary that tries to simultaneously manipulate multiple predictions by crafting a single perturbed input. When each output is primarily focused on a different part of the input, the attacker has to distribute their limited adversarial budget and may be unable to attack all predictions at once.

We propose *localized randomized smoothing*, a novel method for the collective robustness certification of softly local models that exploits this budget allocation problem. It is an extension of randomized smoothing (Lécuyer et al., 2019; Li et al., 2019; Cohen et al., 2019), a versatile black-box certification method which is based on constructing a smoothed classifier that returns the expected prediction of a model under random perturbations of its input (more details in § 2). Randomized smoothing is typically applied to single-output models with *isotropic* Gaussian noise. In localized smoothing however, we smooth each output (or set of outputs) of a multi-output classifier using a *different* distribution that is *anisotropic*. This is illustrated in Fig. 1, where the predicted segmentation masks for each grid cell are smoothed using a different distribution. For instance, the distribution for segmenting the top-right cell applies less noise to the top-right cell. The smoothing distribution for segmenting the bottom-left cell applies significantly more noise to the top-right cell.

Given a specific output of a softly local model, using a low noise level for the most relevant parts of the input lets us preserve a high prediction quality. Less relevant parts can be smoothed with a higher noise level to guarantee more robustness. The resulting certificates (one per output) explicitly quantify how robust each prediction is to perturbations of which part of the input. This information about the smoothed model's locality can then be used to combine the per-prediction certificates into a stronger collective certificate that accounts for the adversary's budget allocation problem.[1]

Our core contributions are:
- *Localized randomized smoothing*, a novel smoothing scheme for multi-output classifiers.
- An efficient anisotropic randomized smoothing certificate for discrete data.
- A collective certificate based on localized randomized smoothing.

## 2 BACKGROUND AND RELATED WORK

**Randomized smoothing.** Randomized smoothing is a certification technique that can be used for various threat models and tasks. For the sake of exposition, let us discuss a certificate for $l_2$ perturbations (Cohen et al., 2019). Assume we have a $D$-dimensional input space $\mathbb{R}^D$, label set $\mathbb{Y}$ and classifier $g : \mathbb{R}^D \to \mathbb{Y}$. We can use isotropic Gaussian noise to construct the *smoothed classifier* $f = \text{argmax}_{y \in \mathbb{Y}} \Pr_{\boldsymbol{z} \sim \mathcal{N}(\boldsymbol{x}, \sigma)} [g(\boldsymbol{z}) = y]$ that returns the most likely prediction of *base classifier* $g$ under the input distribution[2]. Given an input $\boldsymbol{x} \in \mathbb{R}^D$ and smoothed prediction $y = f(\boldsymbol{x})$, we can then easily determine whether $y$ is robust to all $l_2$ perturbations of magnitude $\epsilon$, i.e. whether $\forall \boldsymbol{x}' : ||\boldsymbol{x}' - \boldsymbol{x}||_2 \le \epsilon : f(\boldsymbol{x}') = y$. Let $q = \Pr_{\boldsymbol{z} \sim \mathcal{N}(\boldsymbol{x}, \sigma)} [g(\boldsymbol{z}) = y]$ be the probability of predicting

---

[1]An implementation will be made available at https://www.cs.cit.tum.de/daml/localized-smoothing.

[2]In practice, all probabilities have to be estimated using Monte Carlo sampling (see discussion in § G).

label $y$. The prediction is certifiably robust if $\epsilon < \sigma \Phi^{-1}(q)$ (Cohen et al., 2019). This result showcases a trade-off inherent to randomized smoothing: Increasing the noise level ($\sigma$) may strengthen the certificate, but could also lower the accuracy of $f$ or reduce $q$ and thus weaken the certificate.

**White-box certificates for multi-output classifiers.** There are multiple recent methods for certifying the robustness of multi-output models by analyzing their specific architecture and weights (for example, see (Tran et al., 2021; Zügner & Günnemann, 2019; Bojchevski & Günnemann, 2019; Zügner & Günnemann, 2020; Ko et al., 2019; Ryou et al., 2021; Shi et al., 2020; Bonaert et al., 2021)). They are however not designed to certify collective robustness, i.e. determine whether multiple outputs can be simultaneously attacked using a single perturbed input. They can only determine independently for each prediction whether or not it can be attacked.

**Black-box certificates for multi-output classifiers.** Most directly related to our work is the aforementioned certificate of Schuchardt et al. (2021), which is only beneficial for strictly local models (i.e. models where each output has a small receptive field). In § I we show that, for randomly smoothed models, their certificate is a special case of ours. SegCertify (Fischer et al., 2021) is a collective certificate for segmentation. This method certifies each output independently using isotropic smoothing (ignoring the budget allocation problem) and uses Holm correction (Holm, 1979) to obtain tighter Monte Carlo estimates. It then counts the number of certifiably robust predictions and tests whether it equals the number of predictions. In § H we demonstrate that our method can always provide guarantees that are at least as strong. Another method that can in principle be used to certify collective robustness is center smoothing (Kumar & Goldstein, 2021). It bounds the change of a vector-valued function w.r.t to a distance function. Using the $l_0$ pseudo-norm, it can bound how many predictions can be simultaneously changed. More recently, Chen et al. (2022) proposed a collective certificate for bagging classifiers. Different from our work, they consider poisoning (train-time) instead of evasion (test-time) attacks. Yatsura et al. (2022) prove robustness for segmentation, but consider patch-based instead of $\ell_p$-norm attacks and certify each prediction independently.

**Anisotropic randomized smoothing.** While only designed for single-output classifiers, two recent certificates for anisotropic Gaussian and uniform smoothing (Fischer et al., 2020; Eiras et al., 2022) can be used as a component of our collective certification approach: They can serve as per-prediction certificates, which we can then combine into our stronger collective certificate (more details in § 3.2).

## 3 PRELIMINARIES

### 3.1 COLLECTIVE THREAT MODEL

We assume a multi-output classifier $f : \mathbb{X}^{D_{\text{in}}} \to \mathbb{Y}^{D_{\text{out}}}$, that maps $D_{\text{in}}$-dimensional inputs to $D_{\text{out}}$ labels from label set $\mathbb{Y}$. We further assume that this classifier $f$ is the result of randomly smoothing each output of a base classifier $g$. Given this multi-output classifier $f$, an input $\boldsymbol{x} \in \mathbb{X}^{D_{\text{in}}}$ and the corresponding predictions $\boldsymbol{y} = f(\boldsymbol{x})$, the objective of the adversary is to cause as many predictions from a set of targeted indices $\mathbb{T} \subseteq \{1, \dots, D_{\text{out}}\}$ to change. That is, their objective is $\min_{\boldsymbol{x}' \in \mathbb{B}_{\boldsymbol{x}}} \sum_{n \in \mathbb{T}} \mathrm{I}\left[f_n(\boldsymbol{x}') = y_n\right]$, where $\mathrm{I}$ is the indicator function and $\mathbb{B}_{\boldsymbol{x}} \subseteq \mathbb{X}^{D_{\text{in}}}$ is the perturbation model. As is common in robustness certification, we assume a $\ell_p$-norm perturbation model, i.e. $\mathbb{B}_{\boldsymbol{x}} = \left\{\boldsymbol{x}' \in \mathbb{X}^{D_{\text{in}}} \mid ||\boldsymbol{x}' - \boldsymbol{x}||_p \le \epsilon\right\}$ with $p, \epsilon \ge 0$. Importantly, note that the minimization operator is outside the sum, meaning the predictions have to be attacked using a single input.

### 3.2 A RECIPE FOR COLLECTIVE CERTIFICATES

Before discussing localized randomized smoothing, we show how to combine arbitrary per-prediction certificates into a collective certificate, a procedure that underlies both our method and that of Schuchardt et al. (2021) and Fischer et al. (2021). The first step is to apply an arbitrary certification procedure to each prediction $y_1, \dots, y_{D_{\text{out}}}$ in order to obtain per-prediction *base certificates*.

**Definition 3.1** (Base certificates). A base certificate for a prediction $y_n = f_n(\boldsymbol{x})$ is a set $\mathbb{H}^{(n)} \subseteq \mathbb{X}^{D_{\text{in}}}$ of perturbed inputs s.t. $\forall \boldsymbol{x}' \in \mathbb{H}^{(n)} : f_n(\boldsymbol{x}') = y_n$.

Using these base certificates, one can derive two bounds on the adversary's objective:

$$\min_{\boldsymbol{x}' \in \mathbb{B}_{\boldsymbol{x}}} \sum_{n \in \mathbb{T}} \mathrm{I}\left[f_n(\boldsymbol{x}') = y_n\right] \underset{(1.1)}{\ge} \min_{\boldsymbol{x}' \in \mathbb{B}_{\boldsymbol{x}}} \sum_{n \in \mathbb{T}} \mathrm{I}\left[\boldsymbol{x}' \in \mathbb{H}^{(n)}\right] \underset{(1.2)}{\ge} \sum_{n \in \mathbb{T}} \min_{\boldsymbol{x}' \in \mathbb{B}_{\boldsymbol{x}}} \mathrm{I}\left[\boldsymbol{x}' \in \mathbb{H}^{(n)}\right]. \quad (1)$$

Eq. 1.1 follows from Theorem 3.1 (if a prediction is certifiably robust to $\boldsymbol{x}'$, then $f_n(\boldsymbol{x}') = y_n$), while Eq. 1.2 results from moving the $\min$ operator inside the summation.

Eq. 1.2 is the *naïve collective certificate*: It iterates over the predictions and counts how many are certifiably robust to perturbation model $\mathbb{B}_{\boldsymbol{x}}$. Each summand involves a separate minimization problem. Thus, the certificate neglects that the adversary has to choose a single perturbed input to attack all outputs. SegCertify (Fischer et al., 2021) applies this to isotropic Gaussian smoothing.

While Eq. 1.1 is seemingly tighter than the naïve collective certificate, it may lead to identical results. For example, let us consider the most common case where the base certificates guarantee robustness within an $l_p$ ball, i.e. $\mathbb{H}^{(n)} = \left\{ \boldsymbol{x}'' \mid ||\boldsymbol{x}'' - \boldsymbol{x}||_p \leq r^{(n)} \right\}$ with certified radii $r^{(n)}$. Then, the optimal solution to both Eq. 1.1 and Eq. 1.2 is to choose an arbitrary $\boldsymbol{x}'$ with $||\boldsymbol{x}' - \boldsymbol{x}|| = \epsilon$:

$$\min_{\boldsymbol{x}' \in \mathbb{B}_{\boldsymbol{x}}} \sum_{n \in \mathbb{T}} \mathrm{I}\left[ \boldsymbol{x}' \in \mathbb{H}^{(n)} \right] = \sum_{n \in \mathbb{T}} \mathrm{I}\left[ \epsilon < r^{(n)} \right] = \sum_{n \in \mathbb{T}} \min_{\boldsymbol{x}' \in \mathbb{B}_{\boldsymbol{x}}} \mathrm{I}\left[ \boldsymbol{x}' \in \mathbb{H}^{(n)} \right].$$

The main contribution of Schuchardt et al. (2021) is to notice that, by exploiting strict locality (i.e. the outputs having small receptive fields), one can augment certificate Eq. 1.1 to make it tighter than the naive collective certificate from Eq. 1.2. One must simply mask out all perturbations falling outside a given receptive field when evaluating the corresponding base certificate:

$$\min_{\boldsymbol{x}' \in \mathbb{B}_{\boldsymbol{x}}} \sum_{n \in \mathbb{T}} \mathrm{I}\left[ \left( \boldsymbol{\psi}^{(n)} \odot \boldsymbol{x}' + (1 - \boldsymbol{\psi}^{(n)}) \odot \boldsymbol{x} \right) \in \mathbb{H}^{(n)} \right].$$

Here, $\boldsymbol{\psi}^{(n)} \in \{0,1\}^{D_{\mathrm{in}}}$ encodes the receptive field of $f_n$ and $\odot$ is the elementwise product. If two outputs $f_n$ and $f_m$ have disjoint receptive fields (i.e. $\boldsymbol{\psi}^{(n)^T} \boldsymbol{\psi}^{(m)} = 0$), then the adversary has to split up their limited adversarial budget and may be unable to attack both at once.

## 4 LOCALIZED RANDOMIZED SMOOTHING

The core idea behind localized smoothing is that, rather than improving upon the naïve collective certificate by using external knowledge about *strict* locality, we can use anisotropic randomized smoothing to obtain base certificates that directly encode *soft* locality. Here, we explain our approach in a domain-independent manner before turning to specific distributions and data-types in § 5.

In localized randomized smoothing, we associate base classifier outputs $g_1, \ldots, g_{D_{\mathrm{out}}}$ with distinct anisotropic smoothing distributions $\Psi_{\boldsymbol{x}}^{(1)}, \ldots, \Psi_{\boldsymbol{x}}^{(D_{\mathrm{out}})}$ that depend on input $\boldsymbol{x}$. For example, they could be Gaussian distributions with mean $\boldsymbol{x}$ and distinct covariance matrices – like in Fig. 1, where we use a different distribution for each grid cell. We use these distributions to construct the smoothed classifier $f$, where each output $f_n(\boldsymbol{x})$ is the result of randomly smoothing $g_n(Z)$ with $\Psi_{\boldsymbol{x}}^{(n)}$.

To certify robustness for a vector of predictions $\boldsymbol{y} = f(\boldsymbol{x})$, we follow the procedure discussed in § 3.2, i.e. compute base certificates $\mathbb{H}^{(1)}, \ldots, \mathbb{H}^{(D_{\mathrm{out}})}$ and solve Eq. 1.1. We do not make any assumption about how the base certificates are computed. However, we require that they comply with a common interface, which will later allow us combine them via linear programming:

**Definition 4.1** (Base certificate interface). A base certificate $\mathbb{H}^{(n)} \subseteq \mathbb{X}^{D_{\mathrm{in}}}$ is compliant with our base certificate interface for $l_p$-norm perturbations if there is a $\boldsymbol{w} \in \mathbb{R}_+^{D_{\mathrm{in}}}$ and $\eta^{(n)} \in \mathbb{R}_+$ such that

$$\mathbb{H}^{(n)} = \left\{ \boldsymbol{x}' \;\middle|\; \sum_{d=1}^{D_{\mathrm{in}}} w_d^{(n)} \cdot |x_d' - x_d|^p < \eta^{(n)} \right\}. \tag{2}$$

The weight $w_d^{(n)}$ quantifies how sensitive $y_n$ is to perturbations of input dimension $d$. It will be smaller where the anisotropic smoothing distribution applies more noise. The radius $\eta^{(n)}$ quantifies the overall level of robustness. In § 5 we present different distributions and corresponding certificates that comply with this interface. Inserting Eq. 2 into Eq. 1.1 results in the collective certificate

$$\min_{\boldsymbol{x}' \in \mathbb{B}_{\boldsymbol{x}}} \sum_{n \in \mathbb{T}} \mathrm{I}\left[ \sum_{d=1}^{D_{\mathrm{in}}} w_d^{(n)} \cdot |x_d' - x_d|^p < \eta^{(n)} \right]. \tag{3}$$

Eq. 3 showcases why locally smoothed models admit a collective certificate that is stronger than naïvely certifying each output independently (i.e. Eq. 1.2). Because we use different distributions for different outputs, any two outputs $f^{(n)}$ and $f^{(m)}$ will have distinct certificate weights $\boldsymbol{w}^{(n)}$ and $\boldsymbol{w}^{(m)}$. If they are sensitive in different parts of the input, i.e. $\boldsymbol{w}^{(n)^T}\boldsymbol{w}^{(m)}$ is small, then the adversary has to split up their limited adversarial budget and may be unable to attack both at once. One particularly simple example is the case $\boldsymbol{w}^{(n)^T}\boldsymbol{w}^{(m)} = 0$, where attacking predictions $y_n$ and $y_m$ requires allocating adversarial budget to two entirely disjoint sets of input dimensions. In § I we show that, with appropriately parameterized smoothing distributions, we can obtain base certificates with $\boldsymbol{w}^{(n)} = c \cdot \boldsymbol{\psi}^{(n)}$, with indicator vector $\boldsymbol{\psi}^{(n)}$ encoding the receptive field of output $n$. Hence, the collective guarantees from (Schuchardt et al., 2021) are a special case of our certificate.

## 4.1 Computing the Collective Certificate

While Eq. 3 constitutes a valid certificate, it is not immediately clear how to evaluate it. However, we notice that the perturbation set $\mathbb{B}_{\boldsymbol{x}}$ imposes linear constraints on the elementwise differences $|x'_d - x_d|^p$, the values of the indicator functions are binary variables and that the base certificates inside the indicator functions are characterized by linear inequalities. We can thus reformulate Eq. 3 as a mixed-integer linear program (MILP), which leads us to our main result (proof in § D):

**Theorem 4.2.** *Given locally smoothed model $f$, input $\boldsymbol{x} \in \mathbb{X}^{(D_{\text{in}})}$, smoothed prediction $\boldsymbol{y} = f(\boldsymbol{x})$ and base certificates $\mathbb{H}^{(1)}, \ldots, \mathbb{H}^{D_{\text{out}}}$ complying with interface Eq. 2, the number of simultaneously robust predictions $\min_{\boldsymbol{x}' \in \mathbb{B}_{\boldsymbol{x}}} \sum_{n \in \mathbb{T}} \mathrm{I}\left[f_n(\boldsymbol{x}') = y_n\right]$ is lower-bounded by*

$$\min_{\boldsymbol{b} \in \mathbb{R}_+^{D_{\text{in}}}, \boldsymbol{t} \in \{0,1\}^{D_{\text{out}}}} \sum_{n \in \mathbb{T}} t_n \tag{4}$$

$$s.t. \quad \forall n : \boldsymbol{b}^T \boldsymbol{w}^{(n)} \geq (1 - t_n)\eta^{(n)}, \;\; \text{sum}\{\boldsymbol{b}\} \leq \epsilon^p. \tag{5}$$

The vector $\boldsymbol{b}$ models the allocation of adversarial budget (i.e. the elementwise differences $b_d = |x'_d - x_d|^p$). The vector $\boldsymbol{t}$ serves the same role as the indicator functions from Eq. 3, i.e. it indicates which predictions are certifiably robust. Eq. 5 ensures that $\boldsymbol{b}$ does not exceed the overall budget $\epsilon$ (i.e. $\boldsymbol{x}' \in \mathbb{B}_{\boldsymbol{x}}$) and that $t_n$ can only be set to 0 if $\boldsymbol{b}^T \boldsymbol{w}^{(n)} \geq \eta^{(n)}$, i.e. only when the base certificate cannot guarantee robustness for prediction $y_n$. This problem can be solved using any MILP solver. Its optimal value provably bounds the number of simultaneously robust predictions.

## 4.2 Improving Efficiency

Solving large MILPs is expensive. In § E we show that partitioning the outputs into $N_{\text{out}}$ subsets sharing the same smoothing distribution and the inputs into $N_{\text{in}}$ subsets sharing the same noise level (for example like in Fig. 1, where we partition the image into a $2 \times 3$ grid), as well as quantizing the base certificate parameters $\eta^{(n)}$ into $N_{\text{bin}}$ bins, reduces the number of variables and constraints from $D_{\text{in}} + D_{\text{out}}$ and $D_{\text{out}} + 1$ to $N_{\text{in}} + N_{\text{out}} \cdot N_{\text{bins}}$ and $N_{\text{out}} \cdot N_{\text{bins}} + 1$, respectively. We can thus control the problem size independent of the data's dimensionality. We further derive a linear relaxation of the MILP that can be efficiently solved while preserving the soundness of the certificate.

## 4.3 Accuracy-Robustness Tradeoff

When discussing Eq. 3, we only explained why our collective certificate for *locally smoothed* models is better than a naïve combination of localized smoothing base certificates. However, this does not necessarily mean that our certificate is also stronger than naïvely certifying an *isotropically smoothed* model. This is why we focus on soft locality. With isotropic smoothing, high certified robustness requires using large noise levels, which degrade the model's prediction quality. Localized smoothing, when applied to softly local models, can circumvent this issue. For each output, we can use low noise levels for the most important parts of the input to retain high prediction quality. Our LP-based collective certificate allows us to still provide strong collective robustness guarantees. We investigate this improved accuracy-robustness trade-off in our experimental evaluation (see § 7).

## 5    BASE CERTIFICATES

To apply our collective certificate in practice, we require smoothing distributions $\Psi_{\boldsymbol{x}}^{(n)}$ and corresponding per-prediction base certificates that comply with the interface from Theorem 3.1. As base certificates for $l_2$ and $l_1$ perturbations we can reformulate existing anisotropic Gaussian (Fischer et al., 2020; Kumar & Goldstein, 2021) and uniform (Kumar & Goldstein, 2021) smoothing certificates for single-output models: For $\Psi_{\boldsymbol{x}}^{(n)} = \mathcal{N}(\boldsymbol{x}, \text{diag}(\boldsymbol{s}^{(n)}))$ we have $w_d^{(n)} = 1/(s_d^{(n)})^2$ and $\eta^{(n)} = (\Phi^{-1}(q_{n,y_n}))^2$ with $q_{n,y_n} = \Pr_{\boldsymbol{z} \sim \Psi_{\boldsymbol{x}}^{(n)}}[g_n(\boldsymbol{z}) = y]$. For $\Psi_{\boldsymbol{x}}^{(n)} = \mathcal{U}(\boldsymbol{x}, \boldsymbol{\lambda}^{(n)})$ we have $w_d^{(n)} = 1/\lambda_d^{(n)}$ and $\eta^{(n)} = \Phi^{-1}(q_{n,y_n})$. We prove the correctness of these reformulations in § F.

For $l_0$ perturbations of binary data, we can use a distribution $\mathcal{F}(\boldsymbol{x}, \boldsymbol{\theta})$ that flips $x_d$ with probability $\theta_d \in [0, 1]$, i.e. $\Pr[z_d \neq x_d] = \theta_d$ for $\boldsymbol{z} \sim \mathcal{F}(\boldsymbol{x}, \boldsymbol{\theta})$. Existing methods (e.g. (Lee et al., 2019)) can be used to derive per-prediction certificates for this distribution, but have exponential runtime in the number of unique values in $\boldsymbol{\theta}$. Thus, they are not suitable for localized smoothing, which uses different $\theta_d$ for different parts of the input. We therefore propose a novel, more efficient approach: *Variance-constrained certification*, which smooths the base classifier's softmax scores instead of its predictions and then uses both their expected value and variance to certify robustness (proof in § F.3):

**Theorem 5.1** (Variance-constrained certification). *Given a function* $g : \mathbb{X} \to \Delta_{|\mathbb{Y}|}$ *mapping from discrete set* $\mathbb{X}$ *to scores from the* $(|\mathbb{Y}| - 1)$-*dimensional probability simplex, let* $f(\boldsymbol{x}) = \text{argmax}_{y \in \mathbb{Y}} \mathbb{E}_{\boldsymbol{z} \sim \Psi_{\boldsymbol{x}}} [g(\boldsymbol{z})_y]$ *with smoothing distribution* $\Psi_{\boldsymbol{x}}$ *and probability mass function* $\pi_{\boldsymbol{x}}(\boldsymbol{z}) = \Pr_{\tilde{\boldsymbol{z}} \sim \Psi_{\boldsymbol{x}}}[\tilde{\boldsymbol{z}} = \boldsymbol{z}]$. *Given an input* $\boldsymbol{x} \in \mathbb{X}$ *and smoothed prediction* $y = f(\boldsymbol{x})$, *let* $\mu = \mathbb{E}_{\boldsymbol{z} \sim \Psi_{\boldsymbol{x}}} [g(\boldsymbol{z})_y]$ *and* $\zeta = \mathbb{E}_{\boldsymbol{z} \sim \Psi_{\boldsymbol{x}}} \left[ (g(\boldsymbol{z})_y - \nu)^2 \right]$ *with* $\nu \in \mathbb{R}$. *Assuming* $\nu \leq \mu$, *then* $f(\boldsymbol{x}') = y$ *if*

$$\sum_{\boldsymbol{z} \in \mathbb{X}} \frac{\pi_{\boldsymbol{x}'}(\boldsymbol{z})}{\pi_{\boldsymbol{x}}(\boldsymbol{z})} \cdot \pi_{\boldsymbol{x}'}(\boldsymbol{z}) < 1 + \frac{1}{\zeta - (\mu - \nu)^2} \left( \mu - \frac{1}{2} \right). \tag{6}$$

The l.h.s. of Eq. 6 is the expected ratio between the probability mass functions of the smoothing distributions for the perturbed ($\pi_{\boldsymbol{x}'}$) and unperturbed ($\pi_{\boldsymbol{x}}$) input.[3] It is equal to 1 if both densities are the same, i.e. there is no adversarial perturbation, and greater than 1 otherwise. The r.h.s. of Eq. 6 depends on the expected softmax score $\mu$, a variable $\nu \leq \mu$ and the expected squared difference $\zeta$ between $\mu$ and $\nu$. For $\nu = \mu$ the parameter $\zeta$ is the variance of the softmax score. A higher expected value and a lower variance allow us to certify robustness for larger adversarial perturbations.

Applying Theorem 5.1 with flipping distribution $\mathcal{F}(\boldsymbol{x}, \boldsymbol{\theta})$ to each of the $D$ softmax vectors of our model's outputs yields $l_0$-norm certificates for binary data that can be computed in linear time (see § F.3.1). In § F.3.2, we also apply it to the sparsity-aware smoothing distribution (Bojchevski et al., 2020), allowing us to differentiate between adversarial deletions and additions of bits. Theorem 5.1 can also be generalized to continuous distributions (see § F.3.3). But, for fair comparison with our baselines, we use the certificates of Eiras et al. (2022) as our base certificates for continuous data. In practice, the smoothed classifier and the base certificates cannot be evaluted exactly. One has to use Monte Carlo sampling to provide guarantees that hold with high probability (see § G).

## 6    LIMITATIONS

A limitation of our approach is that it assumes soft locality. It can be applied to arbitrary models, but may not necessarily result in better certificates than isotropic smoothing (recall  § 4.3). Also, choosing the smoothing distributions requires some assumptions about which parts of the input are how relevant to making a prediction. Our experiments show that natural assumptions like homophily can be sufficient. But choosing a distribution may be more challenging for other tasks. A limitation of (most) randomized smoothing methods is that they use sampling to approximate the smoothed classifier. Because we use multiple distributions, we can only use a fraction of the samples per distribution. We can alleviate this problem by sharing smoothing distributions among outputs (see § E.1). Still, future work should try to improve the sample efficiency of randomized smoothing or develop deterministic base certificates (e.g. by generalizing (Levine & Feizi, 2020) to anisotropic distributions), which could then be incorporated into our linear programming framework.

---

[3]This term is equivalent to the exponential of the Rényi-divergence $\exp(\mathcal{D}_\alpha(\Psi_{\boldsymbol{x}'} || \Psi_{\boldsymbol{x}}))$ with $\alpha = 2$.

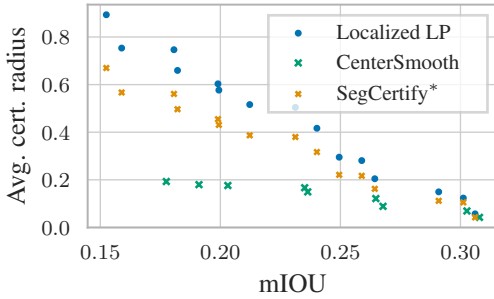 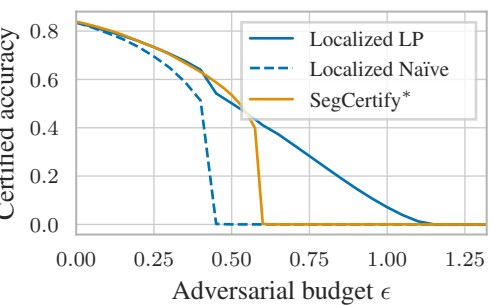

Figure 2: Comparison of isotropic smoothing with $\sigma_{\text{iso}} \in \{0.01, \ldots, 0.5\}$ to our LP-based certificate with $(\sigma_{\text{min}}, \sigma_{\text{max}}) = (\sigma_{\text{iso}}, \infty)$, using a modified, strictly local U-Net on Pascal-VOC. Localized smoothing offers the same mIOU as SegCertify* and stronger robustness certificates.

Figure 3: Certified accuracy of U-Net on Pascal-VOC. We compare SegCertify* ($\sigma_{\text{iso}} = 0.2$) to localized smoothing (($\sigma_{\text{min}}, \sigma_{\text{max}}$) = $(0.15, 1.0)$). Combining the base certificates (dashed blue line) via our collective LP (solid blue line) outperforms the baseline.

## 7 EXPERIMENTAL EVALUATION

In this section, we compare our method to all existing collective certificates for $\ell_p$-norm perturbations: Center smoothing using isotropic Gaussian noise (Kumar & Goldstein, 2021), SegCertify (Fischer et al., 2021) and the collective certificates of Schuchardt et al. (2021). To compare SegCertify to the other methods, we report the number of certifiably robust predictions and not just whether all predictions are robust. We write SegCertify* to highlight this. When considering models that are not strictly local (i.e. all outputs depend on all inputs) the certificates of Schuchardt et al. (2021) and Fischer et al. (2021) are identical, i.e., do not have to be evaluated separately. A more detailed description of the experimental setup, hardware and computational cost can be found in § C.

**Metrics.** Evaluating randomized smoothing methods based on certificate strength alone is not sufficient. Different distributions lead to different tradeoffs between prediction quality and certifiable robustness (as discussed in § 4.3). As metrics for prediction quality, we use *accuracy* and *mean intersection over union* (mIOU).[4] The main metric for certificate strength is the *certified accuracy $\xi(\epsilon)$*, i.e., the percentage of predictions that are correct and certifiably robust, given adversarial budget $\epsilon$. Following (Schuchardt et al., 2021), we use the *average certifiable radius* (ACR) as an aggregate metric, i.e. $\sum_{n=1}^{N-1} \epsilon_n \cdot (\xi(\epsilon_n) - \xi(\epsilon_{n+1})$ with budgets $\epsilon_1 \leq \epsilon_2 \cdots \leq \epsilon_N$ and $\epsilon_1 = 0, \xi(\epsilon_N) = 0$.

**Evaluation procedure.** We assess the accuracy-robustness tradeoff of each method by computing accuracy / mIOU and ACR for a wide range of smoothing distribution parameters. We then eliminate all points that are Pareto-dominated, i.e. for which there exist diffent parameter values that yield higher accuracy / mIOU and ACR. Finally, we assess to if localized smoothing dominates the baselines, i.e. whether it can be parameterized to achieve strictly better accuracy-robustness tradeoffs.

### 7.1 IMAGE SEGMENTATION

**Dataset and model.** We evaluate our certificate for $l_2$ perturbations on 100 images from the Pascal-VOC (Everingham et al., 2010) 2012 segmentation validation set. Training is performed on 10582 samples extracted from SBD, also known as "Pascal trainaug" (Hariharan et al., 2011). Additional experiments on Cityscapes (Cordts et al., 2016) can be found in § A. To increase batch sizes and thus allow a thorough investigation of different smoothing parameters, all images are downscaled to 50% of their original size, similar to (Fischer et al., 2021). Our base model is a U-Net segmentation model (Ronneberger et al., 2015) with a ResNet-18 backbone. For isotropic randomized smoothing, we use Gaussian noise $\mathcal{N}(0, \sigma_{\text{iso}})$ with different $\sigma_{\text{iso}} \in \{0.01, 0.02, \ldots, 0.5\}$. To perform localized randomized smoothing, we choose parameters $\sigma_{\text{min}}, \sigma_{\text{max}} \in \mathbb{R}_+$ and partition all images into regular grids (similar to Fig. 1). To smooth outputs in grid cell $(i, j)$, we sample noise for grid cell $(k, l)$ from $\mathcal{N}(0, \sigma' \cdot \mathbf{1})$, with $\sigma' \in [\sigma_{\text{min}}, \sigma_{\text{max}}]$ chosen proportional to the distance of $(i, j)$ and

---

[4]I.e. add up confusion matrices over the entire dataset, compute per-class IOUs and average over all classes.

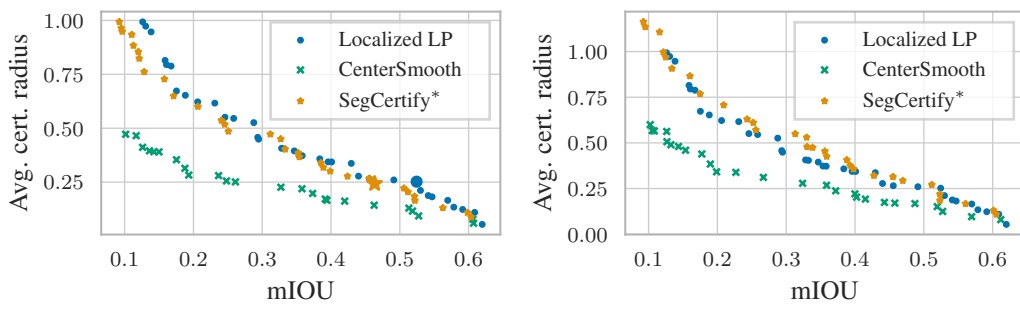

(a) 10240 samples per output pixel.
(b) 15 times as many samples for baselines.

Figure 4: Comparison of isotropic smoothing to our LP-based certificate with a $3 \times 5$ grid and U-Net on Pascal-VOC. U-Net is sufficiently local to benefit from localized smoothing (Fig. 4a), but not enough to offset the increased sample complexity (Fig. 4b) for the probabilistic base certificates.

$(k, l)$ (more details in § C.2). All training data is randomly perturbed using samples from the same smoothing distribution that is used for certification.

**Accuracy-robustness tradeoff under strict locality.** Our goal is to verify that, if a model is sufficiently local, localized smoothing offers a better accuracy-robustness tradeoff than isotropic smoothing. As an extreme example, we construct a strictly local model from our U-Net segmentation model. This modified model partitions each image into a grid of size $2 \times 2$. It then iterates over cells $(i, j)$, sets all values outside $(i, j)$ to $0$ and applies the original model. Finally, it stitches all $4$ segmentation masks into a single one. For such a strictly local model, we can apply localized smoothing with the same $2 \times 2$ grid and $\sigma_{\max} \to \infty$ to recover the certificate of Schuchardt et al. (2021) (see § I).[5] Fig. 2 compares the resulting trade-off for $\sigma_{\min} = \sigma_{\text{iso}}$ to that of both isotropic smoothing baselines using 153600 Monte Carlo samples. Localized smoothing yields the same mIOUs as SegCertify[*], but up to 22.4 p.p. larger ACR. Both approaches Pareto-dominate center smoothing.

**Accuracy-robustness tradeoff under soft locality.** Next, we want to verify our claim about the existence of softly local models for which localized smoothing is beneficial. To this end, we randomly smooth the U-Net model itself, without using masking to enforce strict locality. We perform localized smoothing with grid size $3 \times 5$, various $\sigma_{\min} \in \{0.01, 0.02, \ldots, 0.5\}, \sigma_{\max} \in [0.02, 1.0]$ and 10240 samples per output pixel (i.e. $10240 \cdot 15 = 153600$ samples in total). Isotropic smoothing is also performed with 10240 samples per output pixel. Fig. 4a shows that localized smoothing Pareto-dominates SegCertify[*] for high-accuracy models with mIOU $> 35.3\%$. Importantly the figure is not to be read like a line graph! Even if the vertical distance between two methods is small, one may significantly outperform the other. For example, $\sigma_{\text{iso}} = 0.1$, with an mIOU of $46.34\%$ and an ACR of $0.24$ (highlighted with a bold cross) is dominated by $(\sigma_{\min}, \sigma_{\max}) = (0.09, 0.2)$ (highlighted with a large circle), which has a larger ACR of $0.25$ and a mIOU that is a whole $6.1$ p.p. higher.

**Benefit of linear programming.** Fig. 3 demonstrates how the linear program derived in § 4.1 enables this improved tradeoff. We compare SegCertify[*] with $\sigma_{\text{iso}} = 0.2$ to localized smoothing with $(\sigma_{\min}, \sigma_{\max}) = (0.15, 1.0)$. Naïvely combining the base certificates (dashed line) is not sufficient for outperforming the baseline, as they cannot certify robustness beyond $\epsilon = 0.45$. However, solving the collective LP (solid blue line) extends the maximum certifiable radius to $\epsilon = 1.15$.

**Sample efficiency.** Using the same number of samples per output pixel for both localized and isotropic smoothing neglects that localized smoothing requires sampling from 15 different distributions, i.e. sampling 15 times as many images.[6] In Fig. 4b we allow the baselines to sample the same number of images. Now, localized smoothing is mostly dominated by SegCertify[*], except for high-accuracy models with mIOU $\in [52.4\%, 57.8\%]$ or mIOU $> 60.8\%$. We conclude that U-Net is local enough to benefit from localized smoothing, but not enough to offset the practical problem of having to work with fewer Monte Carlo samples (see also discusion in § 6) in the entire range of possible isotropic smoothing parameters. Note, however, that we can always recover the guarantees of SegCertify[*] by using a $1 \times 1$ grid (see § H).

---

[5]Note that they never evaluated their approach on image segmentation.
[6]This is however not necessary for the previously discussed strictly local model.

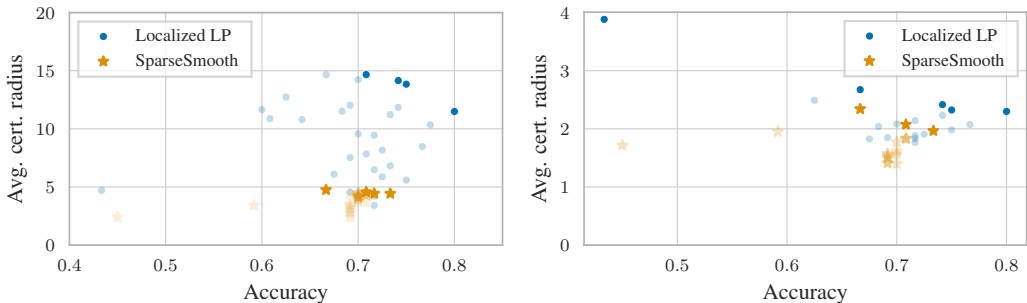

(a) Robustness to deletions, using $\mathcal{S}\left(\boldsymbol{x}, 0.01, \theta^{-}\right)$.    (b) Robustness to additions, using $\mathcal{S}\left(\boldsymbol{x}, 0.01, \theta^{-}\right)$.

Figure 5: Comparison of our LP-based collective certificate to Bojchevski et al. (2020), using APPNP on Citeseer. We consider both adversarial deletions (Fig. 5a) and additions (Fig. 5b) of attribute bits. Locally smoothed models offer a better accuracy-robustness tradeoff , especially for deletions. Transparent points signal that they are Pareto-dominated by points from the same method.

## 7.2 NODE CLASSIFICATION ON CITESEER

**Dataset and model.** Finally, we consider models that are designed with locality in mind: Graph neural networks. We take APPNP (Klicpera et al., 2019), which aggregates per-node predictions from the entire graph based on personalized pagerank scores, and apply it to the Citeseer (Sen et al., 2008) dataset. To certify its robustness, we perform randomized smoothing with sparsity-aware noise $\mathcal{S}\left(\boldsymbol{x}, \theta^{+}, \theta^{-}\right)$, where $\theta^{+}$ and $\theta^{-}$ control the probability of randomly adding or deleting node attributes, respectively (more details in § F.3.2). As a baseline we apply the tight certificate SparseSmooth of Bojchevski et al. (2020) to distributions $\mathcal{S}\left(\boldsymbol{x}, 0.01, \theta_{\mathrm{iso}}^{-}\right)$ with $\theta_{\mathrm{iso}}^{-} \in \{0.1, 0.15, \ldots, 0.95\}$. The small addition probability $0.01$ is meant to preserve the sparsity of the graph's attribute matrix and was used in most experiments in (Bojchevski et al., 2020). For localized smoothing, we partition the graph into $5$ clusters and define a minimum deletion probability $\theta_{\min}^{-} \in \{0.1, 0.15, \ldots, 0.95\}$. We then sample each cluster's attributes from $\mathcal{S}\left(\boldsymbol{x}, 0.01, \theta'^{-}\right)$ with $\theta'^{-} \in \left[\theta_{\min}^{-}, 0.95\right]$ chosen based on cluster affinity. To compute the base certificates, we use the variance-constrained certificate from § F.3.2. In all cases, we take $5 \cdot 10^{5}$ samples (i.e. $10^{5}$ per cluster for localized smoothing). Further discussions, as well as experiments on different models and datasets can be found in § B.

**Accuracy-robustness tradeoff.** Fig. 5 shows the accuracy and ACR pairs achieved by the naïve isotropic smoothing certificate and the LP-based certificate for localized smoothing. Despite having fewer samples per prediction, our method outperforms the baseline, offering higher accuracy certifying larger ACRs, especially for attribute deletions. Notably, in some cases, our approach even improves accuracy by over $7\,\mathrm{p.p.}$ percentage points compared to isotropically smoothed models. Similar to the observation made by Bojchevski et al. (2020) in their Section K, we also find that increasing the probability of attribute perturbations can improve accuracy to some extent. We posit that localized smoothing can leverage this phenomenon as a form of test-time regularization while preserving the crucial attributes of nearby nodes. In § B.1 we show that the stems from the smoothing scheme and is not solely due to using our novel variance-constrained certificate.

## 8 CONCLUSION

We proposed a novel approach to achieve provable collective robustness in multi-output classifiers that extends beyond strict locality, utilizing our introduced localized randomized smoothing scheme. Our approach involves smoothing different outputs with anisotropic smoothing distributions that match the model's soft locality. We demonstrated how per-output certificates obtained through localized smoothing can be combined into a strong collective robustness certificate using (mixed-integer) linear programming. Our experiments indicate that localized smoothing can achieve superior accuracy-robustness tradeoffs compared to isotropic smoothing methods. However, not all models match our distance-based locality assumption, particularly for image segmentation tasks. Node classification tasks are more amenable to localized smoothing due to their inherent locality. Our results highlight the importance of locality in achieving collective robustness and emphasize the need for future research to develop effective local models for multi-output tasks.

## 9 REPRODUCIBILITY STATEMENT

We prove all theoretic results that were not already derived in the main text in § D to § G. To ensure reproducibility of the experimental results we provide detailed descriptions of the evaluation process with the respective parameters in § C. An implementation, including configuration files, will be made available at https://www.cs.cit.tum.de/daml/localized-smoothing.

## 10 ETHICS STATEMENT

In this paper, we propose a method to increase the robustness of machine learning models against adversarial perturbations and to certify their robustness. We see this as an important step towards general usage of models in practice, as many existing methods are brittle to crafted attacks. Through the proposed method, we hope to contribute to the safe usage of machine learning. However, robust models also have to be seen with caution. As they are harder to fool, harmful purposes like mass surveillance are harder to avoid. We believe that it is still necessary to further research robustness of machine learning models as the positive effects can outweigh the negatives, but it is necessary to discuss the ethical implications of the usage in any specific application area.

## 11 ACKNOWLEDGEMENTS

This research is funded by the Bavarian Ministry of Economic Affairs, Regional Development and Energy with funds from the Hightech Agenda Bayern. Further, it is supported by the German Research Foundation, grant GU 1409/4-1.

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

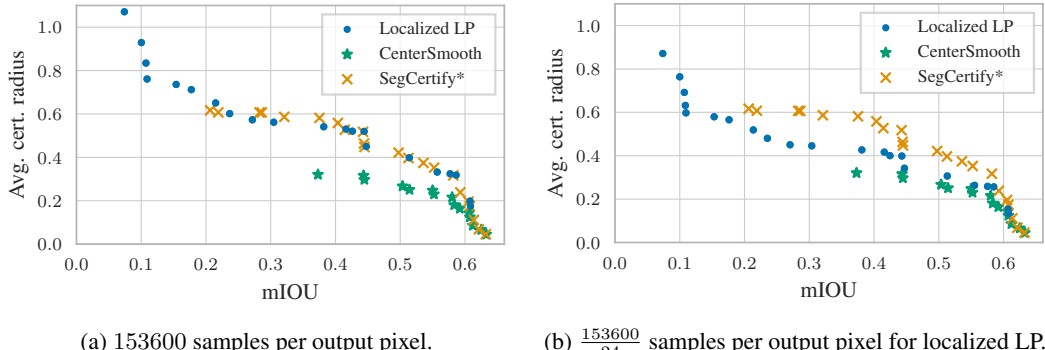

(a) 153600 samples per output pixel.

(b) $\frac{153600}{24}$ samples per output pixel for localized LP.

Figure 6: Comparison of our LP-based collective certificate for localized randomized smoothing with a $3 \times 5$ grid to CenterSmooth and SegCertify*, using DeepLabV3 on Cityscapes. Increasing the number of samples used for certifying each output from $6400 = \frac{153600}{24}$ to $153600$ (same as for the baselines) closes the gap between localized randomized smoothing and SegCertify. Still, localized smoothing only offers stronger certificates for models with $\text{mIOU} \leq 0.21$ (compared to $\text{mIOU} \leq 0.11$ when using fewer samples).

## A    IMAGE SEGMENTATION ON CITYSCAPES

In the following, we apply our approach to DeepLabv3 (Chen et al., 2017) models trained on the Cityscapes (Cordts et al., 2016) training set. We evaluate the certificates on $50$ images from the validation set. For localized smoothing, we partition the image into a grid of shape $4 \times 6$. To limit the number of LP variables despite the increased resolution, we quantize the base certificate parameters $\eta^{(n)}$ into $2048$ bins (see § E.2). Different from our experiments on Pascal-VOC and due to the increased computational cost of using higher-dimensional images, the locally smoothed models are not trained on the localized smoothing distribution with parameters $(\sigma_{\min}, \sigma_{\max})$. Instead, we use model trained with isotropic Gaussian noise with standard deviation $\sigma_{\text{iso}} = \sigma_{\min}$.

Fig. 6a shows that, even when allowing $153600$ samples per output pixel for both localized smoothing and the baselines (i.e. localized smoothing gets to sample $24$ times as many images), most choices of $(\sigma_{\min}, \sigma_{\max})$ do not offer higher accuracy and robustness than SegCertify*, except those leading to a small mIOU below $0.21$. Fig. 6b shows that reducing the number of samples per output pixel for localized smoothing to $6400 = \frac{153600}{24}$ further weakens the certificate. There, localized smoothing only offers stronger certificates for models with an mIOU below $0.11$.

There are three possible explanations for why localized smoothing does not outperform SegCertify*. The first one is that we do not train on the same distribution that we use for certification, so our models are less accurate or less consistent in their predictions, which reduces mIOU or certified robustness. The second one is that our simplisitic choice of localized smoothing based on grid cell distance (see § C.2) does not match the actual locality structure of DeepLabv3. The last one is that DeepLabv3, which uses dilated convolutions to increase the receptive field size in each layer, is just inherently less local than the U-Net architecture used in our experiments on Pascal-VOC. Nevertheless, it should be noted that we can always parameterize localized smoothing to obtain the same results as SegCertify* (see Appendix H).

## B    ADDITIONAL EXPERIMENTS ON NODE CLASSIFICATION

In the following, we perform additional experiments on graph neural networks for node classification, including a different model and an additional dataset. Unless otherwise stated, all details of the experimental setup are identical to § 7.2. In particular, we use sparsity-aware smoothing distribution $\mathcal{S}(\boldsymbol{x}, 0.01, \theta^-)$, where probability of deleting bits $\theta^-$ is either constant across the entire graph (for the isotropic randomized smoothing baseline) or adjusted per output and cluster based on cluster affinity (for localized randomized smoothing).

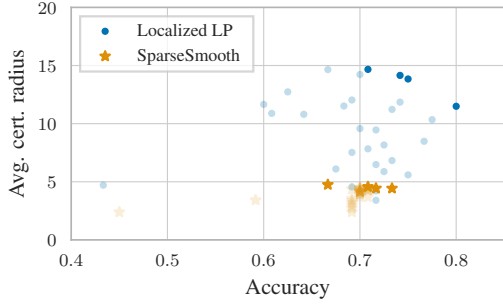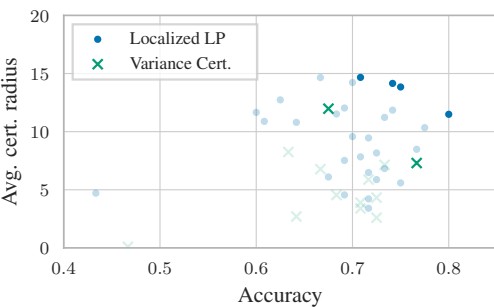

(a) Using (Bojchevski et al., 2020) for the naïve isotropic smoothing baseline.

(b) Using variance-constrained certification for the naïve isotropic smoothing baseline.

Figure 7: Analysis of our LP-based collective certificate using APPNP on Citeseer. We use the sparsity-aware smoothing with $\mathcal{S}(\boldsymbol{x}, 0.01, \theta^-)$ to certify robustness to deletions. In Fig. 7a we use the certificate of Bojchevski et al. (2020) for baseline (identical to Fig. 5a). In Fig. 7b we use variance-constrained certification (see Theorem 5.1) as baseline. In both cases, there are locally smoothed models with a higher accuracy than any of the isotropically smoothed models and significantly larger average certifiable radii.

### B.1    COMPARISON TO THE NAÏVE VARIANCE-CONSTRAINED ISOTROPIC SMOOTHING CERTIFICATE

In Fig. 5 of § 7.2, we observed that locally smoothed models surprisingly did not only achieve up to three times higher average certifiable radii, but simultaneously had higher accuracy than any of the isotropically smoothed models. One potential explanation is that we used variance-constrained certification (see Theorem 5.1) (i.e. smoothing the models' softmax scores instead of their predicted labels) for localized smoothing, but not for the isotropic smoothing baseline. This might result in two substantially different models. To investigate this, we repeat the experiment from Fig. 5a, using variance-constrained certification for both localized smoothing and the isotropic smoothing baseline. Fig. 7 shows that, no matter which smoothing paradigm we use for our isotropic smoothing baseline, there is a c.a. 7 p.p. difference in accuracy between the most accurate isotropically smoothed model and the most accurate locally smoothed model.

Interestingly, even variance-constrained smoothing with isotropic noise (green crosses in Fig. 7b) is sufficient for outperforming the isotropic smoothing certificate of Bojchevski et al. (2020) (orange stars in Fig. 7a). This showcases that variance-constrained certification does not only present a very efficient, but also a very effective way of certifying robustness on discrete data (even when entirely ignoring the collective robustness aspect).

### B.2    NODE CLASSIFICATION USING GRAPH CONVOLUTIONAL NETWORKS

So far, we have only used APPNP models as our base classifier. Now, we repeat our experiments using 6-layer Graph Convolutional Networks (GCN) (Kipf & Welling, 2017). In each layer, GCNs first apply a linear layer to each node's latent vector and then average over each node's 1-hop neighborhood. Thus, a 6-layer GCN classifies each node using attributes from all nodes in its 6-hop neighborhood, which covers most or all of the Citeseer graph. Aside from using GCN instead of APPNP as the base model, we leave the experimental setup from § 7.2 unchanged. Note that GCNs

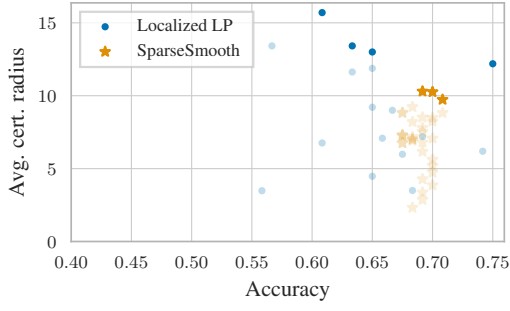 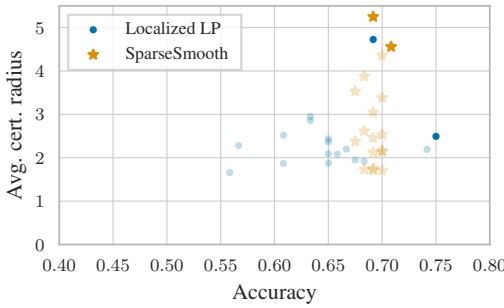

(a) Robustness to deletions, using $\mathcal{S}\left(\boldsymbol{x}, 0.01, \theta^-\right)$.  (b) Robustness to additions, using $\mathcal{S}\left(\boldsymbol{x}, 0.01, \theta^-\right)$.

Figure 8: Comparison of our LP-based collective certificate for localized randomized smoothing to SparseSmooth, using a 6-layer GCN on Citeseer. We consider both adversarial deletions (Fig. 8a) and additions (Fig. 8b). Some locally smoothed models have a higher accuracy than any of the isotropically smoothed models. However, our certificate only dominates the best isotropically smoothed models when considering robustness to deletions, not when considering robustness to additions. This can either be attributed to a lower locality in deep GCNs or variance-constrained certification yielding weak base certificates for addition when $\theta^+$ is small.

are typically used with fewer layers. However, these shallow models are strictly local and it has already been established that the certificate Schuchardt et al. (2021) – which is subsumed by our certificate (see § I.2) – can provide very strong robustness guarantees for them. We therefore increase the number of layers to obtain a model that is not strictly local.

Fig. 8 shows the results for both robustness to deletions and robustness to additions. Similar to APPNP, some locally smoothed models have an up to $4\,\mathrm{p.p.}$ higher accuracy than the most accurate isotropically smoothed model. When considering robustness to deletions, the locally smoothed models Pareto-dominate all of the isotropically smoothed models, i.e. offer better accuracy-robustness tradeoffs. Some can guarantee average certifiable radii that are at least $50\%$ larger than those of the baseline. When considering robustness to additions however, some of the isotropically smoothed models have a higher certifiably robustness.

We see two potential causes for our method's lower certifiable robustness to additions: The first potential cause is that the GCN may be less local than APPNP or that it has a different form of locality that does not match our clustering-based localized smoothing distributions. This appears plausible, as GCN averages uniformly over each neighborhood, whereas APPNP aggregates predictions based on pagerank scores. APPNP may thus primarily attend to specific, densely connected nodes, making it more local than GCN. The second potential cause is that the variance-constrained certificate we use as our base certificate may be less effective when certifying robustness to adversarial additions by using a very small addition probablity like $\theta^+ = 0.01$. Afterall, we have also seen in our experiments with APPNP in § 7.2 that the gap in average certifiable radii between localized and isotropic smoothing was significantly smaller when considering additions. We investigate this second potential cause in more detail in § B.4.

### B.3  NODE CLASSIFICATION ON CORA-ML

Next, we repeat our experiments with APPNP on the Cora-ML (McCallum et al., 2000; Bojchevski & Günnemann, 2018) node classification dataset, keeping all other parameters fixed. The results are shown in Fig. 9. Unlike on Citeseer, the locally smoothed models have a slightly reduced accuracy compared to the isotropically smoothed models. This can either be attributed to one smoothing approach having a more desirable regularizing effect on the neural network, or the fact that we smooth softmax scores instead of predicted labels when constructing the locally smoothed models. Nevertheless, when considering adversarial deletions, localized smoothing makes it possible to achieve average certifiable radii that are at least $50\%$ larger than any of the isotropically smoothed models' – at the cost of slightly reduced accuracy $8.6\%$. Or, for another point of the pareto front, we in-

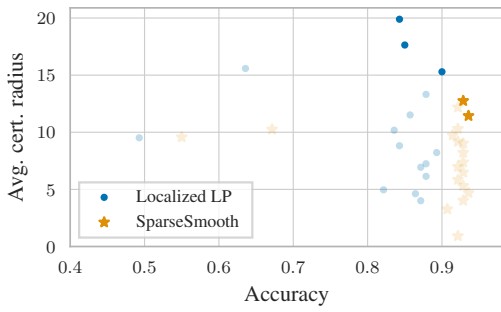 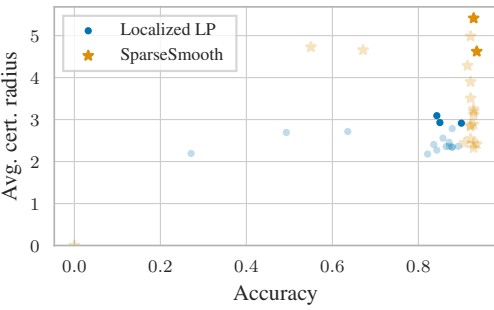

(a) Robustness to deletions, using $\mathcal{S}\left(\boldsymbol{x}, 0.01, \theta^{-}\right)$.      (b) Robustness to additions, using $\mathcal{S}\left(\boldsymbol{x}, 0.01, \theta^{-}\right)$.

Figure 9: Comparison of our LP-based collective certificate for localized randomized smoothing to the SparseSmooth (Bojchevski et al., 2020), using APPNP on Cora-ML. We consider both adversarial deletions (Fig. 9a) and additions (Fig. 9b). Some locally smoothed models that have a higher accuracy than any of the isotropically smoothed models. However, our method is only able to dominate all isotropically smoothed models when considering robustness to deletions, not when considering robustness to additions. This can either be attributed to a lower locality in deep GCNs or variance-constrained certification yielding weak base certificates for addition when $\theta^{+}$ is small.

crease the certificate by $20\%$ while reducing the accuracy by $2.8$ percentage points. As before, the certificates for attribute additions are significantly weaker.

## B.4   LOWER CERTIFIABLE ROBUSTNESS TO ADDITIONS

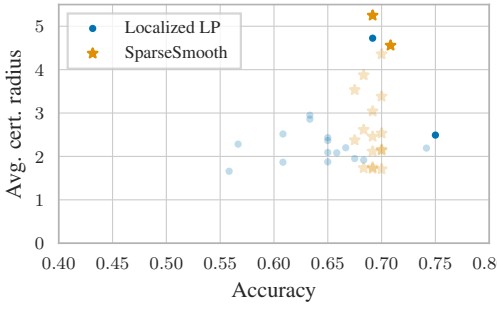 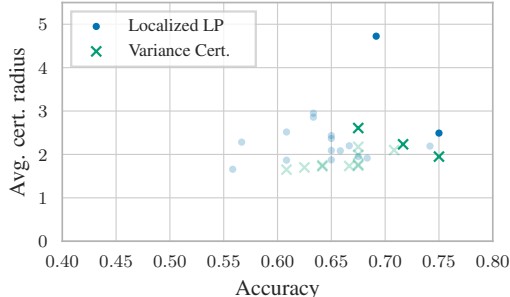

(a) Using (Bojchevski et al., 2020) for the naïve isotropic smoothing certificate.      (b) Using variance-constrained certification for the naïve isotropic smoothing certificate.

Figure 10: Comparison of our LP-based collective certificate for localized randomized smoothing to SparseSmooth and to a naïve combination of its base certificates, using GCN and adversarial additions on Citeseer. Fig. 10a shows that the LP-based certificate is outperformed by naïve isotropic smoothing. Fig. 10b shows that this is largely due to the variance-constrained base certificates (green crosses) for adversarial additions being much weaker than the isotropic smoothing certificate of (Bojchevski et al., 2020) in Fig. 10a.

While our certificates for adversarial deletions have compared favorably to the isotropic smoothing baseline in all previous experiments, our certificates for adversarial additions were comparatively weaker on Cora-ML and when using GCNs as base models. In the following, we investigate to what extend this can be attributed to our use of variance-constrained certification for our base certificates.

Fig. 10a shows both our linear programming collective certificate and the naïve isotropic smoothing certificate based on (Bojchevski et al., 2020) for GCNs on Citeseer under adversarial additions. In Fig. 10b, we plot not only the LP-based certificates, but also our variance-constrained base certificates (drawn as green crosses). Comparing both figures shows that our base certificate's average certifiable radii are at least $50\%$ smaller than the largest ACR achieved by (Bojchevski et al., 2020)

in Fig. 10a. While our linear program significantly improves upon them, it is not sufficient to overcome this significant gap. This result is in stark contrast to our results for attribute deletions § B.1, where the variance-constrained base certificates alone were enough to significantly outperform the certificate of (Bojchevski et al., 2020).

Now that we have established that the variance-constrained base certificates appear significantly weaker for additions, we can analyze why. For this, recall that our base certificates are parameterized by a weight vector $w$ (see Definition 4.1), with smaller values corresponding to higher robustness – or two weight vectors $w^+$, $w^-$ quantifying robustness to adversarial additions and deletions, respectively (see § F.3.2). Using our results from § F.3.2, we can draw the weights $w^+$ resulting from smoothing distribution $\mathcal{S}(x, 0.01, \theta^-)$ as a function of $\theta^-$. Fig. 11a shows that $\theta^-$ has to be brought very close to 1 in order to guarantee high robustness to deletions, effectively deleting almost all attributes in the graph. Alternatively, one can also increase the addition probability $\theta^+$ to perhaps 10% or 20%. But this would utterly destroy the sparsity of the graph's attribute matrix. We can conclude that, while variance-constrained certification can in principle provide strong certificates for attribute deletions, it might be a worse choice than the method of Bojchevski et al. (2020) for very sparse datasets that force the use of very low addition probabilities $\theta^+$.

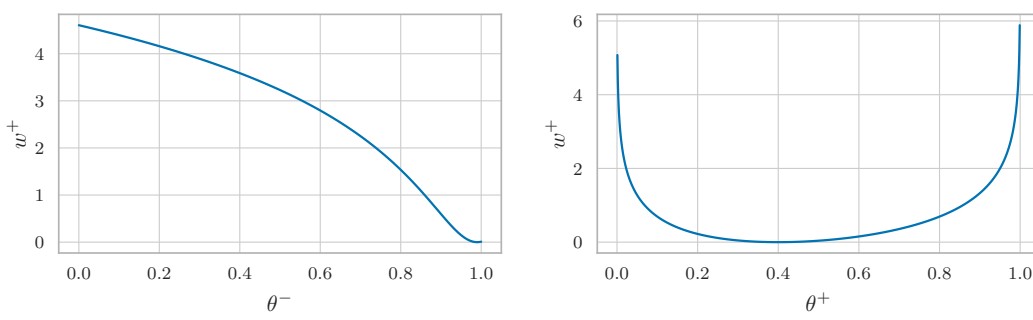

(a) Certificate weight $w^+$ for $\mathcal{S}(x, 0.01, \theta^-)$ for varying $\theta^-$.

(b) Certificate weight $w^+$ for $\mathcal{S}(x, \theta^+, 0.6)$ for varying $\theta^+$.

Figure 11: Base certificate weight $w^+$ of the variance-constrained sparsity-aware smoothing certificate for varying distribution parameters. Certifying high robustness to adversarial additions (i.e. obtaining small weights) requires either setting a high probability for random additions or an even higher probability for random deletions.

### B.5  BENEFIT OF LINEAR PROGRAMMING CERTIFICATES

As we did for our experiments on image segmentation (see Fig. 3), we can inspect the certified accuracy curves of specific smoothed models in more detail to gain a better understanding of how the collective linear programming certificate enables larger average certifiable radii. We use the same experimental setup as in § 7.2, i.e. APPNP on Citeseer, and certify robustness to deletions. We compare the certifiably most robust isotropically smoothed model ($\theta^-_{\text{iso}} = 0.8$, ACR $= 5.67$ to the locally smoothed model with $\theta^-_{\text{min}} = 0.75, \theta^+_{\text{max}} = 0.95$. For the locally smoothed models, we compute both LP-based collective certificate, as well as the naïve collective certificate.

Fig. 12 shows that even naïvely combining the localized smoothing base certificates obtained via variance-constrained certification (dashed blue line) is sufficient for outperforming the naïve isotropic smoothing certificate. This speaks to its effectiveness as a certificate against adversarial deletions. Combining the base certificates via linear programming (solid blue line) significantly enlarges this gap, leading to even larger maximum and average certifiable radii.

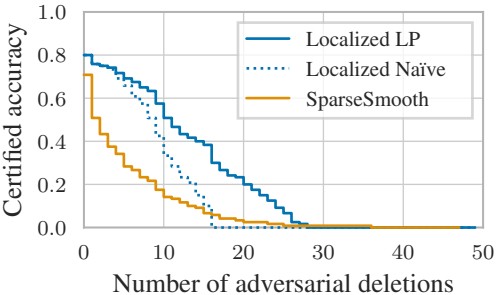

Figure 12: Certified accuracy of APPNP on Citeseer. We compare the naïve isotropic smoothing certificate of the most robust baseline model ($\theta^{-}_{\mathrm{iso}} = 0.8$) to localized smoothing ($\theta^{-}_{\mathrm{min}} = 0.75$). Even naïvely combining the variance-constrained base certificates (dashed blue line) is sufficient for outperforming the SparseSmooth certificate for 15 deletions or less. Combining the base certificates via our LP (solid blue line) further extends the certifiable radius and significantly increases the certified accuracy for perturbations with 5 or more deletions.

## C    DETAILED EXPERIMENTAL SETUP

In the following, we first explain the metrics we use for measuring the strength of certificates, and how they can be applied to the different types of randomized smoothing certificates used in our experiments. We then discuss the specific parameters and hyperparameters for our semantic segmentation and node classification experiments. We conclude by specifying the used hardware and comparing the computational cost of Monte Carlo sampling to that of solving the collective linear program.

### C.1    CERTIFICATE STRENGTH METRICS

We use two metrics for measuring certificate strength: For specific adversarial budgets $\epsilon$, we compute the certified accuracy $\xi(\epsilon)$ (i.e. the percentage of correct and certifiably robust predictions). As an aggregate metric, we compute the average certifiable radius, i.e. the lower Riemann integral of $\xi(\epsilon)$ evaluated at $\epsilon_1, \ldots, \epsilon_N$ with $\epsilon_1 = 0$ and $\xi(\epsilon_N) = 0$. For our experiments on image segmentation, we use $81$ equidistant points in $[0, 4]$. For our experiments on node classification, where we certify robustness to a discrete number of perturbations, we use $\epsilon_n = n$, i.e. natural numbers. In all experiments, we perform Monte Carlo randomized smoothing (see § G). Therefore, we may have to abstain from making predictions. Abstentions are counted as non-robust and incorrect. In the case of center smoothing, either all or no predictions abstain (this is inherent to the method. In our experiments, center smoothing never abstained).

#### C.1.1    COMPUTING CERTIFIED ACCURACY

The three different types of collective certificate considered in our experiments each require a different procedure for computing the certified accuracy. In the following, let $\mathbb{Z} = \{d \in \{1, \ldots, D_{\text{out}}\} \mid f_n(\boldsymbol{x}) = \hat{y}_n\}$ be the indices of correct predictions, given an input $\boldsymbol{x}$.

**Naïve collective certificate**. The naïve collective certificate certifies each prediction independently. Let $\mathbb{H}^{(n)}$ be the set of perturbed inputs $y_n$ is certifiably robust to (see Definition 3.1). Let $\mathbb{B}_{\boldsymbol{x}}$ be the collective perturbation model. Then $\mathbb{L} = \{d \in \{1, \ldots, D_{\text{out}}\} \mid \mathbb{B}_{\boldsymbol{x}} \subseteq \mathbb{H}^{(n)}\}$ is the set of all certifiably robust predictions. The certified accuracy can be computed as $\frac{|\mathbb{L} \cap \mathbb{Z}|}{D_{\text{out}}}$.

**Center smoothing** Center smoothing used for collective robustness certification does not determine which predictions are robust, but only the number of robust predictions. We therefore have to make the worst-case assumption that the correct predictions are the first to be changed by the adversary. Let $l$ be the number of certifiably robust predictions. The certified accuracy can then be computed as $\frac{\max(0, |\mathbb{Z}| - (D_{\text{out}} - l))}{D_{\text{out}}}$.

**Collective certificate**. Let $l(\mathbb{T})$ be the optimal value of our collective certificate for the set of targeted nodes $\mathbb{T}$. Then the certified accuracy can be computed via $\frac{l(\mathbb{T})}{D_{\text{out}}}$ with $\mathbb{T} = \mathbb{Z}$.

### C.2    SEMANTIC SEGMENTATION

Here, we provide all parameters of our experiments on image segmentation.

**Models.** As base models for the semantic segmentation tasks, we use U-Net (Ronneberger et al., 2015) and DeepLabv3 (Chen et al., 2017) segmentation heads with a ResNet-18 (He et al., 2016) backbone, as implemented by the Pytorch Segmentation Models library (version 0.13) (Yakubovskiy, 2020). We use the library's default parameters. In particular, the inputs to the U-Net segmentation head are the features of the ResNet model after the first convolutional layer and after each ResNet block (i.e. after every fourth of the subsequent layers). The U-Net segmentation head uses (starting with the original resolution) 16, 32, 64, 128 and 256 convolutional filters for processing the features at the different scales. For the DeepLabv3 segmentation head, we use all default parameters from Chen et al. (2017) and an output stride of 16. To avoid dimension mismatches in the segmentation head, all input images are zero-padded to a height and width that is the next multiple of 32.

**Data and preprocessing.** We evaluate our certificates on the Pascal-VOC 2012 and Cityscapes segmentation validation set. We do not use the test set, because evaluating metrics like the certified

accuracy requires access to the ground-truth labels. For training the U-Net models on Pascal, we use the 10582 Pascal segmentation masks extracted from the SBD dataset (Hariharan et al., 2011) (referred to as "Pascal trainaug" or "Pascal augmented training set" in other papers). SBD uses a different data split than the official Pascal-VOC 2012 segmentation dataset. We avoid data leakage by removing all training images that appear in the validation set. For training the DeepLabv3 model on Cityscapes, we use the default training set. We downscale both the training and the validation images and ground-truth masks to $50\%$ of their original height and width, so that we can use larger batch sizes and thus use our compute time to more thoroughly evaluate a larger range of different smoothing distributions. The segmentation masks are downscaled using nearest-neighbor interpolation, the images are downscaled using the INTER_AREA operation implemented in OpenCV (Bradski, 2000).

**Training and data augmentation.** We initialize our model weights using the weights provided by the Pytorch Segmentation Models library, which were obtained by pre-training on ImageNet. We train our models for 512 epochs, using Dice loss and Adam($lr = 0.001, \beta_1 = 0.9, \beta_2 = 0.999, \epsilon = 10^{-8}, \text{weight\_decay} = 0$). We use a batch size of 128 for Pascal-VOC and a batch size of 32 for Cityscapes. Every 8 epochs, we compute the mean IOU on the validation set. After training, we use the model that achieved the highest validation mean IOU. We apply the following train-time augmentations: With $50\%$ probability, each image is randomly scaled by a factor from $[1, 2.0]$ using the ShiftScaleRotate augmentation implemented by the Albumentations library (version $0.5.2$) (Buslaev et al., 2020). The images are than cropped to a fixed size of $160 \times 256$ (for Pascal-VOC) or $384 \times 384$ (for Cityscapes). Where necessary, the images are padded with zeros. Padded parts of the segmentation mask are ignored by the loss function. After these operations, each input is randomly perturbed using Gaussian noise. For isotropic smoothing, we use a fixed standard deviation $\sigma_{\text{iso}} \in \{0, 0.01, \ldots, 0.5\}$, i.e. we train 51 different models on different isotropic smoothing distributions. For localized smoothing with grid shape $H \times W$ and parameters $(\sigma_{\min}, \sigma_{\max})$ we perform localized smoothing with a single sample per image. Since this generates $H \cdot W$ as many perturbed images, we perform gradient accumulation, processing $\frac{1}{H \cdot W}$ of each batch at a time. All samples are clipped to $[0, 1]$ to retain valid RGB-values.

**Certification.** For Pascal-VOC, we evaluate all certificates on the first 100 images from the validation set that – after downscaling – have a resolution of $166 \times 250$. For Cityscapes, we use every tenth image from the validation set. For all certificates, we use Monte Carlo randomized smoothing (see discussion in § G). We use the significance parameter $\alpha$ to $0.01$, i.e. all certificates hold with probability $0.99$. For the center smoothing baseline, we use the default parameters suggested by the authors ($\Delta = 0.05, \beta = 2, \alpha_1 = \alpha_2$). For the naïve isotropic randomized smoothing baseline and for localized smoothing, we use Holm correction to account for the multiple comparisons problem, which yields strictly better results than Bonferroni correction (see § G.4). For our localized smoothing distribution, we partition the input image into a regular grid of size $H \times W$ (specified in the different paragraphs of § 7.1) and define minimum standard deviation $\sigma_{\min}$ and maximum standard deviation $\sigma_{\max}$. Let $\mathbb{J}^{(k,l)}$ be the set of all pixel coordinates in grid cell $(k, l)$. To smooth outputs in grid cell $(i, j)$, we use a smoothing distribution $\mathcal{N}(\mathbf{0}, \text{diag}(\boldsymbol{\sigma}))$ with $\forall k \in \{1, \ldots, H\}, l \in \{1, \ldots, W\}, d \in \mathbb{J}^{(k,l)}$,

$$\sigma_d = \sigma_{\min} + (\sigma_{\max} - \sigma_{\min}) \cdot \frac{\max(|i - k|, |l - j|)}{W}, \tag{7}$$

i.e. we linearly interpolate between $\sigma_{\min}$ and $\sigma_{\max}$ based on the $l_\infty$ distance of grid cells $(i, j)$ and $(k, l)$. All results are reported for the relaxed linear programming formulation of our collective certificate (see § E.4). The collective linear program is solved using MOSEK (version 9.2.46) (MOSEK ApS, 2019) through the CVXPY interface (version 1.1.13)

## C.3   NODE CLASSIFICATION

Here, we provide all parameters of our experiments on node classification.

**Model** We test two different models: 2-layer APPNP (Klicpera et al., 2019) and 6-layer GCN (Kipf & Welling, 2017). For both models we use a hidden size of $64$ and dropout with a probability of $0.5$. For the propagation step of APPNP we use 10 for the number of iterations and $0.15$ as the teleport probability.

**Data and preprocessing.** We evaluate our approach on the Cora-ML and Citeseer node classification datasets. We perform standard preprocessing, i.e., remove self-loops, make the graph undirected and select the largest connected component. We use the same data split as in (Schuchardt et al., 2021), i.e. 20 nodes per class for the train and validation set.

**Training and data augmentation** All models are trained with a learning rate of $0.001$ and weight decay of $0.001$. The models we use for sparse smoothing are trained with the noise distribution that is also reported for certification. The localized smoothing models are trained on the their minimal noise level, i.e., not with localized noise but with only $\theta_{\min}^+$ and $\theta_{\min}^-$.

**Certification** We evaluate our certificates on the validation nodes. For all certificates, we use Monte Carlo randomized smoothing (see discussion in § G). We use $1000$ samples for making smoothed predictions and $5 \cdot 10^5$ samples for certification. We use the significance parameter $\alpha$ to $0.01$, i.e. all certificates hold with probability $0.99$. For the naïve isotropic randomized smoothing baseline, we use Holm correction to account for the multiple comparisons problem, which yields strictly better results than Bonferroni correction (see § G.4). For our localized smoothing certificates, we use Bonferroni correction. To parameterize the localized smoothing distribution, we first perform Metis clustering (Karypis & Kumar, 1998) to partition the graph into 5 clusters. We create an affinity ranking by counting the number of edges which are connecting cluster $i$ and $j$. Specifically, let $\mathcal{C}$ be the set of clusters given by the Metis clustering. Then we count the number of edges between all cluster pairs and denote it by $N_{i,j}$, $i, j \in \mathcal{C}$. If the number of edges of the pair $(i, j)$ is higher than the number for all other pairs $(k, j)$ $\forall j \in \mathcal{C}$, i.e. $N_{i,j} > N_{k,j}$ $\forall k \in \mathcal{C}$, we can say that, due to the homophily assumption, cluster $i$ is the most important one for cluster $j$. We create this ranking for all pairs and use it to select the noise parameter $\theta'^-$ for smoothing the attributes of cluster $j$ while classifying a node of cluster $i$ out of the discrete steps of the linear interpolation between $\theta_{\min}$ and $\theta_{\max}$ based on its previously defined ranking between the clusters. An example would be, given 11 clusters, $\theta_{\min} = 0.0$, and $\theta_{\max} = 1.0$. If cluster $j$ second most important cluster to $i$, then we would take the second value out of $\{0.0, 0.1, \ldots, 1.0\}$. All results are reported for the relaxed linear programming formulation of our collective certificate (see § E.4). For each cluster, we use $\frac{1}{5}$ of the samples, which corresponds to 200 samples for prediction and $10^5$ samples for certification. The collective linear program is solved using MOSEK (version 9.2.46) (MOSEK ApS, 2019) through the CVXPY interface (version 1.1.13) (Diamond & Boyd, 2016).

## C.4 HARDWARE AND RUNTIME

The experiments on Pascal-VOC with strictly local models (Fig. 2) were performed using a Xeon E5-2630 v4 CPU @ 2.20GHz, an NVIDA GTX 1080TI GPU and $128$ GB of RAM. All other experiments were performed using an AMD EPYC 7543 CPU @ 2.80GHz, an NVIDA A100 GPU and $128$ GB of RAM.

In all cases, the time needed for obtaining the Monte Carlo samples required by both localized and isotropic smoothing was much larger than the cost of solving the collective linear program.

- For the strictly local model in Fig. 2, taking $153600$ samples took $294\,\mathrm{s}$ on average. Averaged over all images and adversarial budgets, solving each LP only took $0.91\,\mathrm{s}$.
- For the standard U-Net model in Fig. 4, taking $153600$ samples took $70.3\,\mathrm{s}$ on average. Each LP took $1.8\,\mathrm{s}$ on average.
- For the DeepLabv3 model in Fig. 6b, taking $153600$ samples took $1204\,\mathrm{s}$ on average. Each LP took $2.78\,\mathrm{s}$ on average.
- For the APPNP model in Fig. 5, taking $5 \cdot 10^6$ samples took $1034\,\mathrm{s}$ on average. Each LP took $10.9\,\mathrm{s}$ on average.

For graphs, the reported time for solving a single instance of the collective linear program is much higher than for image segmentation, even though the graph datasets require fewer variables. That is because we used a different, not as well vectorized formulation of the linear program in CVXPY.

In all cases, the time for calculating the isotropic smoothing certificates and base certificates from the Monte Carlo samples was too small to be measured accurately, since they can be implemented in a few simple vector operations.

## D    PROOF OF THEOREM 4.2

In the following, we prove Theorem 4.2, i.e. we derive the mixed-integer linear program that under-lies our collective certificate and prove that it provides a valid bound on the number of simultane-ously robust predictions. The derivation bears some semblance to that of (Schuchardt et al., 2021), in that both use standard techniques to model indicator functions using binary variables and that both convert optimization in input space to optimization in adversarial budget space. Nevertheless, both methods differ in how they encode and evaluate base certificates, ultimately leading to significantly different results (our method encodes each base certificate using only a single linear constraint and does not perform any masking operations).

*Theorem* 4.2.  Given locally smoothed model $f$, input $\boldsymbol{x} \in \mathbb{X}^{(D_{\text{in}})}$, smoothed prediction $\boldsymbol{y} = f(\boldsymbol{x})$ and base certificates $\mathbb{H}^{(1)}, \ldots, \mathbb{H}^{D_{\text{out}}}$ complying with interface Eq. 2, the number of simultaneously robust predictions $\min_{\boldsymbol{x}' \in \mathbb{B}_{\boldsymbol{x}}} \sum_{n \in \mathbb{T}} \mathrm{I}\left[f_n(\boldsymbol{x}') = y_n\right]$ is lower-bounded by

$$\min_{\boldsymbol{b} \in \mathbb{R}_+^{D_{\text{in}}}, \boldsymbol{t} \in \{0,1\}^{D_{\text{out}}}} \sum_{n \in \mathbb{T}} t_n \tag{8}$$

$$\text{s.t.} \quad \forall n: \boldsymbol{b}^T \boldsymbol{w}^{(n)} \geq (1 - t_n)\eta^{(n)}, \quad \text{sum}\{\boldsymbol{b}\} \leq \epsilon^p. \tag{9}$$

*Proof.* We begin by inserting the definition of our perturbation model $\mathbb{B}_{\boldsymbol{x}}$ and the base certificates $\mathbb{H}^{(n)}$ into Eq. 1.1:

$$\min_{\boldsymbol{x}' \in \mathbb{B}_{\boldsymbol{x}}} \sum_{n \in \mathbb{T}} \mathrm{I}\left[f_n(\boldsymbol{x}') = y_n\right] \geq \min_{\boldsymbol{x}' \in \mathbb{B}_{\boldsymbol{x}}} \sum_{n \in \mathbb{T}} \mathrm{I}\left[\boldsymbol{x}' \in \mathbb{H}^{(n)}\right] \tag{10}$$

$$= \min_{\boldsymbol{x}' \in \mathbb{X}^{D_{\text{in}}}} \sum_{n \in \mathbb{T}} \mathrm{I}\left[\sum_{d=1}^{D_{\text{in}}} w_d^{(n)} \cdot |x_d' - x_d|^p < \eta^{(n)}\right] \quad \text{s.t.} \quad \sum_{d=1}^{D_{\text{in}}} |x_d' - x_d|^p \leq \epsilon^p. \tag{11}$$

Evidently, input $\boldsymbol{x}'$ only affects the elementwise distances $|x_d' - x_d|^p$. Rather than optimizing $\boldsymbol{x}'$, we can directly optimize these distances, i.e. determine how much adversarial budget is allocated to each input dimension. For this, we define a vector of variables $\boldsymbol{b} \in \mathbb{R}_+^{D_{\text{in}}}$ (or $\boldsymbol{b} \in \{0,1\}^{D_{\text{in}}}$ for binary data). Replacing sums with inner products, we can restate Eq. 11 as

$$\min_{\boldsymbol{b} \in \mathbb{R}_+^{D_{\text{in}}}} \sum_{n \in \mathbb{T}} \mathrm{I}\left[\boldsymbol{b}^T \boldsymbol{w}^{(n)} < \eta^{(n)}\right] \quad \text{s.t.} \quad \text{sum}\{\boldsymbol{b}\} \leq \epsilon^p. \tag{12}$$

In a final step, we replace the indicator functions in Eq. 12 with a vector of boolean variables $\boldsymbol{t} \in \{0,1\}^{D_{\text{out}}}$.

$$\min_{\boldsymbol{b} \in \mathbb{R}_+^{D_{\text{in}}}, \boldsymbol{t} \in \{0,1\}^{D_{\text{out}}}} \sum_{n \in \mathbb{T}} t_n \tag{13}$$

$$\text{s.t.} \quad \forall n: \boldsymbol{b}^T \boldsymbol{w}^{(n)} \geq (1 - t_n)\eta^{(n)}, \quad \text{sum}\{\boldsymbol{b}\} \leq \epsilon^p. \tag{14}$$

The first constraint in Eq. 5 ensures that $t_n = 0 \iff \mathrm{I}\left[\boldsymbol{b}^T \boldsymbol{w}^{(n)} \geq \eta^{(n)}\right]$. Therefore, the optimiza-tion problem in Eq. 13 and Eq. 5 is equivalent to Eq. 12, which by transitivity is a lower bound on $\min_{\boldsymbol{x}' \in \mathbb{B}_{\boldsymbol{x}}} \sum_{n \in \mathbb{T}} \mathrm{I}\left[f_n(\boldsymbol{x}') = y_n\right]$. $\qquad \square$

# E  IMPROVING EFFICIENCY

In this section, we discuss different modifications to our collective certificate that improve its sample efficiency and allow us fine-grained control over the size of the collective linear program. We further discuss a linear relaxation of our collective linear program. All of the modifications preserve the soundness of our collective certificate, i.e. we still obtain a provable bound on the number of predictions that can be simultaneously attacked by an adversary. To avoid constant case distinctions, we first present all results for real-valued data, i.e. $\mathbb{X} = \mathbb{R}$, before mentioning any additional precautions that may be needed when working with binary data.

## E.1  SHARING SMOOTHING DISTRIBUTIONS AMONG OUTPUTS

In principle, our proposed certificate allows a different smoothing distribution $\Psi_{\boldsymbol{x}}^{(n)}$ to be used per output $g_n$ of our base model. In practice, where we have to estimate properties of the smoothed classifier using Monte Carlo methods, this is problematic: Samples cannot be re-used, each of the many outputs requires its own round of sampling. We can increase the efficiency of our localized smoothing approach by partitioning our $D_{\text{out}}$ outputs into $N_{\text{out}}$ subsets that share the same smoothing distributions. When making smoothed predictions or computing base certificates, we can then reuse the same samples for all outputs within each subsets.

More formally, we partition our $D_{\text{out}}$ output dimensions into sets $\mathbb{K}^{(1)}, \ldots, \mathbb{K}^{(N_{\text{out}})}$ with

$$\dot{\bigcup}_{i=1}^{N_{\text{out}}} \mathbb{K}^{(i)} = \{1, \ldots, D_{\text{out}}\}. \tag{15}$$

We then associate each set $\mathbb{K}^{(i)}$ with a smoothing distribution $\Psi_{\boldsymbol{x}}^{(i)}$. For each base model output $g_n$ with $n \in \mathbb{K}^{(i)}$, we then use smoothing distribution $\Psi_{\boldsymbol{x}}^{(i)}$ to construct the smoothed output $f_n$, e.g. $f_n(\boldsymbol{x}) = \operatorname{argmax}_{y \in \mathbb{Y}} \operatorname{Pr}_{\boldsymbol{z} \sim \Psi_{\boldsymbol{x}}^{(i)}} [f(\boldsymbol{x} + \boldsymbol{z}) = y]$ (note that for our variance-constrained certificate we smooth the softmax scores instead, see § 5).

## E.2  QUANTIZING CERTIFICATE PARAMETERS

Recall that our base certificates from § 5 are defined by a linear inequality: A prediction $y_n = f_n(\boldsymbol{x})$ is robust to a perturbed input $\boldsymbol{x}' \in \mathbb{X}^{D_{\text{in}}}$ if $\sum_{d=1}^{D} w_d^{(n)} \cdot |x_d' - x_d|^p < \eta^{(n)}$, for some $p \geq 0$. The weight vectors $\boldsymbol{w}^{(n)} \in \mathbb{R}^{D_{\text{in}}}$ only depend on the smoothing distributions. A side of effect of sharing the same distribution $\Psi_{\boldsymbol{x}}^{(i)}$ among all outputs from a set $\mathbb{K}^{(i)}$, as discussed in the previous section, is that the outputs also share the same weight vector $\boldsymbol{w}^{(i)} \in \mathbb{R}^{D_{\text{in}}}$ with $\forall n \in \mathbb{K}^{(i)} : \boldsymbol{w}^{(i)} = \boldsymbol{w}^{(n)}$. Thus, for all smoothed outputs $f_n$ with $n \in \mathbb{K}^{(i)}$, the smoothed prediction $y_n$ is robust if $\sum_{d=1}^{D} w_d^{(i)} \cdot |x_d' - x_d|^p < \eta^{(n)}$.

Evidently, the base certificates for outputs from a set $\mathbb{K}^{(i)}$ only differ in their parameter $\eta^{(n)}$. Recall that in our collective linear program we use a vector of variables $\boldsymbol{t} \in \{0, 1\}^{D_{\text{out}}}$ to indicate which predictions are robust according to their base certificates (see Theorem 4.2). If there are two outputs $f_n$ and $f_m$ with $\eta^{(n)} = \eta^{(m)}$, then $f_n$ and $f_m$ have the same base certificate and their robustness can be modelled by the same indicator variable. Conversely, for each set of outputs $\mathbb{K}^{(i)}$, we only need one indicator variable per unique $\eta^{(n)}$. By quantizing the $\eta^{(n)}$ within each subset $\mathbb{K}^{(i)}$ (for example by defining equally sized bins between $\min_{n \in \mathbb{K}^{(i)}} \eta^{(n)}$ and $\max_{n \in \mathbb{K}^{(i)}} \eta^{(n)}$), we can ensure that there is always a fixed number $N_{\text{bins}}$ of indicator variables per subset. This way, we can reduce the number of indicator variables from $D_{\text{out}}$ to $N_{\text{out}} \cdot N_{\text{bins}}$.

To implement this idea, we define a matrix of thresholds $\boldsymbol{E} \in \mathbb{R}^{N_{\text{out}} \times N_{\text{bins}}}$ with $\forall i : \min\{\boldsymbol{E}_{i,:}\} \leq \min_{n \in \mathbb{K}^{(i)}} (\{\eta^{(n)} \mid n \in \mathbb{K}^{(i)}\})$. We then define a function $\xi : \{1, \ldots, N_{\text{out}}\} \times \mathbb{R} \to \mathbb{R}$ with

$$\xi(i, \eta) = \max\left(\{E_{i,j} \mid j \in \{1, \ldots, N_{\text{bins}} \wedge E_{i,j} < \eta\}\right) \tag{16}$$

that quantizes base certificate parameter $\eta$ from output subset $\mathbb{K}^{(i)}$ by mapping it to the next smallest threshold in $\boldsymbol{E}_{i,:}$. We can then bound the collective robustness of the targeted dimensions $\mathbb{T}$ of our

prediction vector $\boldsymbol{y} = f(\boldsymbol{x})$ as follows:

$$\min \sum_{i \in \{1, \dots, N_{\text{out}}\}} \sum_{j \in \{1, \dots, N_{\text{bins}}\}} T_{i,j} \left| \left\{ n \in \mathbb{T} \cap \mathbb{K}^{(i)} \left| \xi\left(i, \eta^{(n)}\right) = E_{i,j} \right. \right\} \right| \tag{17}$$

$$\text{s.t.} \quad \forall i, j : \boldsymbol{b}^T \boldsymbol{w}^{(i)} \geq (1 - T_{i,j}) E_{i,j}, \quad \text{sum}\{\boldsymbol{b}\} \leq \epsilon^p \tag{18}$$

$$\boldsymbol{b} \in \mathbb{R}_+^{D_{\text{in}}}, \quad \boldsymbol{T} \in \{0,1\}^{N_{\text{out}} \times N_{\text{bins}}}. \tag{19}$$

Constraint Eq. 18 ensures that $T_{i,j}$ is only set to $0$ if $\boldsymbol{b}^T \boldsymbol{w}^{(i)} \geq E_{i,j}$, i.e. all predictions from subset $\mathbb{K}^{(i)}$ whose base certificate parameter $\eta^{(n)}$ is quantized to $E_{i,j}$ are no longer robust. When this is the case, the objective function decreases by the number of these predictions. For $N_{\text{out}} = D_{\text{out}}$, $N_{\text{bins}} = 1$ and $E_{n,1} = \eta^{(n)}$, we recover our general certificate from Theorem 4.2. Note that, if the quantization maps any parameter $\eta^{(n)}$ to a smaller number, the base certificate $\mathbb{H}^{(n)}$ becomes more restrictive, i.e. $y_n$ is considered robust to a smaller set of perturbed inputs. Thus, Eq. 17 is a lower bound on our general certificate from Theorem 4.2.

### E.3 Sharing Noise Levels Among Inputs

Similar to how partitioning the output dimensions allows us to control the number of output variables $\boldsymbol{t}$, partitioning the input dimensions and using the same noise level within each partition allows us to control the number of budget variables $\boldsymbol{b}$.

Assume that we have partitioned our output dimensions into $N_{\text{out}}$ subsets $\mathbb{K}^{(1)}, \dots, \mathbb{K}^{(N_{\text{out}})}$, with outputs in each subset sharing the same smoothing distribution $\Psi_{\boldsymbol{x}}^{(i)}$, as explained in § E.1. Let us now define $N_{\text{in}}$ input subsets $\mathbb{J}^{(1)}, \dots, \mathbb{J}^{(N_{\text{in}})}$ with

$$\dot{\bigcup}_{l=1}^{N_{\text{in}}} \mathbb{J}^{(l)} = \{1, \dots, D_{\text{out}}\}. \tag{20}$$

Recall that a prediction $y_n = f_n(\boldsymbol{x})$ with $n \in \mathbb{K}^{(i)}$ is robust to a perturbed input $\boldsymbol{x}' \in \mathbb{X}^{D_{\text{in}}}$ if $\sum_{d=1}^{D} w_d^{(i)} \cdot |x_d' - x_d|^p < \eta^{(n)}$ and that the weight vectors $\boldsymbol{w}^{(i)}$ only depend on the smoothing distributions. Assume that we choose each smoothing distribution $\Psi_{\boldsymbol{x}}^{(i)}$ such that $\forall l \in \{1, \dots, N_{\text{in}}\}, \forall d, d' \in \mathbb{J}^{(l)} : w_d^{(i)} = w_{d'}^{(i)}$, i.e. all input dimensions within each set $\mathbb{J}^{(l)}$ have the same weight. This can be achieved by choosing $\Psi_{\boldsymbol{x}}^{(i)}$ so that all dimensions in each input subset $\mathbb{J}^l$ are smoothed with the noise level (note that we can still use a different smoothing distribution $\Psi_{\boldsymbol{x}}^{(i)}$ for each set of outputs $\mathbb{K}^{(i)}$). For example, one could use a Gaussian distribution with covariance matrix $\boldsymbol{\Sigma} = \text{diag}\left(\boldsymbol{\sigma}\right)^2$ with $\forall l \in \{1, \dots, N_{\text{in}}\}, \forall d, d' \in \mathbb{J}^{(l)} : \sigma_d = \sigma_{d'}$.

In this case, the evaluation of our base certificates can be simplified. Prediction $y_n = f_n(\boldsymbol{x})$ with $n \in \mathbb{K}^{(n)}$ is robust to a perturbed input $\boldsymbol{x}' \in \mathbb{X}^{D_{\text{in}}}$ if

$$\sum_{d=1}^{D_{\text{in}}} w_d^{(i)} \cdot |x_d' - x_d|^p < \eta^{(n)} \tag{21}$$

$$= \sum_{l=1}^{N_{\text{in}}} \left( u^{(i)} \cdot \sum_{d \in \mathbb{J}^{(l)}} |x_d' - x_d|^p \right) < \eta^{(n)}, \tag{22}$$

with $\boldsymbol{u} \in \mathbb{R}_+^{N_{\text{in}}}$ and $\forall i \in \{1, \dots, N_{\text{out}}\}, \forall l \in \{1, \dots, N_{\text{in}}\}, \forall d \in \mathbb{J}^{(l)} : u_l^i = w_d^i$. That is, we can replace each weight vector $\boldsymbol{w}^{(i)}$ that has one weight $w_d^{(i)}$ per input dimension $d$ with a smaller weight vector $\boldsymbol{u}^{(i)}$ featuring one weight $u_l^{(i)}$ per input subset $\mathbb{J}^{(l)}$.

For our linear program, this means that we no longer need a budget vector $\boldsymbol{b} \in \mathbb{R}_+^{D_{\text{in}}}$ to model the elementwise distance $|x_d' - x_d|^p$ in each dimension $d$. Instead, we can use a smaller budget vector $\boldsymbol{b} \in \mathbb{R}_+^{N_{\text{in}}}$ to model the overall distance within each input subset $\mathbb{J}^{(l)}$, i.e. $b^{(l)} = \sum_{d \in \mathbb{J}^{(l)}} |x_d' - x_d|^p$. Combined with the quantization of certificate parameters from the previous section, our optimization

problem becomes

$$\min \sum_{i \in \{1,\ldots,N_{\text{out}}\}} \sum_{j \in \{1,\ldots,N_{\text{bins}}\}} T_{i,j} \left| \left\{ n \in \mathbb{T} \cap \mathbb{K}^{(i)} \,\middle|\, \xi\left(i, \eta^{(n)}\right) = E_{i,j} \right\} \right| \tag{23}$$

$$\text{s.t.} \quad \forall i,j : \boldsymbol{b}^T \boldsymbol{u}^{(i)} \geq (1 - T_{i,j}) E_{i,j}, \quad \text{sum}\{\boldsymbol{b}\} \leq \epsilon^p, \tag{24}$$

$$\boldsymbol{b} \in \mathbb{R}_+^{N_{\text{in}}}, \quad \boldsymbol{T} \in \{0,1\}^{N_{\text{out}} \times N_{\text{bins}}}. \tag{25}$$

with $\boldsymbol{u} \in \mathbb{R}^{N_{\text{in}}}$ and $\forall i \in \{1,\ldots,N_{\text{out}}\}, \forall l \in \{1,\ldots,N_{\text{in}}\}, \forall d \in \mathbb{J} : u_l^i = w_d^i$. For $N_{\text{out}} = D_{\text{out}}$, $N_{\text{in}} = D_{\text{in}}$, $N_{\text{bins}} = 1$ and $E_{n,1} = \eta^{(n)}$, we recover our general certificate from Theorem 4.2.

When certifying robustness for binary data, we impose different constraints on $\boldsymbol{b}$. To model that the adversary can not flip more bits than are present within each subset, we use a budget vector $\boldsymbol{b} \in \mathbb{N}_0^{N_{\text{in}}}$ with $\forall l \in \{1,\ldots,N_{\text{in}}\} : b_l \leq \left| \mathbb{J}^{(l)} \right|$, instead of a continuous budget vector $\boldsymbol{b} \in \mathbb{R}_+^{N_{\text{in}}}$.

### E.4 Linear Relaxation

Combining the previous steps allows us to reduce the number of problem variables and linear constraints from $D_{\text{in}} + D_{\text{out}}$ and $D_{\text{out}} + 1$ to $N_{\text{in}} + N_{\text{out}} \cdot N_{\text{bins}}$ and $N_{\text{out}} \cdot N_{\text{bins}} + 1$, respectively. Still, finding an optimal solution to the mixed-integer linear program may be too expensive. One can obtain a lower bound on the optimal value and thus a valid, albeit more pessimistic, robustness certificate by relaxing all discrete variables to be continuous.

When using the general certificate from Theorem 4.2, the binary vector $\boldsymbol{t} \in \{0,1\}^{D_{\text{out}}}$ can be relaxed to $\boldsymbol{t} \in [0,1]^{D_{\text{out}}}$. When using the certificate with quantized base certificate parameters from § E.2 or § E.3, the binary matrix $\boldsymbol{T} \in [0,1]^{N_{\text{out}} \times N_{\text{bins}}}$ can be relaxed to $\boldsymbol{T} \in [0,1]^{N_{\text{out}} \times N_{\text{bins}}}$. Conceptually, this means that predictions can be partially certified, i.e. $t_n \in (0,1)$ or $T_{i,j} \in (0,1)$. In particular, a prediction can be partially certified even if we know that is impossible to attack under the collective perturbation model $\mathbb{B}_{\boldsymbol{x}} = \left\{ \boldsymbol{x}' \in \mathbb{X}^{D_{\text{in}}} \mid ||\boldsymbol{x}' - \boldsymbol{x}||_p \leq \epsilon \right\}$. Just like Schuchardt et al. (2021), who encountered the same problem with their collective certificate, we circumvent this issue by first computing a set $\mathbb{L} \subseteq \mathbb{T}$ of all targeted predictions in $\mathbb{T}$ that are guaranteed to always be robust under the collective perturbation model:

$$\mathbb{L} = \left\{ n \in \mathbb{T} \,\middle|\, \left( \max_{x \in \mathbb{B}_{\boldsymbol{x}}} \sum_{d=1}^{D} w_d^{(n)} \cdot |x_d' - x_d|^p \right) < \eta^{(n)} \right\} \tag{26}$$

$$= \left\{ n \in \mathbb{T} \,\middle|\, \max_n \left\{ \boldsymbol{w}^{(n)} \right\} \cdot \epsilon^p < \eta^{(n)} \right\}. \tag{27}$$

The equality follows from the fact that the most effective way of attacking a prediction is to allocate all adversarial budget to the least robust dimension, i.e. the dimension with the largest weight. Because we know that all predictions with indices in $\mathbb{L}$ are robust, we do not have to include them in the collective optimization problem and can instead compute

$$|\mathbb{L}| + \min_{\boldsymbol{x}' \in \mathbb{B}_{\boldsymbol{x}}} \sum_{n \in \mathbb{T} \setminus \mathbb{L}} \mathrm{I}\left[ \boldsymbol{x}' \in \mathbb{H}^{(n)} \right]. \tag{28}$$

The r.h.s. optimization can be solved using the general collective certificate from Theorem 4.2 or any of the more efficient, modified certificates from previous sections.

When using the general collective certificate from Theorem 4.2 with binary data, the budget variables $\boldsymbol{b} \in \{0,1\}^{D_{\text{in}}}$ can be relaxed to $\boldsymbol{b} \in [0,1]^{D_{\text{in}}}$. When using the modified collective certificate from § E.3, the budget variables with $\boldsymbol{b} \in \mathbb{N}_0^{N_{\text{in}}}$ can be relaxed to $\boldsymbol{b} \in \mathbb{R}_+^{N_{\text{in}}}$. The additional constraint $\forall l \in \{1,\ldots,N_{\text{in}}\} : b_l \leq \left| \mathbb{J}^{(l)} \right|$ can be kept in order to model that the adversary cannot flip (or partially flip) more bits than are present within each input subset $\mathbb{J}^{(l)}$.

# F  BASE CERTIFICATES

In the following, we show why the base certificates discussed in § 5 and summarized in Table 1 hold. In § F.3.2 we further present a base certificate (and corresponding collective certificate) that can distinguish between adversarial addition and deletion of bits in binary data.

## F.1  GAUSSIAN SMOOTHING FOR $l_2$ PERTURBATIONS OF CONTINUOUS DATA

**Proposition F.1.** *Given an output* $g_n : \mathbb{R}^{D_{\text{in}}} \to \mathbb{Y}$, *let* $f_n(\boldsymbol{x}) = \text{argmax}_{y \in \mathbb{Y}} \Pr_{\boldsymbol{z} \sim \mathcal{N}(\boldsymbol{x}, \boldsymbol{\Sigma})} [g_n(\boldsymbol{z}) = y]$ *be the corresponding smoothed output with* $\boldsymbol{\Sigma} = \text{diag}(\boldsymbol{\sigma})^2$ *and* $\boldsymbol{\sigma} \in \mathbb{R}_+^{D_{\text{in}}}$. *Given an input* $\boldsymbol{x} \in \mathbb{R}^{D_{\text{in}}}$ *and smoothed prediction* $y_n = f_n(\boldsymbol{x})$, *let* $q = \Pr_{\boldsymbol{z} \sim \mathcal{N}(\boldsymbol{x}, \boldsymbol{\Sigma})} [g_n(\boldsymbol{z}) = y_n]$. *Then,* $\forall \boldsymbol{x}' \in \mathbb{H}^{(n)} : f_n(\boldsymbol{x}') = y_n$ *with* $\mathbb{H}^{(n)}$ *defined as in Eq. 2,* $w_d = \frac{1}{\sigma_d^2}$, $\eta = \left(\Phi^{(-1)}(q)\right)^2$ *and* $p = 2$.

*Proof.* Based on the definition of the base certificate interface, we need to show that, $\forall \boldsymbol{x}' \in \mathbb{H} : f_n(\boldsymbol{x}') = y_n$ with

$$\mathbb{H} = \left\{ \boldsymbol{x}' \in \mathbb{R}^{D_{\text{in}}} \,\middle|\, \sum_{d=1}^{D_{\text{in}}} \frac{1}{\sigma_d^2} \cdot |x_d - x_d'|^2 < \left(\Phi^{-1}(q)\right)^2 \right\}. \tag{29}$$

Eiras et al. (2022) have shown that under the same conditions as above, but with a general covariance matrix $\boldsymbol{\Sigma} \in \mathbb{R}_+^{D_{\text{in}} \times D_{\text{in}}}$, a prediction $y_n$ is certifiably robust to a perturbed input $\boldsymbol{x}'$ if

$$\sqrt{(\boldsymbol{x} - \boldsymbol{x}')\boldsymbol{\Sigma}^{-1}(\boldsymbol{x} - \boldsymbol{x}')} < \frac{1}{2} \left(\Phi^{-1}(q) - \Phi^{-1}(q')\right), \tag{30}$$

where $q' = \max_{y_n' \neq y_n} \Pr_{\boldsymbol{z} \sim \mathcal{N}(\boldsymbol{x}, \boldsymbol{\Sigma})} [g_n(\boldsymbol{z}) = y_n']$ is the probability of the second most likely prediction under the smoothing distribution. Because the probabilities of all possible predictions have to sum up to 1, we have $q' \leq 1 - q$. Since $\Phi^{-1}$ is monotonically increasing, we can obtain a lower bound on the r.h.s. of Eq. 30 and thus a more pessimistic certificate by substituting $1 - q$ for $q'$ (deriving such a "binary certificate" from a "multiclass certificate" is common in randomized smoothing and was already discussed in (Cohen et al., 2019)):

$$\sqrt{(\boldsymbol{x} - \boldsymbol{x}')\boldsymbol{\Sigma}^{-1}(\boldsymbol{x} - \boldsymbol{x}')} < \frac{1}{2} \left(\Phi^{-1}(q) - \Phi^{-1}(1 - q)\right), \tag{31}$$

In our case, $\boldsymbol{\Sigma}$ is a diagonal matrix $\text{diag}(\boldsymbol{\sigma})^2$ with $\boldsymbol{\sigma} \in \mathbb{R}_+^{D_{\text{in}}}$. Thus Eq. 31 is equivalent to

$$\sqrt{\sum_{d=1}^{D_{\text{in}}} (x_d - x_d') \frac{1}{\sigma_d^2} (x_d - x_d')} < \frac{1}{2} \left(\Phi^{-1}(q) - \Phi^{-1}(1 - q)\right). \tag{32}$$

Finally, using the fact that $\Phi^{-1}(q) - \Phi^{-1}(1 - q) = 2\Phi^{-1}(q)$ and eliminating the square root shows that we are certifiably robust if

$$\sum_{d=1}^{D_{\text{in}}} \frac{1}{\sigma_d^2} \cdot |x_d - x_d'|^2 < \left(\Phi^{-1}(q)\right)^2. \tag{33}$$

$\square$

Table 1: Base certificates complying with interface Eq. 2 with parameters $\boldsymbol{w}^{(n)}$ and $\eta^{(n)}$. Here, $y_n = f_n(\boldsymbol{x})$ is the prediction of $f_n(\boldsymbol{x}) = \text{argmax}_{y \in \mathbb{Y}} q_{n,y}$. With the $l_0$ certificate, $g_n(\boldsymbol{z})_y$ refers to the softmax score of class $y$ and $\zeta = \text{Var}_{\boldsymbol{z} \sim \mathcal{F}(\boldsymbol{x}, \boldsymbol{\theta})} [g_n(\boldsymbol{z})_{y_n}]$ is the variance of $y_n$'s softmax score.

| Norm | $\Psi_{\boldsymbol{x}}^{(n)}$ | $q_{n,y}$ | $w_d^{(n)}$ | $\eta^{(n)}$ |
|------|---------------------------|-----------|-------------|--------------|
| $l_2$ | $\mathcal{N}\left(\boldsymbol{x}, \text{diag}(\boldsymbol{s})^2\right)$ | $\Pr_{\boldsymbol{z} \sim \Psi_{\boldsymbol{x}}^{(n)}} [g_n(\boldsymbol{z}) = y]$ | $\frac{1}{s_d^2}$ | $\left(\Phi^{-1}(q_{n,y_n})\right)^2$ |
| $l_1$ | $\mathcal{U}(\boldsymbol{x}, \boldsymbol{\lambda})$ | $\Pr_{\boldsymbol{z} \sim \Psi_{\boldsymbol{x}}^{(n)}} [g_n(\boldsymbol{z}) = y]$ | $\frac{1}{\lambda_d}$ | $\Phi^{-1}(q_{n,y_n})$ |
| $l_0$ | $\mathcal{F}(\boldsymbol{x}, \boldsymbol{\theta})$ | $\mathbb{E}_{\boldsymbol{z} \sim \Psi_{\boldsymbol{x}}^{(n)}} [g_n(\boldsymbol{z})_y]$ | $\ln\left(\frac{(1-\theta_d)^2}{\theta_d} + \frac{(\theta_d)^2}{1-\theta_d}\right)$ | $\ln\left(1 + \frac{1}{\zeta}\left(q_{n,y_n} - \frac{1}{2}\right)^2\right)$ |

### F.2 UNIFORM SMOOTHING FOR $l_1$ PERTURBATIONS OF CONTINUOUS DATA

An alternative base certificate for $l_1$ perturbations is again due to Eiras et al. (2022). Using uniform instead of Gaussian noise allows us to collective certify robustness to $l_1$-norm-bound perturbations. In the following $\mathcal{U}(\boldsymbol{x}, \boldsymbol{\lambda})$ with $\boldsymbol{x} \in \mathbb{R}^D$, $\boldsymbol{\lambda} \in \mathbb{R}_+^D$ refers to a vector-valued random distribution in which the $d$-th element is uniformly distributed in $[x_d - \lambda_d, x_d + \lambda_d]$.

**Proposition F.2.** *Given an output* $g_n : \mathbb{R}^{D_{\text{in}}} \to \mathbb{Y}$, *let* $f(\boldsymbol{x}) = \operatorname{argmax}_{y \in \mathbb{Y}} \Pr_{\boldsymbol{z} \sim \mathcal{U}(\boldsymbol{x}, \boldsymbol{\lambda})} [g(\boldsymbol{z}) = y]$ *be the corresponding smoothed classifier with* $\boldsymbol{\lambda} \in \mathbb{R}_+^{D_{\text{in}}}$. *Given an input* $\boldsymbol{x} \in \mathbb{R}^{D_{\text{in}}}$ *and smoothed prediction* $y = f(\boldsymbol{x})$, *let* $p = \Pr_{\boldsymbol{z} \sim \mathcal{U}(\boldsymbol{x}, \boldsymbol{\lambda})} [g(\boldsymbol{z}) = y]$. *Then,* $\forall \boldsymbol{x}' \in \mathbb{H}^{(n)} : f_n(\boldsymbol{x}') = y_n$ *with* $\mathbb{H}^{(n)}$ *defined as in Eq. 2,* $w_d = 1/\lambda_d$, $\eta = \Phi^{-1}(q)$ *and* $p = 1$.

*Proof.* Based on the definition of $\mathbb{H}^{(n)}$, we need to prove that $\forall \boldsymbol{x}' \in \mathbb{H} : f_n(\boldsymbol{x}') = y_n$ with

$$\mathbb{H} = \left\{ \boldsymbol{x}' \in \mathbb{R}^{D_{\text{in}}} \mid \sum_{d=1}^{D_{\text{in}}} \frac{1}{\lambda_d} \cdot |x_d - x_d'| < \Phi^{-1}(q) \right\}, \tag{34}$$

Eiras et al. (2022) have shown that under the same conditions as above, a prediction $y_n$ is certifiably robust to a perturbed input $\boldsymbol{x}'$ if

$$\sum_{d=1}^{D_{\text{in}}} |\frac{1}{\lambda_d} \cdot (x_d - x_d')| < \frac{1}{2} \left( \Phi^{-1}(q) - \Phi^{-1}(1 - q) \right), \tag{35}$$

where $q' = \max_{y_n' \neq y_n} \Pr_{\boldsymbol{z} \sim \mathcal{U}(\boldsymbol{x}, \boldsymbol{\lambda})} [g_n(\boldsymbol{z}) = y_n']$ is the probability of the second most likely prediction under the smoothing distribution. As in our previous proof for Gaussian smoothing, we can obtain a more pessimistic certificate by substituting $1 - q$ for $q'$. Since $\Phi^{-1}(q) - \Phi^{-1}(1 - q) = 2\Phi^{-1}(q)$ and all $\lambda_d$ are non-negative, we know that our prediction is certifiably robust if

$$\sum_{d=1}^{D_{\text{in}}} \frac{1}{\lambda_d} \cdot |x_d - x_d'| < \Phi^{-1}(p). \tag{36}$$

$\square$

### F.3 VARIANCE-CONSTRAINED CERTIFICATION

In the following, we derive the general variance-constrained randomized smoothing certificate from Theorem 5.1, before discussing specific certificates for binary data in § F.3.1 and § F.3.2.

Variance smoothing assumes that we make predictions by randomly smoothing a base model's softmax scores. That is, given base model $g : \mathbb{X} \to \Delta_{|\mathbb{Y}|}$ mapping from an arbitrary discrete input space $\mathbb{X}$ to scores from the $(|\mathbb{Y}| - 1)$-dimensional probability simplex $\Delta_{|\mathbb{Y}|}$, we define the smoothed classifier $f(\boldsymbol{x}) = \operatorname{argmax}_{y \in \mathbb{Y}} \mathbb{E}_{\boldsymbol{z} \sim \Psi(\boldsymbol{x})} [g(\boldsymbol{z})_y]$. Here, $\Psi(\boldsymbol{x})$ is an arbitrary distribution over $\mathbb{X}$ parameterized by $\boldsymbol{x}$, e.g a Normal distribution with mean $\boldsymbol{x}$. The smoothed classifier does not return the most likely prediction, but the prediction associated with the highest expected softmax score.

Given an input $\boldsymbol{x} \in \mathbb{X}$, smoothed prediction $y = f(\boldsymbol{x})$ and a perturbed input $\boldsymbol{x}' \in \mathbb{X}$, we want to determine whether $f(\boldsymbol{x}') = y$. By definition of our smoothed classifier, we know that $f(\boldsymbol{x}') = y$ if $y$ is the label with the highest expected softmax score. In particular, we know that $f(\boldsymbol{x}') = y$ if $y$'s softmax score is larger than all other softmax scores combined, i.e.

$$\mathbb{E}_{\boldsymbol{z} \sim \Psi(\boldsymbol{x}')} [g(\boldsymbol{z})_y] > 0.5 \implies f(\boldsymbol{x}') = y. \tag{37}$$

Computing $\mathbb{E}_{\boldsymbol{z} \sim \Psi(\boldsymbol{x}')} [g(\boldsymbol{z})_y]$ exactly is usually not tractable – especially if we later want to evaluate robustness to many $\boldsymbol{x}'$ from a whole perturbation model $\mathbb{B} \subseteq \mathbb{X}$. Therefore, we compute a lower bound on $\mathbb{E}_{\boldsymbol{z} \sim \Psi(\boldsymbol{x}')} [g(\boldsymbol{z})_y]$. If even this lower bound is larger than $0.5$, we know that prediction $y$ is certainly robust. For this, we define a set of functions $\mathbb{F}$ with $g_y \in \mathbb{H}$ and compute the minimum softmax score across all functions from $\mathbb{F}$:

$$\min_{h \in \mathbb{F}} \mathbb{E}_{\boldsymbol{z} \sim \Psi(\boldsymbol{x}')} [h(\boldsymbol{z})] > 0.5 \implies f(\boldsymbol{x}') = y. \tag{38}$$

For our variance smoothing approach, we define $\mathbb{F}$ to be the set of all functions that have a larger or equal expected value and a smaller or equal variance under $\Psi(\boldsymbol{x})$, compared to our base model $g$. Let $\mu = \mathbb{E}_{\boldsymbol{z} \sim \Psi(\boldsymbol{x})} [g(\boldsymbol{z})_y]$ be the expected softmax score of our base model $g$ for label $y$. Let $\zeta = \mathbb{E}_{\boldsymbol{z} \sim \Psi(\boldsymbol{x})} \left[ (g(\boldsymbol{z})_y - \nu)^2 \right]$ be the expected squared distance of the softmax score from a scalar $\nu \in \mathbb{R}$. (Choosing $\nu = \mu$ yields the variance of the softmax score. An arbitrary $\nu$ is only needed for technical reasons related to Monte Carlo estimation § G.2). Then, we define

$$\mathbb{F} = \left\{ h : \mathbb{X} \to \mathbb{R} \,\middle|\, \mathbb{E}_{\boldsymbol{z} \sim \Psi(\boldsymbol{x})} [h(\boldsymbol{z})] \geq \mu \wedge \mathbb{E}_{\boldsymbol{z} \sim \Psi(\boldsymbol{x})} \left[ (h(\boldsymbol{z}) - \nu)^2 \right] \leq \zeta \right\} \tag{39}$$

Clearly, by the definition of $\mu$ and $\zeta$, we have $g_y \in \mathbb{F}$. Note that we do not restrict functions from $\mathbb{H}$ to the domain $[0, 1]$, but allow arbitrary real-valued outputs.

By evaluating Eq. 37 with $\mathbb{F}$ defined as in Eq. 38, we can determine if our prediciton is robust. To compute the optimal value, we need the following two Lemmata:

**Lemma F.3.** *Given a discrete set $\mathbb{X}$ and the set $\Pi$ of all probability mass functions over $\mathbb{X}$, any two probability mass functions $\pi_1$, $\pi_2 \in \Pi$ fulfill*

$$\sum_{z \in \mathbb{X}} \frac{\pi_2(z)}{\pi_1(z)} \pi_2(z) \geq 1. \tag{40}$$

*Proof.* For a fixed probability mass function $\pi_1$, Eq. 40 is lower-bounded by the minimal expected likelihood ratio that can be achieved by another $\tilde{\pi}(z) \in \Pi$:

$$\sum_{z \in \mathbb{X}} \frac{\pi_2(z)}{\pi_1(z)} \pi_2(z) \geq \min_{\tilde{\pi} \in \Pi} \sum_{z \in \mathbb{X}} \frac{\tilde{\pi}(z)}{\pi_1(z)} \tilde{\pi}(z). \tag{41}$$

The r.h.s. term can be expressed as the constrained optimization problem

$$\min_{\tilde{\pi}} \sum_{z \in \mathbb{X}} \frac{\tilde{\pi}(z)}{\pi_1(z)} \tilde{\pi}(z) \quad \text{s.t.} \quad \sum_{z \in \mathbb{X}} \tilde{\pi}(z) = 1 \tag{42}$$

with the corresponding dual problem

$$\max_{\lambda \in \mathbb{R}} \min_{\tilde{\pi}} \sum_{z \in \mathbb{X}} \frac{\tilde{\pi}(z)}{\pi_1(z)} \tilde{\pi}(z) + \lambda \left( -1 + \sum_{z \in \mathbb{X}} \tilde{\pi}(z) \right). \tag{43}$$

The inner problem is convex in each $\tilde{\pi}(z)$. Taking the gradient w.r.t. to $\tilde{\pi}(z)$ for all $z \in \mathbb{X}$ shows that it has its minimum at $\forall z \in \mathbb{X} : \tilde{\pi}(z) = -\frac{\lambda \pi_1(z)}{2}$. Substituting into Eq. 43 results in

$$\max_{\lambda \in \mathbb{R}} \sum_{z \in \mathbb{X}} \frac{\lambda^2 \pi_1(z)^2}{4 \pi_1(z)} + \lambda \left( -1 - \sum_{z \in \mathbb{X}} \frac{\lambda \pi_1(z)}{2} \right) \tag{44}$$

$$= \max_{\lambda \in \mathbb{R}} -\lambda^2 \sum_{z \in \mathbb{X}} \frac{\pi_1(z)}{4} - \lambda \tag{45}$$

$$= \max_{\lambda \in \mathbb{R}} -\frac{\lambda^2}{4} - \lambda \tag{46}$$

$$= 1. \tag{47}$$

Eq. 46 follows from the fact that $\pi_1(z)$ is a valid probability mass function. Due to duality, the optimal dual value 1 is a lower bound on the optimal value of our primal problem Eq. 40. $\square$

**Lemma F.4.** *Given a probability distribution $\mathcal{D}$ over a $\mathbb{R}$ and a scalar $\nu \in \mathbb{R}$, let $\mu = \mathbb{E}_{z \sim \mathcal{D}} [z]$ and $\xi = \mathbb{E}_{z \sim \mathcal{D}} \left[ (z - \nu)^2 \right]$. Then $\xi \geq (\mu - \nu)^2$*

*Proof.* Using the definitions of $\mu$ and $\xi$, as well as some simple algebra, we can show:

$$\xi \geq (\mu - \nu)^2 \tag{48}$$

$$\iff \mathbb{E}_{z\sim\mathcal{D}}\left[(z-\nu)^2\right] \geq \mu^2 - 2\mu\nu + \nu^2 \tag{49}$$

$$\iff \mathbb{E}_{z\sim\mathcal{D}}\left[z^2 - 2z\nu + \nu^2\right] \geq \mu^2 - 2\mu\nu + \nu^2 \tag{50}$$

$$\iff \mathbb{E}_{z\sim\mathcal{D}}\left[z^2 - 2z\nu + \nu^2\right] \geq \mu^2 - 2\mu\nu + \nu^2 \tag{51}$$

$$\iff \mathbb{E}_{z\sim\mathcal{D}}\left[z^2\right] - 2\mu\nu + \nu^2 \geq \mu^2 - 2\mu\nu + \nu^2 \tag{52}$$

$$\iff \mathbb{E}_{z\sim\mathcal{D}}\left[z^2\right] \geq \mu^2 \tag{53}$$

It is well known for the variance that $\mathbb{E}_{z\sim\mathcal{D}}\left[(z-\mu)^2\right] = \mathbb{E}_{z\sim\mathcal{D}}\left[z^2\right] - \mu^2$. Because the variance is always non-negative, the above inequality holds. $\square$

Using the previously described approach and lemmata, we can show the soundness of the following robustness certificate:

*Theorem* 5.1 (Variance-constrained certification). a function $g : \mathbb{X} \to \Delta_{|\mathbb{Y}|}$ mapping from discrete set $\mathbb{X}$ to scores from the $(|\mathbb{Y}| - 1)$-dimensional probability simplex, let $f(\boldsymbol{x}) = \arg\max_{y\in\mathbb{Y}} \mathbb{E}_{\boldsymbol{z}\sim\Psi_{\boldsymbol{x}}}\left[g(\boldsymbol{z})_y\right]$ with smoothing distribution $\Psi_{\boldsymbol{x}}$ and probability mass function $\pi_{\boldsymbol{x}}(\boldsymbol{z}) = \Pr_{\tilde{\boldsymbol{z}}\sim\Psi_{\boldsymbol{x}}}\left[\tilde{\boldsymbol{z}} = \boldsymbol{z}\right]$. Given an input $\boldsymbol{x} \in \mathbb{X}$ and smoothed prediction $y = f(\boldsymbol{x})$, let $\mu = \mathbb{E}_{\boldsymbol{z}\sim\Psi_{\boldsymbol{x}}}\left[g(\boldsymbol{z})_y\right]$ and $\zeta = \mathbb{E}_{\boldsymbol{z}\sim\Psi_{\boldsymbol{x}}}\left[(g(\boldsymbol{z})_y - \nu)^2\right]$ with $\nu \in \mathbb{R}$. Assuming $\nu \leq \mu$, then $f(\boldsymbol{x}') = y$ if

$$\sum_{\boldsymbol{z}\in\mathbb{X}} \frac{\pi_{\boldsymbol{x}'}(\boldsymbol{z})}{\pi_{\boldsymbol{x}}(\boldsymbol{z})} \cdot \pi_{\boldsymbol{x}'}(\boldsymbol{z}) < 1 + \frac{1}{\zeta - (\mu - \nu)^2}\left(\mu - \frac{1}{2}\right). \tag{54}$$

*Proof.* Following our discussion above, we know that $f(\boldsymbol{x}') = y$ if $\mathbb{E}_{\boldsymbol{z}\sim\Psi(\boldsymbol{x}')}\left[g(\boldsymbol{z})_y\right] > 0.5$ with $\mathbb{F}$ defined as in Eq. 39. We can compute a (tight) lower bound on $\min_{h\in\mathbb{F}} \mathbb{E}_{\boldsymbol{z}\sim\Psi(\boldsymbol{x}')}$ by following the functional optimization approach for randomized smoothing proposed by Zhang et al. (2020). That is, we solve a dual problem in which we optimize the value $h(\boldsymbol{z})$ for each $\boldsymbol{z} \in \mathbb{X}$. By the definition of the set $\mathbb{F}$, our optimization problem is

$$\min_{h:\mathbb{X}\to\mathbb{R}} \mathbb{E}_{\boldsymbol{z}\sim\Psi(\boldsymbol{x}')}\left[h(\boldsymbol{z})\right] \tag{55}$$

$$\text{s.t.} \quad \mathbb{E}_{\boldsymbol{z}\sim\Psi(\boldsymbol{x})}\left[h(\boldsymbol{z})\right] \geq \mu, \quad \mathbb{E}_{\boldsymbol{z}\sim\Psi(\boldsymbol{x})}\left[(h(\boldsymbol{z}) - \nu)^2\right] \leq \zeta. \tag{56}$$

The corresponding dual problem with dual variables $\alpha, \beta \geq 0$ is

$$\max_{\alpha,\beta\geq 0} \min_{h:\mathbb{X}\to\mathbb{R}} \mathbb{E}_{\boldsymbol{z}\sim\Psi(\boldsymbol{x}')}\left[h(\boldsymbol{z})\right]$$
$$+ \alpha\left(\mu - \mathbb{E}_{\boldsymbol{z}\sim\Psi(\boldsymbol{x})}\left[h(\boldsymbol{z})\right]\right) + \beta\left(\mathbb{E}_{\boldsymbol{z}\sim\Psi(\boldsymbol{x})}\left[(h(\boldsymbol{z}) - \nu)^2\right] - \zeta\right). \tag{57}$$

We first move move all terms that don't involve $h$ out of the inner optimization problem:

$$= \max_{\alpha,\beta\geq 0} \alpha\mu - \beta\zeta + \min_{h:\mathbb{X}\to\mathbb{R}} \mathbb{E}_{\boldsymbol{z}\sim\Psi(\boldsymbol{x}')}\left[h(\boldsymbol{z})\right] - \alpha\mathbb{E}_{\boldsymbol{z}\sim\Psi(\boldsymbol{x})}\left[h(\boldsymbol{z})\right] + \beta\mathbb{E}_{\boldsymbol{z}\sim\Psi(\boldsymbol{x})}\left[(h(\boldsymbol{z}) - \nu)^2\right]. \tag{58}$$

Writing out the expectation terms and combining them into one sum (or – in the case of continuous $\mathbb{X}$ – one integral), our dual problem becomes

$$= \max_{\alpha,\beta\geq 0} \alpha\mu - \beta\zeta + \min_{h:\mathbb{X}\to\mathbb{R}} \sum_{\boldsymbol{z}\in\mathbb{X}} h(\boldsymbol{z})\pi_{\boldsymbol{x}'}(\boldsymbol{z}) - \alpha h(\boldsymbol{z})\pi_{\boldsymbol{x}}(\boldsymbol{z}) + \beta(h(\boldsymbol{z}) - \nu)^2 \pi_{\boldsymbol{x}}(\boldsymbol{z}) \tag{59}$$

(recall that $\pi_{\boldsymbol{x}'}$ and $\pi_{\boldsymbol{x}'}$ refer to the probability mass functions of the smoothing distributions). The inner optimization problem can be solved by finding the optimal $h(\boldsymbol{z})$ in each point $\boldsymbol{z}$:

$$= \max_{\alpha,\beta\geq 0} \alpha\mu - \beta\zeta + \sum_{\boldsymbol{z}\in\mathbb{X}} \min_{h(\boldsymbol{z})\in\mathbb{R}} h(\boldsymbol{z})\pi_{\boldsymbol{x}'}(\boldsymbol{z}) - \alpha h(\boldsymbol{z})\pi_{\boldsymbol{x}}(\boldsymbol{z}) + \beta(h(\boldsymbol{z}) - \nu)^2 \pi_{\boldsymbol{x}}(\boldsymbol{z}). \tag{60}$$

Because $\beta \geq 0$, each inner optimization problem is convex in $h(\boldsymbol{z})$. We can thus find the optimal $h^*(\boldsymbol{z})$ by setting the derivative to zero:

$$\frac{d}{dh(\boldsymbol{z})}h(\boldsymbol{z})\pi_{\boldsymbol{x}'}(\boldsymbol{z}) - \alpha h(\boldsymbol{z})\pi_{\boldsymbol{x}}(\boldsymbol{z}) + \beta\left(h(\boldsymbol{z}) - \nu\right)^2\pi_{\boldsymbol{x}}(\boldsymbol{z}) \overset{!}{=} 0 \tag{61}$$

$$\iff \pi_{\boldsymbol{x}'}(\boldsymbol{z}) - \alpha\pi_{\boldsymbol{x}}(\boldsymbol{z}) + 2\beta\left(h(\boldsymbol{z}) - \nu\right)\pi_{\boldsymbol{x}}(\boldsymbol{z}) \overset{!}{=} 0 \tag{62}$$

$$\implies h^*(\boldsymbol{z}) = -\frac{\pi_{\boldsymbol{x}'}(\boldsymbol{z})}{2\beta\pi_{\boldsymbol{x}}(\boldsymbol{z})} + \frac{\alpha}{2\beta} + \nu. \tag{63}$$

Substituting into Eq. 59 and simplifying leaves us with the dual problem

$$\max_{\alpha,\beta\geq 0} \alpha\mu - \beta\zeta - \frac{\alpha^2}{4\beta} + \frac{\alpha}{2\beta} - \alpha\nu + \nu - \frac{1}{4\beta}\sum_{\boldsymbol{z}\in\mathbb{X}}\frac{\pi_{\boldsymbol{x}'}(\boldsymbol{z})^2}{\pi_{\boldsymbol{x}}(\boldsymbol{z})}. \tag{64}$$

In the following, let us use $\rho = \sum_{\boldsymbol{z}\in\mathbb{X}}\frac{\pi_{\boldsymbol{x}'}(\boldsymbol{z})^2}{\pi_{\boldsymbol{x}}(\boldsymbol{z})}$ as a shorthand for the expected likelihood ratio. The problem is concave in $\alpha$. We can thus find the optimum $\alpha^*$ by setting the derivative to zero, which gives us $\alpha^* = 2\beta(\mu - \nu) + 1$. Because $\beta \geq 0$ and our theorem assumes that $\nu \leq \mu$, the value $\alpha^*$ is a feasible solution to the dual problem. Substituting into Eq. 64 and simplifying results in

$$\max_{\beta\geq 0} \alpha^*\mu - \beta\zeta - \frac{\alpha^{*2}}{4\beta} + \frac{\alpha^*}{2\beta} - \alpha^*\nu + \nu - \frac{1}{4\beta}\rho \tag{65}$$

$$= \max_{\beta\geq 0} \beta\left((\mu - \nu)^2 - \sigma^2\right) + \mu + \frac{1}{4\beta}\left(1 - \rho\right). \tag{66}$$

Lemma F.3 shows that the expected likelihood ratio $\rho$ is always greater than or equal to 1. Lemma F.4 shows that $(\mu - \nu)^2 - \sigma^2 \leq 0$. Therefore Eq. 66 is concave in $\beta$. The optimal value of $\beta$ can again be found by setting the derivative to zero:

$$\beta^* = \sqrt{\frac{1 - \rho}{4\left((\mu - \nu)^2 - \sigma^2\right)}}. \tag{67}$$

Recall that our theorem assumes $\sigma^2 \geq (\mu - \nu)^2$ and thus $\beta^*$ is real valued. Substituting Eq. 67 into Eq. 66 shows that the maximum of our dual problem is

$$\mu + \sqrt{(1 - p)\left((\mu - \nu)^2 - \sigma^2\right)}. \tag{68}$$

By duality, this is a lower bound on our primal problem $\min_{h\in\mathbb{F}}\mathbb{E}_{\boldsymbol{z}\sim\Psi(\boldsymbol{x}')}[h(\boldsymbol{z})]$. We know that our prediction is certifiably robust, i.e. $f(\boldsymbol{x}) = y$, if $\min_{h\in\mathbb{F}}\mathbb{E}_{\boldsymbol{z}\sim\Psi(\boldsymbol{x}')}[h(\boldsymbol{z})] > 0.5$. So, in particular, our prediction is robust if

$$\mu + \sqrt{(1 - \rho)\left((\mu - \nu)^2 - \sigma^2\right)} > 0.5 \tag{69}$$

$$\iff \rho < 1 + \frac{1}{\sigma^2 - (\mu - \nu)^2}\left(\mu - \frac{1}{2}\right)^2 \tag{70}$$

$$\iff \sum_{\boldsymbol{z}\in\mathbb{X}}\frac{\pi_{\boldsymbol{x}'}(\boldsymbol{z})^2}{\pi_{\boldsymbol{x}}(\boldsymbol{z})} < 1 + \frac{1}{\sigma^2 - (\mu - \nu)^2}\left(\mu - \frac{1}{2}\right)^2 \tag{71}$$

The last equivalence is the result of inserting the definition of the expected likelihood ratio $\rho$. □

With Theorem 5.1 in place, we can certify robustness for arbitrary smoothing distributions, assuming we can compute the expected likelihood ratio. When we are working with discrete data and the smoothing distributions factorize, this can be done efficiently, as the two following base certificates for binary data demonstrate.

### F.3.1 Bernoulli Smoothing for Perturbations of Binary Data

We begin by proving the base certificate presented in § 5. Recall that we we use a smoothing distribution $\mathcal{F}(\boldsymbol{x}, \boldsymbol{\theta})$ with $\boldsymbol{\theta} \in [0, 1]^{D_{\text{in}}}$ that independently flips the $d$'th bit with probability $\theta_d$, i.e. for $\boldsymbol{x}, \boldsymbol{z} \in \{0, 1\}^{D_{\text{in}}}$ and $\boldsymbol{z} \sim \mathcal{F}(\boldsymbol{x}, \boldsymbol{\theta})$ we have $\Pr[z_d \neq x_d] = \theta_d$.

**Corollary F.5.** *Given an output $g_n : \{0,1\}^{D_{\text{in}}} \to \Delta_{|\mathbb{Y}|}$ mapping to scores from the $(|\mathbb{Y}| - 1)$-dimensional probability simplex, let $f_n(\boldsymbol{x}) = \operatorname{argmax}_{y \in \mathbb{Y}} \mathbb{E}_{\boldsymbol{z} \sim \mathcal{F}(\boldsymbol{x},\boldsymbol{\theta})} [g_n(\boldsymbol{z})_y]$ be the corresponding smoothed classifier with $\boldsymbol{\theta} \in [0,1]^{D_{\text{in}}}$. Given an input $\boldsymbol{x} \in \{0,1\}^{D_{\text{in}}}$ and smoothed prediction $y_n = f_n(\boldsymbol{x})$, let $\mu = \mathbb{E}_{\boldsymbol{z} \sim \mathcal{F}(\boldsymbol{x},\boldsymbol{\theta})} [g_n(\boldsymbol{z})_y]$ and $\zeta = \operatorname{Var}_{\boldsymbol{z} \sim \mathcal{F}(\boldsymbol{x},\boldsymbol{\theta})} [g_n(\boldsymbol{z})_y]$. Then, $\forall \boldsymbol{x}' \in \mathbb{H}^{(n)}$ : $f_n(\boldsymbol{x}') = y_n$ with $\mathbb{H}^{(n)}$ defined as in Eq. 2, $w_d = \ln \left( \frac{(1-\theta_d)^2}{\theta_d} + \frac{(\theta_d)^2}{1-\theta_d} \right)$, $\eta = \ln \left( 1 + \frac{1}{\zeta} \left( \mu - \frac{1}{2} \right)^2 \right)$ and $p = 0$.*

*Proof.* Based on our definition of the base certificates interface (see Definition 4.1, we must show that $\forall \boldsymbol{x}' \in \mathbb{H} : f_n(\boldsymbol{x}') = y_n$ with

$$\mathbb{H} = \left\{ \boldsymbol{x}' \in \{0,1\}^{D_{\text{in}}} \;\middle|\; \sum_{d=1}^{D_{\text{in}}} \ln \left( \frac{(1-\theta_d)^2}{\theta_d} + \frac{(\theta_d)^2}{1-\theta_d} \right) \cdot |x'_d - x_d|^0 < \ln \left( 1 + \frac{1}{\zeta} \left( \mu - \frac{1}{2} \right)^2 \right) \right\}, \tag{72}$$

Because all bits are flipped independently, our probability mass function $\pi_{\boldsymbol{x}}(\boldsymbol{z}) = \operatorname{Pr}_{\tilde{\boldsymbol{z}} \sim \Psi(\boldsymbol{x})} [\tilde{\boldsymbol{z}} = \boldsymbol{z}]$ factorizes:

$$\pi_{\boldsymbol{x}}(\boldsymbol{z}) = \prod_{d=1}^{D_{\text{in}}} \pi_{x_d}(z_d) \tag{73}$$

with

$$\pi_{x_d}(z_d) = \begin{cases} \theta_d & \text{if } z_d \neq x_d \\ 1 - \theta_d & \text{else} \end{cases}. \tag{74}$$

Thus, our expected likelihood ratio can be written as

$$\sum_{\boldsymbol{z} \in \{0,1\}^{D_{\text{in}}}} \frac{\pi_{\boldsymbol{x}'}(\boldsymbol{z})^2}{\pi_{\boldsymbol{x}}(\boldsymbol{z})} = \sum_{\boldsymbol{z} \in \{0,1\}^{D_{\text{in}}}} \prod_{d=1}^{D_{\text{in}}} \frac{\pi_{x'_d}(z_d)^2}{\pi_{x_d}(z_d)} = \prod_{d=1}^{D_{\text{in}}} \sum_{z_d \in \{0,1\}} \frac{\pi_{x'_d}(z_d)^2}{\pi_{x_d}(z_d)}. \tag{75}$$

For each dimension $d$, we can distinguish two cases: If both the perturbed and unperturbed input are the same in dimension $d$, i.e. $x'_d = x_d$, then $\frac{\pi_{x'_d}(\boldsymbol{z})}{\pi_{x_d}(\boldsymbol{z})} = 1$ and thus

$$\sum_{z_d \in \{0,1\}} \frac{\pi_{x'_d}(z_d)^2}{\pi_{x_d}(z_d)} = \sum_{z_d \in \{0,1\}} \pi_{x'_d}(z_d) = \theta_d + (1 - \theta_d) = 1. \tag{76}$$

If the perturbed and unperturbed input differ in dimension $d$, then

$$\sum_{z_d \in \{0,1\}} \frac{\pi_{x'_d}(z_d)^2}{\pi_{x_d}(z_d)} = \frac{(1-\theta_d)^2}{\theta_d} + \frac{(\theta_d)^2}{1-\theta_d}. \tag{77}$$

Therefore, the expected likelihood ratio is

$$\prod_{d=1}^{D_{\text{in}}} \sum_{z_d \in \{0,1\}} \frac{\pi_{x'_d}(z_d)^2}{\pi_{x_d}(z_d)} = \prod_{d=1}^{D_{\text{in}}} \left( \frac{(1-\theta_d)^2}{\theta_d} + \frac{(\theta_d)^2}{1-\theta_d} \right)^{|x'_d - x_d|}. \tag{78}$$

Due to Theorem 5.1 (and using $\nu = \mu$ when computing the variance), we know that our prediction is robust, i.e. $f_n(\boldsymbol{x}') = y_n$, if

$$\sum_{\boldsymbol{z} \in \{0,1\}^{D_{\text{in}}}} \frac{\pi_{\boldsymbol{x}'}(\boldsymbol{z})^2}{\pi_{\boldsymbol{x}}(\boldsymbol{z})} < 1 + \frac{1}{\zeta} \left( \mu - \frac{1}{2} \right)^2 \tag{79}$$

$$\iff \prod_{d=1}^{D_{\text{in}}} \left( \frac{(1-\theta_d)^2}{\theta_d} + \frac{(\theta_d)^2}{1-\theta_d} \right)^{|x'_d - x_d|} < 1 + \frac{1}{\zeta} \left( \mu - \frac{1}{2} \right)^2 \tag{80}$$

$$\iff \sum_{d=1}^{D_{\text{in}}} \ln \left( \frac{(1-\theta_d)^2}{\theta_d} + \frac{(\theta_d)^2}{1-\theta_d} \right) |x'_d - x_d| < \ln \left( 1 + \frac{1}{\zeta} \left( \mu - \frac{1}{2} \right)^2 \right). \tag{81}$$

Because $x_d$ and $x'_d$ are binary, the last inequality is equivalent to

$$\sum_{d=1}^{D_{\text{in}}} \ln \left( \frac{(1-\theta_d)^2}{\theta_d} + \frac{(\theta_d)^2}{1-\theta_d} \right) |x'_d - x_d|^0 < \ln \left( 1 + \frac{1}{\zeta} \left( \mu - \frac{1}{2} \right)^2 \right). \tag{82}$$

$\square$

### F.3.2 Sparsity-aware Smoothing for Perturbations of Binary Data

Sparsity-aware randomized smoothing (Bojchevski et al., 2020) is an alternative smoothing approach for binary data. It uses different probabilities for randomly deleting $(1 \rightarrow 0)$ and adding $(0 \rightarrow 1)$ bits to preserve data sparsity. For a random variable $z$ distributed according to the sparsity-aware distribution $\mathcal{S}(x, \theta^+, \theta^-)$ with $x \in \{0, 1\}^{D_{\text{in}}}$ and addition and deletion probabilities $\theta^+, \theta^- \in [0, 1]^{D_{\text{in}}}$, we have:

$$\Pr[z_d = 0] = \left(1 - \theta_d^+\right)^{1-x_d} \cdot \left(\theta_d^-\right)^{x_d},$$
$$\Pr[z_d = 1] = \left(\theta_d^+\right)^{1-x_d} \cdot \left(1 - \theta_d^-\right)^{x_d}.$$

The Bernoulli smoothing distribution we discussed in the previous section is a special case of sparsity-aware smoothing with $\theta^+ = \theta^-$. The runtime of the robustness certificate derived by Bojchevski et al. (2020) increases exponentially with the number of unique values in $\theta^+$ and $\theta^-$, which makes it unsuitable for localized smoothing. Variance-constrained smoothing, on the other hand, allows us to efficiently compute a certificate in closed form.

**Corollary F.6.** *Given an output $g_n : \mathbb{R}^{D_{\text{in}}} \rightarrow \Delta_{|\mathbb{Y}|}$ mapping to scores from the $(|\mathbb{Y}| - 1)$-dimensional probability simplex, let $f_n(x) = \text{argmax}_{y \in \mathbb{Y}} \mathbb{E}_{z \sim \mathcal{S}(x, \theta^+, \theta^-)} [g_n(z)_y]$ be the corresponding smoothed classifier with $\theta^+, \theta^- \in [0, 1]^{D_{\text{in}}}$. Given an input $x \in \{0, 1\}^{D_{\text{in}}}$ and smoothed prediction $y_n = f_n(x)$, let $\mu = \mathbb{E}_{z \sim \mathcal{S}(x, \theta^+, \theta^-)} [g_n(z)_y]$ and $\zeta = \text{Var}_{z \sim \mathcal{S}(x, \theta^+, \theta^-)} [g_n(z)_y]$. Then, $\forall x' \in \mathbb{H} : f_n(x') = y_n$ for*

$$\mathbb{H} = \left\{ x' \in \{0, 1\}^{D_{\text{in}}} \mid \sum_{d=1}^{D_{\text{in}}} \gamma_d^+ \cdot \text{I}\left[x_d = 0 \neq x_d'\right] + \gamma_d^- \cdot \text{I}\left[x_d = 1 \neq x_d'\right] < \eta \right\}, \quad (83)$$

*where $\gamma^+, \gamma^- \in \mathbb{R}^{D_{\text{in}}}$, $\gamma_d^+ = \ln\left(\frac{\left(\theta_d^-\right)^2}{1-\theta_d^+} + \frac{\left(1-\theta_d^-\right)^2}{\theta_d^+}\right)$, $\gamma_d^- = \ln\left(\frac{\left(1-\theta_d^+\right)^2}{\theta_d^-} + \frac{\left(\theta_d^+\right)^2}{1-\theta_d^-}\right)$ and $\eta = \ln\left(1 + \frac{1}{\zeta}\left(\mu - \frac{1}{2}\right)^2\right)$.*

*Proof.* Just like with the Bernoulli distribution we discussed in the previous section, all bits are flipped independently, meaning our probability mass function $\pi_x(z) = \Pr_{\tilde{z} \sim \Psi(x)} [\tilde{z} = z]$ factorizes:

$$\pi_x(z) = \prod_{d=1}^{D_{\text{in}}} \pi_{x_d}(z_d) \quad (84)$$

with

$$\pi_{x_d}(z_d) = \begin{cases} \theta_d & \text{if } z_d \neq x_d \\ 1 - \theta_d & \text{else} \end{cases}. \quad (85)$$

As before, our expected likelihood ratio can be written as

$$\sum_{z \in \{0,1\}^{D_{\text{in}}}} \frac{\pi_{x'}(z)^2}{\pi_x(z)} = \sum_{z \in \{0,1\}^{D_{\text{in}}}} \prod_{d=1}^{D_{\text{in}}} \frac{\pi_{x_d'}(z_d)^2}{\pi_{x_d}(z_d)} = \prod_{d=1}^{D_{\text{in}}} \sum_{z_d \in \{0,1\}} \frac{\pi_{x_d'}(z_d)^2}{\pi_{x_d}(z_d)}. \quad (86)$$

We can now distinguish three cases. If both the perturbed and unperturbed input are the same in dimension $d$, i.e. $x_d' = x_d$, then $\frac{\pi_{x_d'}(z)}{\pi_{x_d}(z)} = 1$ and thus

$$\sum_{z_d \in \{0,1\}} \frac{\pi_{x_d'}(z_d)^2}{\pi_{x_d}(z_d)} = \sum_{z_d \in \{0,1\}} \pi_{x_d'}(z_d) = 1. \quad (87)$$

If $x_d' = 1$ and $x_d = 0$, i.e. a bit was added, then

$$\sum_{z_d \in \{0,1\}} \frac{\pi_{x_d'}(z)^2}{\pi_{x_d}(z)} = \sum_{z_d \in \{0,1\}} \frac{\pi_1(z_d)^2}{\pi_0(z_d)} = \frac{\pi_1(0)^2}{\pi_0(0)} + \frac{\pi_1(1)^2}{\pi_0(1)} = \frac{\left(\theta_d^-\right)^2}{1-\theta_d^+} + \frac{\left(1-\theta_d^-\right)^2}{\theta_d^+} \quad (88)$$

If $x'_d = 0$ and $x_d = 1$, i.e. a bit was deleted, then

$$\sum_{z_d \in \{0,1\}} \frac{\pi_{x'_d}(\boldsymbol{z})^2}{\pi_{x_d}(\boldsymbol{z})} = \sum_{z_d \in \{0,1\}} \frac{\pi_0(z_d)^2}{\pi_1(z_d)} = \frac{\pi_0(0)^2}{\pi_1(0)} + \frac{\pi_0(1)^2}{\pi_1(1)} = \frac{\left(1 - \theta_d^+\right)^2}{\theta_d^-} + \frac{\left(\theta_d^+\right)^2}{1 - \theta_d^-}. \tag{89}$$

Therefore, the expected likelihood ratio is

$$\prod_{d=1}^{D_{\text{in}}} \sum_{z_d \in \{0,1\}} \frac{\pi_{x'_d}(z_d)^2}{\pi_{x_d}(z_d)} \tag{90}$$

$$= \prod_{d=1}^{D_{\text{in}}} \left( \frac{\left(\theta_d^-\right)^2}{1 - \theta_d^+} + \frac{\left(1 - \theta_d^-\right)^2}{\theta_d^+} \right)^{\text{I}\left[x_d = 0 \neq x'_d\right]} \left( \frac{\left(1 - \theta_d^+\right)^2}{\theta_d^-} + \frac{\left(\theta_d^+\right)^2}{1 - \theta_d^-} \right)^{\text{I}\left[x_d = 1 \neq x'_d\right]} \tag{91}$$

$$= \prod_{d=1}^{D_{\text{in}}} \exp\left(\gamma_d^+\right)^{\text{I}\left[x_d = 0 \neq x'_d\right]} \cdot \exp\left(\gamma_d^-\right)^{\text{I}\left[x_d = 1 \neq x'_d\right]}. \tag{92}$$

In the last equation, we have simply used the shorthands $\gamma_d^+$ and $\gamma_d^-$ defined in Corollary F.6. Due to Theorem 5.1 (and using $\nu = \mu$ when computing the variance), we know that our prediction is robust, i.e. $f_n(\boldsymbol{x}') = y_n$, if

$$\sum_{\boldsymbol{z} \in \{0,1\}^{D_{\text{in}}}} \frac{\pi_{\boldsymbol{x}'}(\boldsymbol{z})^2}{\pi_{\boldsymbol{x}}(\boldsymbol{z})} < 1 + \frac{1}{\zeta} \left( \mu - \frac{1}{2} \right)^2 \tag{93}$$

$$\iff \prod_{d=1}^{D_{\text{in}}} \exp\left(\gamma_d^+\right)^{\text{I}\left[x_d = 0 \neq x'_d\right]} \cdot \exp\left(\gamma_d^-\right)^{\text{I}\left[x_d = 1 \neq x'_d\right]} < 1 + \frac{1}{\zeta} \left( \mu - \frac{1}{2} \right)^2 \tag{94}$$

$$\iff \sum_{d=1}^{D_{\text{in}}} \gamma_d^+ \cdot \text{I}\left[x_d = 0 \neq x'_d\right] \cdot \gamma_d^- \cdot \text{I}\left[x_d = 1 \neq x'_d\right] < \ln\left(1 + \frac{1}{\zeta}\left(\mu - \frac{1}{2}\right)^2\right). \tag{95}$$

$\square$

**Use for collective certification.** It should be noted that this certificate does not comply with our interface for base certificates (see Definition 4.1), meaning we can not directly use it to certify robustness to norm-bound perturbations using our collective linear program from Theorem 4.2. We can however use it to certify collective robustness to the more refined threat model used in (Schuchardt et al., 2021): Let the set of admissible perturbed inputs be $\mathbb{B}_{\boldsymbol{x}} = \left\{ \boldsymbol{x}' \in \{0,1\}^{D_{\text{in}}} \mid \sum_{d=1}^{D_{\text{in}}} [x_d = 0 \neq x'_d] \leq \epsilon^+ \wedge \sum_{d=1}^{D_{\text{in}}} [x_d = 1 \neq x'_d] \leq \epsilon^- \right\}$ with $\epsilon^+, \epsilon^y \in \mathbb{N}_0$ specifying the number of bits the adversary is allowed to add or delete. We can now follow the procedure outlined in § 3.2 to combine the per-prediction base certificates into a collective certificate for our new collective perturbation model. As discussed in, we can bound the number of predictions that are robust to simultaneous attacks by minimizing the number of predictions that are certifiably robust according to their base certificates:

$$\min_{\boldsymbol{x}' \in \mathbb{B}_{\boldsymbol{x}}} \sum_{n \in \mathbb{T}} \text{I}\left[f_n(\boldsymbol{x}') = y_n\right] \geq \min_{\boldsymbol{x}' \in \mathbb{B}_{\boldsymbol{x}}} \sum_{n \in \mathbb{T}} \text{I}\left[\boldsymbol{x}' \in \mathbb{H}^{(n)}\right]. \tag{96}$$

Inserting the linear inequalities characterizing our perturbation model and base certificates results in:

$$\min_{\boldsymbol{x}' \in \{0,1\}^{D_{\text{in}}}} \sum_{n \in \mathbb{T}} \text{I}\left[\sum_{d=1}^{D_{\text{in}}} \gamma_d^+ \cdot \text{I}\left[x_d = 0 \neq x'_d\right] + \gamma_d^- \cdot \text{I}\left[x_d = 1 \neq x'_d\right] < \eta^{(n)}\right] \tag{97}$$

$$\text{s.t.} \quad \sum_{d=1}^{D_{\text{in}}} [x_d = 0 \neq x'_d] \leq \epsilon^+, \quad \sum_{d=1}^{D_{\text{in}}} [x_d = 1 \neq x'_d] \leq \epsilon^-. \tag{98}$$

Instead of optimizing over the perturbed input $\boldsymbol{x}'$, we can define two vectors $\boldsymbol{b}^+, \boldsymbol{b}- \in \{0,1\}^{D_{\text{in}}}$ that indicate in which dimension bits were added or deleted. Using these new variables, Eq. 97 can be rewritten as

$$\min_{\boldsymbol{b}^+, \boldsymbol{b}^- \in \{0,1\}^{D_{\text{in}}}} \sum_{n \in \mathbb{T}} \mathrm{I}\left[\left(\boldsymbol{\gamma}^+\right)^T \boldsymbol{b}^+ + \left(\boldsymbol{\gamma}^-\right)^T \boldsymbol{b}^- < \eta^{(n)}\right] \tag{99}$$

$$\text{s.t.} \quad \text{sum}\{\boldsymbol{b}^+\} \leq \epsilon^+, \quad \text{sum}\{\boldsymbol{b}^-\} \leq \epsilon^-, \tag{100}$$

$$\sum_{d|x_d=1} b_d^+ = 0, \quad \sum_{d|x_d=0} b_d^- = 0. \tag{101}$$

The last two constraints ensure that bits can only be deleted where $x_d = 1$ and bits can only be added where $x_d = 0$. Finally, we can use the procedure for replacing the indicator functions with indicator variables that we discussed in § D to restate the above problem as the mixed-integer problem

$$\min_{\boldsymbol{b}^+, \boldsymbol{b}^- \in \{0,1\}^{D_{\text{in}}}, \boldsymbol{t} \in \{0,1\}^{D_{\text{out}}}} \sum_{n \in \mathbb{T}} t_n \tag{102}$$

$$\text{s.t.} \quad \left(\boldsymbol{\gamma}^+\right)^T \boldsymbol{b}^+ + \left(\boldsymbol{\gamma}^-\right)^T \boldsymbol{b}^- \geq (1 - t_n)\eta^{(n)}, \tag{103}$$

$$\text{sum}\{\boldsymbol{b}^+\} \leq \epsilon^+, \quad \text{sum}\{\boldsymbol{b}^-\} \leq \epsilon^-, \tag{104}$$

$$\sum_{d|x_d=1} b_d^+ = 0, \quad \sum_{d|x_d=0} b_d^- = 0. \tag{105}$$

The first constraint ensures that $t_n$ can only be set to 0 if the l.h.s. is greater or equal $\eta_n$, i.e. only when the base certificate can no longer guarantee robustness. The efficiency of the certificate can be improved by applying any of the techniques discussed in § E.

### F.3.3 GAUSSIAN SMOOTHING FOR PERTURBATIONS OF CONTINUOUS DATA

Even though we specifically proposed variance-constrained certification as a means of efficiently certifying anisotropically smoothed classifiers for discrete data, it can be generalized to continuous distributions by replacing sums with integrals and mass functions with density functions (the proof is analogous to that in Appendix F.3).

In the following, we assume Gaussian smoothing, i.e. $\Psi(\boldsymbol{x}) \sim \mathcal{N}(\boldsymbol{x}, \boldsymbol{\Sigma})$ with $\boldsymbol{\Sigma} \in \mathbb{R}_+^{D \times D}$ with density function $\pi_{\boldsymbol{x}}$. In this case, the expected ratio between $\pi_{\boldsymbol{x}'}$ and $\pi_{\boldsymbol{x}}$ is the exponential of the squared Mahalanobis distance (see Table 2 of (Gil et al., 2013) with $\alpha = 2$), i.e.

$$\int_{\mathbb{R}^D} \frac{\pi_{\boldsymbol{x}'}(\boldsymbol{z})}{\pi_{\boldsymbol{x}}(\boldsymbol{z})} \pi_{\boldsymbol{x}'}(\boldsymbol{z}) \, d\boldsymbol{z} = \exp\left((\boldsymbol{x}' - \boldsymbol{x})\boldsymbol{\Sigma}^{-1}(\boldsymbol{x}' - \boldsymbol{x})\right).$$

This leads us to the following corollary of Theorem 5.1:

**Corollary F.7.** *Given a function $h : \mathbb{R}^D \to \Delta_{|\mathbb{Y}|}$ mapping to scores from the $(|\mathbb{Y}| - 1)$-dimensional probability simplex, let $f(\boldsymbol{x}) = \text{argmax}_{y \in \mathbb{Y}} \mathbb{E}_{\boldsymbol{z} \sim \mathcal{N}(\boldsymbol{x}, \boldsymbol{\Sigma})} [h(\boldsymbol{z})_y]$ with covariance matrix $\boldsymbol{\Sigma} \in \mathbb{R}_+^{D \times D}$. Given an input $\boldsymbol{x} \in \mathbb{X}$ and smoothed prediction $y = f(\boldsymbol{x})$, let $\mu = \mathbb{E}_{\boldsymbol{z} \sim \mathcal{N}(\boldsymbol{x}, \boldsymbol{\Sigma})} [h(\boldsymbol{z})_y]$ and $\zeta = \mathbb{E}_{\boldsymbol{z} \sim \mathcal{N}(\boldsymbol{x}, \boldsymbol{\Sigma})} \left[(h(\boldsymbol{z})_y - \nu)^2\right]$ with $\nu \in \mathbb{R}$. Assuming $\nu \leq \mu$, then $f(\boldsymbol{x}') = y$ if*

$$(\boldsymbol{x}' - \boldsymbol{x})\boldsymbol{\Sigma}^{-1}(\boldsymbol{x}' - \boldsymbol{x}) < \ln\left(1 + \frac{1}{\zeta - (\mu - \nu)^2}\left(\mu - \frac{1}{2}\right)\right). \tag{106}$$

As with Theorem 5.1, The r.h.s. of Eq. 106 depends on the expected softmax score $\mu$, a variable $\nu \leq \mu$ and the expected squared difference $\zeta$ between $\mu$ and $\nu$. For $\nu = \mu$ the parameter $\zeta$ is the variance of the softmax score. A higher expected value and a lower variance allow us to certify robustness for larger adversarial perturbations.

For comparison, ANCER (Eiras et al., 2022) guarantees robustness for the smoothed prediction $y_n = \text{argmax}_{y \in \mathbb{Y}} \Pr[g(\boldsymbol{x}) = y]$ if

$$(\boldsymbol{x}' - \boldsymbol{x})\boldsymbol{\Sigma}^{-1}(\boldsymbol{x}' - \boldsymbol{x}) < \Phi^{-1}(q_{y_n})^2, \tag{107}$$

where $q_{y_n}$ is the probability of predicting class $y_n$, i.e. $q_{y_n} = \mathrm{Pr}_{\boldsymbol{z} \sim \mathcal{N}(\boldsymbol{x}, \Sigma)}[g(\boldsymbol{z}) = y_n]$. Here, $g : \mathbb{R}^D \to \mathbb{Y}$ directly outputs a class label instead of a softmax score. We see that both the variance-constrained certificate and ANCER yield the same certified ellipsoid, scaled by a different factor. This factor is the certifiable radius $\eta$, i.e. the r.h.s. term of Eqs. (106) and (107). We also see that both certificates have the same computational complexity – they both involve calculation of the squared Mahalanobis distance and a constant number of operations for evaluation of the certifiable radius.

In the following, we briefly assess under which conditions which certificate yields a larger certifiable radius $\eta$. For this evaluation, we assume that $g(\boldsymbol{x}) = \operatorname{argmax}_y h(\boldsymbol{x})$, i.e. $g$ predicts the class with the highest softmax score. We then vary the prediction probability $q_{y_n}$ and the expected softmax score $\mu$ within $[0.5, 1.0]$. For each $\mu$, we calculate the largest possible variance $\zeta$ (using the Bhatia–Davis inequality $\zeta \le (1 - \mu) \cdot \mu$), which will give us the weakest possible variance-constrained certificate (see Eq. (106)).

Fig. 13 shows the difference in certifiable radius $\eta$, with the dashed line indicating parameters for which both certificates are identical. We have omitted all combinations of $q_{y_n}$ and $\mu$ that are not possible, namely $\mu > q_{y_n} + \frac{1}{2}(1 - q_{y_n})$. We see that ANCER is stronger when $q_{y_n}$ is large, i.e. almost all samples from the smoothing distribution are correctly classified, but not necessarily with high confidence. The variance-constrained certificate is stronger when $q_{y_n}$ is smaller and $\mu$ is larger, i.e. some samples are misclassified but the correctly classified ones have high confidence. Note however, that this is the worst case for the variance-constrained certificate. For $\zeta \to 0$, much larger radii can be certified (see Fig. 14 and Eq. (106)).

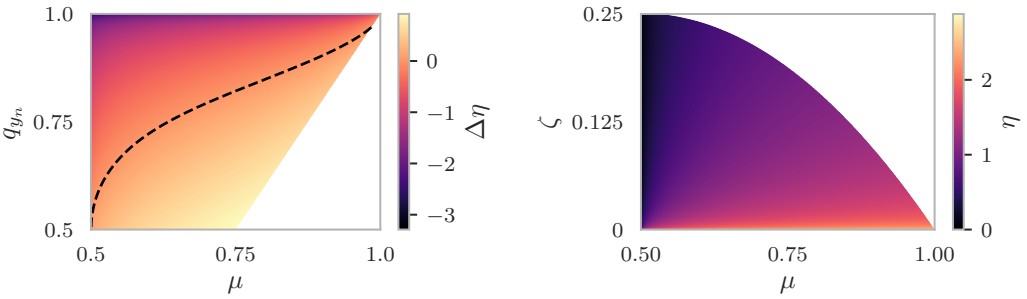

Figure 13: Worst-case difference in certifiable radius $\eta$ between ANCER (Eiras et al., 2022) and the variance-constrained certificate for anisotropic Gaussian smoothing. The dashed line indicates combinations of prediction probability $q_{y_n}$ and expected softmax score $\mu$ for which both certificates are equally strong.

Figure 14: Certifiable radius $\eta$ of the variance-constrained randomized smoothing certificate for anisotropic Gaussian smoothing as a function of the expected value $\mu$ and the variance $\zeta$ of the softmax score. If the variance is small, large radii can be certified – even if the expected softmax score is small.

## G    Monte Carlo Randomized Smoothing

To make predictions and certify robustness, randomized smoothing requires computing certain properties of the distribution of a base model's output, given an input smoothing distribution. For example, the certificate of Cohen et al. (2019) assumes that the smoothed model $f$ predicts the most likely label output by base model $g$, given a smoothing distribution $\mathcal{N}(\mathbf{0}, \sigma \cdot \mathbf{1})$: $f(\boldsymbol{x}) = \text{argmax}_{y \in \mathbb{Y}} \Pr_{\boldsymbol{z} \sim \mathcal{N}(\mathbf{0}, \sigma \cdot \mathbf{1})} [g(\boldsymbol{x} + \boldsymbol{z}) = y]$. To certify the robustness of a smoothed prediction $y = f(\boldsymbol{x})$ for a specific input x, we have to compute the probability $q = \Pr_{\boldsymbol{z} \sim \mathcal{N}(\mathbf{0}, \sigma \cdot \mathbf{1})} [g(\boldsymbol{x} + \boldsymbol{z}) = y]$ to then calculate the maximum certifiable radius $\sigma \Phi^{-1}(q)$ with standard-normal inverse CDF $\Phi^{-1}$. For complicated models like deep neural networks, computing such properties in closed form is usually not tractable. Instead, they have to be estimated using Monte Carlo sampling. The result are predictions and certificates that only hold with a certain probability.

Randomized smoothing with Monte Carlo sampling usually consists of three distinct steps:

1. First, a small number of samples $N_1$ from the smoothing distribution are used to generate a candidate prediction $\hat{y}$, e.g. the most frequently predicted class.

2. Then, a second round of $N_2$ samples is taken and a statistical test is used to determine whether the candidate prediction is likely to be the actual prediction of smoothed classifier $f$, i.e. whether $\hat{y} = f(\boldsymbol{x})$ with a certain probability (1 - $\alpha_1$). If this is not the case, one has to abstain from making a prediction (or generate a new candidate prediction).

3. To certify the robustness of prediction $\hat{y}$, a final round of $N_3$ samples is taken to estimate all quantities needed for the certificate.

In the case of (Cohen et al., 2019), we need to estimate the probability $q = \Pr_{\boldsymbol{z} \sim \mathcal{N}(\mathbf{0}, \sigma \cdot \mathbf{1})} [g(\boldsymbol{x} + \boldsymbol{z}) = \hat{y}]$ to compute the certificate $\sigma \Phi^{-1}(q)$, whose strength is monotonically increasing in $q$. To ensure that the certificate holds with high probability (1 - $\alpha_2$), we have to compute a probabilistic lower bound $\underline{q} \leq q$. Instead of performing two separate round of sampling, one can also re-use the same samples for the abstention test and certification. One particularly simple abstention mechanism is to just compute the Monte Carlo randomized smoothing certificate to determine whether $\forall \boldsymbol{x}' \in \{\boldsymbol{x}\} : f(\boldsymbol{x}') = \hat{y}$ with high probability, i.e. whether the prediction is robust to input $\boldsymbol{x}'$ that is the result of "perturbing" clean input $\boldsymbol{x}$ with zero adversarial budget.

In the following, we discuss how we perform Monte Carlo randomized smoothing for our base certificates, as well as the baselines we use for our experimental evaluation. In § G.4, we discuss how we account for the multiple comparisons problem, i.e. the fact that we are not just trying to probabilistically certify a single prediction, but multiple predictions at once.

### G.1    Monte Carlo Base Certificates for Continuous Data

For our base certificates for continuous data, we follow the approach we already discussed in the previous paragraphs (recall that the certificate of Cohen et al. (2019) is a special case of our certificate with Gaussian noise for $l_2$ perturbations). We are given an input space $\mathbb{X}^{D_{\text{in}}}$, label space $\mathbb{Y}$, base model (or – in the case of multi-output classifiers – base model output) $g : \mathbb{X}^{D_{\text{in}}} \to \mathbb{Y}$ and smoothing distribution $\Psi(\boldsymbol{x})$ (either multivariate Gaussian or multivariate uniform). To generate a candidate prediction, we apply the base classifier to $N_1$ samples from the smoothing distribution in order to obtain predictions $(y^{(1)}, \ldots, y^{(N_1)})$ and compute the majority prediction $\hat{y} = \text{argmax}_{y \in \mathbb{Y}} \{n \mid y^{(n)} = \hat{y}\}$. Recall that for Gaussian and uniform noise, our certificate guarantees $\forall \boldsymbol{x}' \in \mathbb{H} : f(\boldsymbol{x}) = \hat{y}$ for

$$\mathbb{H} = \left\{ \boldsymbol{x}' \in \mathbb{X}^{D_{\text{in}}} \; \middle| \; \sum_{d=1}^{D_{\text{in}}} w_d \cdot |x'_d - x_d|^p < \eta \right\},$$

with $\eta = \left(\Phi^{-1}(q)\right)^2$ or $\eta = \Phi^{-1}(q)$ (depending on the distribution), $q = \Pr_{\boldsymbol{z} \sim \mathcal{N}(\mathbf{0}, \sigma \cdot \mathbf{1})} [g(\boldsymbol{x} + \boldsymbol{z}) = \hat{y}]$ and standard-normal inverse CDF $\Phi^{-1}$. To obtain a probabilistic certificate that holds with high probability $1 - \alpha$, we need a probabilistic lower bound on $\eta$. Both $\eta$ are monotonically increasing in $q$, i.e. we can bound them by finding a lower bound $\underline{q}$ on $q$. For this, we

take $N_2$ more samples from the smoothing distribution and compute a Clopper-Pearson lower confidence bound (Clopper & Pearson, 1934) on $q$. For abstentions, we use the aforementioned simple mechanism: We test whether $\boldsymbol{x} \in \mathbb{H}$. Given the definition of $\mathbb{H}$, this is equivalent to testing whether

$$0 < \Phi^{-1}(\underline{q})$$
$$\iff \Phi(0) < \underline{q}$$
$$\iff 0.5 < \underline{q}.$$

If $\underline{q} \leq 0.5$, we abstain.

## G.2 Monte Carlo Variance-Constrained Certification

For variance-constrained certification, we smooth a model's softmax scores. That is, we are given an input space $\mathbb{X}^{D_{\text{in}}}$, label space $\mathbb{Y}$, base model (or – in the case of multi-output classifiers – base model output) $g : \mathbb{X}^{D_{\text{in}}} \to \Delta_{|\mathbb{Y}|}$ with $(|\mathbb{Y}| - 1)$-dimensional probability simplex $\Delta_{|\mathbb{Y}|}$ and smoothing distribution $\Psi(\boldsymbol{x})$ (Bernoullli or sparsity-aware noise, in the case of binary data). To generate a candidate prediction, we apply the base classifier to $N_1$ samples from the smoothing distribution in order to obtain vectors $\left(\boldsymbol{s}^{(1)}, \ldots, \boldsymbol{s}^{(N_1)}\right)$ with $\boldsymbol{s} \in \Delta_{|\mathbb{Y}|}$, compute the average softmax scores $\overline{\boldsymbol{s}} = \frac{1}{N_1} \sum_{n=1}^{N} \boldsymbol{s}$ and select the label with the highest score $\hat{y} = \arg\max_y \overline{s}_y$.

Recall that our certificate guarantees robustness if the optimal value of the following optimization problem is greater than $0.5$:

$$\min_{h:\mathbb{X}\to\mathbb{R}} \mathbb{E}_{\boldsymbol{z}\sim\Psi(\boldsymbol{x}')} [h(\boldsymbol{z})] \tag{108}$$

$$\text{s.t.} \quad \mathbb{E}_{\boldsymbol{z}\sim\Psi(\boldsymbol{x})} [h(\boldsymbol{z})] \geq \mu, \quad \mathbb{E}_{\boldsymbol{z}\sim\Psi(\boldsymbol{x})} \left[(h(\boldsymbol{z}) - \nu)^2\right] \leq \zeta, \tag{109}$$

with $\mu = \mathbb{E}_{\boldsymbol{z}\sim\Psi(\boldsymbol{x})} [g(\boldsymbol{z})_{\hat{y}}]$, $\zeta = \mathbb{E}_{\boldsymbol{z}\sim\Psi(\boldsymbol{x})} \left[(g(\boldsymbol{z})_{\hat{y}} - \nu)^2\right]$ and a fixed scalar $\nu \in \mathbb{R}$. To obtain a probabilistic certificate, we have to compute a probabilistic lower bound on the optimal value of the optimization problem. Because it is a minimization problem, this can be achieved by loosening its constraints, i.e. computing a probabilistic lower bound $\underline{\mu}$ on $\mu$ and a probabilistic upper bound $\overline{\zeta}$ on $\zeta$.

Like in CDF-smoothing (Kumar et al., 2020), we bound the parameters using CDF-based non-parametric confidence intervals. Let $F(s) = \Pr_{\boldsymbol{z}\sim\Psi(\boldsymbol{x})} [g(\boldsymbol{z})_{\hat{y}} \leq s]$ be the CDF of $g_{\hat{y}}(Z)$ with $Z \sim \Psi(\boldsymbol{x})$. Define $M$ thresholds $\leq 0\tau_1 \leq \tau_2 \ldots, \tau_{M-1} \leq \tau_M \leq 1$ with $\forall m : \tau_m \in [0, 1]$. We then take $N_2$ samples $\boldsymbol{x}^{(1)}, \ldots, \boldsymbol{x}^{(N_2)}$ from the smoothing distribution to compute the empirical CDF $\tilde{F}(s) = \sum_{n=1}^{N_2} \mathrm{I}\left[g(\boldsymbol{z}^{(n)})_{\hat{y}} \leq s\right]$. We can then use the Dvoretzky-Keifer-Wolfowitz inequality (Dvoretzky et al., 1956) to compute an upper bound $\hat{F}$ and a lower bound $\underline{F}$ on the CDF of $g_{\hat{y}}$:

$$\underline{F}(s) = \max\left(\tilde{F}(s) - \upsilon, 0\right) \leq F(s) \leq \min\left(\tilde{F}(s) + \upsilon, 1\right) = \overline{F}(s), \tag{110}$$

with $\upsilon = \sqrt{\frac{\ln 2/\alpha}{2 \cdot N_2}}$, which holds with high probability $(1 - \alpha)$. Using these bounds on the CDF, we can bound $\mu = \mathbb{E}_{\boldsymbol{z}\sim\Psi(\boldsymbol{x})} [g(\boldsymbol{z})_{\hat{y}}]$ as follows (Anderson, 1969):

$$\mu \geq \tau_M - \tau_1 \overline{F}(\tau_1) + \sum_{m=1}^{M-1} (\tau_{m+1} - \tau_m) \overline{F}(\tau_m). \tag{111}$$

The parameter $\zeta = \mathbb{E}_{\boldsymbol{z}\sim\Psi(\boldsymbol{x})} \left[(g(\boldsymbol{z})_{\hat{y}} - \nu)^2\right]$ can be bounded in a similar fashion. Define $\xi_0, \ldots, \xi_M \in \mathbb{R}_+$ with:

$$\xi_0 = \max_{\kappa\in[0,\tau_1]} \left((\kappa - \nu)^2\right)$$
$$\xi_M = \max_{\kappa\in[\tau_M,1]} \left((\kappa - \nu)^2\right) \tag{112}$$
$$\xi_m = \max_{\kappa\in[\tau_m,\tau_{m+1}]} \left((\kappa - \nu)^2\right) \quad \forall m \in \{1, \ldots, M-1\},$$

i.e. compute the maximum squared distance to $\nu$ within each bin $[\tau_m, \tau_{m+1}]$. Then:

$$\zeta \le \xi_0 F(\tau_1) + \xi_M \left(1 - F(\tau_M)\right) + \sum_{m=1}^{M-1} \xi_m \left(F(\tau_{m+1}) - F(\tau_m)\right) \tag{113}$$

$$= \xi_M + \sum_{m=1}^{M-1} \left(\xi_{m-1} - \xi_m\right) F(\tau_m) \tag{114}$$

$$\le \xi_M + \sum_{m=1}^{M-1} \left(\xi_{m-1} - \xi_m\right) \left(\operatorname{sgn}\left(\xi_{m-1} - \xi_m\right) \overline{F}(\tau_m) + \left(1 - \operatorname{sgn}\left(\xi_{m-1} - \xi_m\right)\right) \underline{F}(\tau_m)\right) \tag{115}$$

with probability $(1 - \alpha)$. In the first inequality, we bound the expected squared distance from $\nu$ by assuming that the probability mass in each bin $[\tau_m, \tau_{m+1}]$ is concentrated at the farthest point from $\nu$. The equality is a result of reordering the telescope sum. In the second inequality, we upper-bound the CDF where it is multiplied with a non-negative value and lower-bound it where it is multiplied with a negative value.

With the probabilistic bounds $\underline{\mu}$ and $\overline{\zeta}$ we can now – in principle – evaluate our robustness certificate, i.e. check whether

$$\sum_{z \in \mathbb{X}} \frac{\pi_{x'}(z)^2}{\pi_x(z)} < 1 + \frac{1}{\overline{\zeta} - \left(\underline{\mu} - \nu\right)^2} \left(\underline{\mu} - \frac{1}{2}\right)^2. \tag{116}$$

where the $\pi$ are the probability mass functions of smoothing distributions $\Psi(x)$ and $\Psi(x')$. But one crucial detail of Theorem 5.1 underlying the certificate was that it only holds for $\nu \le \mu$. To use the method with Monte Carlo sampling, one has to ensure that $\nu \le \underline{\mu}$ by first computing $\underline{\mu}$ and then choosing some smaller $\nu$.

In our experiments, we use an alternative method that allows us to use arbitrary $\nu$: From our proof of Theorem 5.1 we know that the dual problem of Eq. 108 is

$$\max_{\alpha, \beta \ge 0} \alpha \underline{\mu} - \beta \overline{\zeta} - \frac{\alpha^2}{4\beta} + \frac{\alpha}{2\beta} - \alpha \nu + \nu - \frac{1}{4\beta} \sum_{z \in \mathbb{X}} \frac{\pi_{x'}(z)^2}{\pi_x(z)}, \tag{117}$$

Instead of trying to find an optimal $\alpha$ (which causes problems in subsequent derivations if $\nu \not\le \underline{\mu}$), we can simply choose $\alpha = 1$. By duality, the result is still a lower bound on the primal problem, i.e. the certificate remains valid. The dual problem becomes

$$\max_{\beta \ge 0} \underline{\mu} - \beta \overline{\zeta} + \frac{1}{4\beta} - \frac{1}{4\beta} \sum_{z \in \mathbb{X}} \frac{\pi_{x'}(z)^2}{\pi_x(z)}. \tag{118}$$

The problem is concave in $\beta$ (because the expected likelihood ratio is $\ge 1$). Finding the optimal $\beta$, comparing the result to $0.5$ and solving for the expected likelihood ratio, shows that a prediction is robust if

$$\sum_{z \in \mathbb{X}} \frac{\pi_{x'}(z)^2}{\pi_x(z)} < 1 + \frac{1}{\overline{\zeta}} \left(\underline{\mu} - \frac{1}{2}\right)^2. \tag{119}$$

For our abstention mechanism, like in the previous section, we compute the certificate $\mathbb{H}$ and then test whether $x \in \mathbb{H}$. In the case of Bernoulli smoothing and sparsity-aware smoothing), this corresponds to testing whether

$$1 < \ln\left(1 + \frac{1}{\overline{\zeta}} \left(\underline{\mu} - \frac{1}{2}\right)\right) \tag{120}$$

$$\iff \underline{\mu} > \frac{1}{2}. \tag{121}$$

## G.3 MONTE CARLO CENTER SMOOTHING

While we can not use center smoothing as a base certificate, we benchmark our method against it during our experimental evaluation. The generation of candidate predictions, the abstention mechanism and the certificate are explained in (Kumar & Goldstein, 2021). The authors allow multiple options for generating candidate predictions. We use the "$\beta$ minimum enclosing ball" with $\beta = 2$ that is based on pair-wise distance calculations.

## G.4    MULTIPLE COMPARISONS PROBLEM

The first step of our collective certificate is to compute one base certificate for each of the $D_{\text{out}}$ predictions of the multi-output classifier. With Monte Carlo randomized smoothing, we want all of these probabilistic certificates to simultaneously hold with a high probability $(1 - \alpha)$. But as the number of certificates increases, so does the probability of at least one of them being invalid. To account for this *multiple comparisons problem*, we use Bonferroni (Bonferroni, 1936) correction, i.e. compute each Monte Carlo certificate such that it holds with probability $(1 - \frac{\alpha}{n})$.

For base certificates that only depend on $q_n = \Pr_{\boldsymbol{z} \sim \Psi^{(n)}} [g_n(\boldsymbol{z}) = \hat{y}_n]$, i.e. the probability of the base classifier predicting a particular label $\hat{y}_n$ under the smoothing distribution, one can also use the strictly better Holm correction (Holm, 1979). This includes our Gaussian and uniform smoothing certificates for continuous data. Holm correction is a procedure than can be used to correct for the multiple comparisons problem when performing multiple arbitrary hypothesis tests. Given $N$ hypotheses, their p-values are ordered in ascending order $p_1, \ldots, p_N$. Starting at $i = 1$, the $i$'th hypothesis is rejected if $p_i < \frac{\alpha}{N+1-i}$, until one reaches an $i$ such that $p_i \geq \frac{\alpha}{N+1-i}$.

Fischer et al. (2021) proposed to use Holm correction as part of their procedure for certifying that all (non-abstaining) predictions of an image segmentation model are robust to adversarial perturbations. In the following, we first summarize their approach and then discuss how Holm correction can be used for certifying our notion of collective robustness, i.e. certifying the number of robust predictions. As in § G.1, the goal is to obtain a lower bound $\underline{q}_n$ on $q_n = \Pr_{\boldsymbol{z} \sim \Psi^{(n)}} [g_n(\boldsymbol{z}) = \hat{y}_n]$ for each of the $D_{\text{out}}$ classifier outputs. Assume we take $N_2$ samples $\boldsymbol{z}^{(1)}, \ldots, \boldsymbol{z}^{(N_2)}$ from the smoothing distribution. Let $\nu_n = \sum_{i=1}^{N_2} \mathrm{I} \left[ g_n(\boldsymbol{z}^{(i)}) = \hat{y}_n \right]$ and let $\pi : \{1, \ldots, D_{\text{out}}\} \rightarrow \{1, \ldots, D_{\text{out}}\}$ be a bijection that orders the $\nu_n$ in descending order, i.e. $\nu_{\pi(1)} \geq \nu_{\pi(2)} \cdots \geq \nu_{\pi(D_{\text{out}})}$. Instead of using Clopper-Pearson confidence intervals to obtain tight lower bounds on the $q_n$, Fischer et al. (2021) define a threshold $\tau \in [0.5, 1)$ and use Binomial tests to determine for which $n$ the bound $\tau \leq q_n$ holds with high-probability. Let $\mathrm{BinP}(\nu_n, N_2, \leq, \tau)$ be the p-value of the one-sided binomial test, which is monotonically decreasing in $\nu_n$. Following the Holm correction scheme, the authors test whether

$$\mathrm{BinP}\left(\nu_{\pi(k)}, N_2, \leq, \tau\right) < \frac{\alpha}{D_{\text{out}} + 1 - k} \tag{122}$$

for $k = 1, \ldots, D_{\text{out}}$ until reaching a $k^*$ for which the null-hypothesis can no longer be rejected, i.e. the p-value is g.e.q. $\frac{\alpha}{D_{\text{out}} + 1 - k^*}$. They then know that with probability $1 - \alpha$, the bound $\tau \leq q_n$ holds for all $n \in \{\pi(k) \mid k \in \{1, \ldots, k^*\}\}$. For these outputs, they use the lower bound $\tau$ to compute robustness certificates. They abstain with all other outputs.

This approach is sensible when one is concerned with the least robust prediction from a set of predictions. But our collective certificate benefits from having tight robustness guarantees for each of the individual predictions. Holm correction can be used with arbitrary hypothesis tests. For instance, we can use a different threshold $\tau_n$ per output $g_n$, i.e. test whether

$$\mathrm{BinP}\left(\nu_{\pi(k)}, N_2, \leq, \tau_{\pi(k)}\right) < \frac{\alpha}{D_{\text{out}} + 1 - k} \tag{123}$$

for $k = 1, \ldots, D_{\text{out}}$. In particular, we can use

$$\tau_n = \sup_t \text{ s.t. } \mathrm{BinP}\left(\nu_n, N_2, \leq, t\right) < \frac{\alpha}{D_{\text{out}} + 1 - \pi^{-1}(n)}, \tag{124}$$

i.e. choose the largest threshold such that the null hypothesis can still be rejected. Eq. 124 is the lower Clopper-Pearson confidence bound with significance $\frac{\alpha}{D_{\text{out}} + 1 - \pi^{-1}(n)}$. This means that, instead of performing hypothesis tests, we can obtain probabilistic lower bounds $\underline{q}_n \leq q_n$ by computing Clopper-Pearson confidence bounds with significance parameters $\frac{\alpha}{D_{\text{out}}}, \ldots, \frac{\alpha}{1}$. The $\underline{q}_n$ can then be used to compute the base certificates. Due to the definition of the $\tau_n$, all of the null hypotheses are rejected, i.e. we obtain valid probabilistic lower bounds on all $q_n$. We can thus use the abstention mechanism from § G.1, i.e. only abstain if $\underline{q}_n \leq 0.5$.

## H Comparison to the Collective Certificate of Fischer et al. (2021)

Our collective certificate based on localized smoothing is designed to bound the number of simultaneously robust predictions. Fischer et al. (2021) designed SegCertify to determine whether all predictions are simultaneously robust. As discussed in § 3.2, their work is based on the naïve collective certification approach applied to isotropic Gaussian smoothing: They first certify each output independently, then count the number of certifiably robust predictions for a specific adversarial budget and then test whether the number of certifiably robust predictions equals the overall number of predictions. To obtain better guarantees in practical scenarios, they further propose to

- use Holm correction to address the multiple comparisons problem (see § G.4),
- Abstain at a higher rate to avoid "bad componets", i.e. predictions $y_n$ that have a low consistency $q_n = \Pr_{\boldsymbol{z} \sim \mathcal{N}(\boldsymbol{x}, \sigma)}[g(\boldsymbol{z}) = y]$ and thus very small certifiable radii.

A more technical summary of their method can be found in § G.4.

In the following, we discuss why our certificate can always offer guarantees that are at least as strong as SegCertify, both for our notion of collective robustness (number of robust predictions) and their notion of collective robustness (robustness of all predictions). In short, isotropic smoothing is a special case of localized smoothing and Holm correction can also be used for our base certificates. Before proceedings, please read the discussion on Monte Carlo base certificates and Clopper-Pearson confidence intervals in § G.1 and the multiple comparisons problem in § G.4.

A direct consequence of the results in § G.4 is that using Clopper-Pearson confidence intervals and Holm correction will yield stronger per-prediction robustness guarantees and lower abstention rates than the method of Fischer et al. (2021). The Clopper-Pearson-based method only abstains if one cannot guarantee that $q_n > 0.5$ with high probability, while their method abstains if one cannot guarantee that $q_n \geq \tau$ with $\tau \geq 0.5$ (or specific other predictions abstain). For all non-abstaining predictions, the Clopper-Pearson-based certificate will be at least as strong as the one obtained using a single threshold $\tau$, as it computes the tightest bound for which the null hypothesis can still be rejected (see Eq. 124).

Consequently, when certifying our notion of collective robustness, i.e. determining *the number* of robust predictions given adversarial budget $\epsilon$, a naïve collective robustness certificate (i.e. counting the number of predictions whose robustness are guaranteed by the base certificates) based on Clopper-Pearson bounds will also be stronger than the method of Fischer et al. (2021). It should however be noted that their method could potentially be used with other methods of family-wise error rate correction, although they state that "these methods do not scale to realistic segmentation problems" and do not discuss any further details.

Conversely, when certifying their notion of collective robustness, i.e. determining whether *all* non-abstaining predictions are robust given adversarial budget $\epsilon$, the certificate based on Clopper-Pearson confidence bounds is also at least as strong as that of Fischer et al. (2021). To certify their notion of robustness, they iterate over all predictions and determine whether all non-abstaining predictions are certifiably robust, given $\epsilon$. Naturally, as the Clopper-Pearson-based certificates are stronger, any prediction that is robust according to (Fischer et al., 2021) is also robust acccording to the Clopper-Pearson-based certificates. The only difference is that, for $\tau > 0.5$, their method will have more abstaining predictions. But, due to the direct correspondence of Clopper-Pearson confidence bounds and Binomial tests, we can modify our abstention mechanism to obtain exactly the same set of abstaining predictions: We simply have to use $\underline{q}_n \leq \tau$ instead of $\underline{q_n} \leq 0.5$ as our criterion.

Finally, it should be noted that our proposed collective certificate based on linear programming is at least as strong as the naïve collective certificate (see Eq. 1.1 and Eq. 1.2 in § 3.2). Thus, letting the set of targeted predictions $\mathbb{T}$ be the set of all non-abstaining predictions and checking whether the collective certificate guarantees robustness for all of $\mathbb{T}$ will also result in a certificate that is at least as strong as that of Fischer et al. (2021) in their setting.

# I   COMPARISON TO THE COLLECTIVE CERTIFICATE OF SCHUCHARDT ET AL. (2021)

In the following, we first present the collective certificate for binary graph-structured data proposed by Schuchardt et al. (2021) (see § I.1. We then show that, when using sparsity-aware smoothing distributions (Bojchevski et al., 2020) – the family of smoothing distributions used both in our work and that of Schuchardt et al. (2021) – our certificate subsumes their certificate. That is, our collective robustness certificate based on localized randomized smoothing can provide the same robustness guarantees (see § I.2).

## I.1   THE COLLECTIVE CERTIFICATE

Their certificate assumes the input space to be $\mathbb{G} = \{0, 1\}^{N \times D} \times \{0, 1\}^{N \times N}$ – the set of undirected attributed graphs with $N$ nodes and $D$ attributes per node. The model is assumed to be a multi-output classifier $f : \mathbb{G} \to \mathbb{Y}^N$ that assigns a label from label set $\mathbb{Y}$ to each of the nodes. Given an input graph $\mathcal{G} = (\boldsymbol{X}, \boldsymbol{A})$ and a corresponding prediction $\boldsymbol{y} = f(G)$, they want to certify collective robustness to a set of perturbed graphs $\mathbb{B} \subseteq \mathbb{G}$. The perturbation model $\mathbb{B}$ is characterized by four scalar parameters $r_{\boldsymbol{X}}^+, r_{\boldsymbol{X}}^-, r_{\boldsymbol{A}}^+, r_{\boldsymbol{A}}^+ \in \mathbb{N}_0$, specifying the number of bits the adversary is allowed to add ($0 \to 1$) and delete ($1 \to 0$) in the attribute and adjacency matrix, respectively. It can also be extended to feature additional constraints (e.g. per-node budgets). We discuss how these can be integrated after showing our main result. A formal definition of the perturbation model can be found in Section B of (Schuchardt et al., 2021).

The goal of their work is to certify collective robustness for a set of targeted nodes $\mathbb{T} \subseteq \{1, \dots, N\}$, i.e. compute a lower bound on

$$\min_{G' \in \mathbb{B}} \sum_{n \in \mathbb{T}} \mathrm{I} \left[ f_n(G') = y_n \right]. \tag{125}$$

Their approach to obtaining this lower-bound shares the same high-level idea as ours (see § 3.2): Combining per-prediction base certificates and leveraging some notion of locality. But while our method uses localized randomized smoothing, i.e. smoothing different outputs with different non-i.i.d. smoothing distributions to obtain base certificates that encode locality, their method uses a-priori knowledge about the strict locality of the classifier $f$. A model is strictly local if each of its outputs $f_n$ only operates on a well-defined subset of the input data. To encode this strict locality, Schuchardt et al. (2021) associate each output $f_n$ with an indicator vector $\boldsymbol{\psi}^{(n)}$ and an indicator matrix $\boldsymbol{\Psi}^{(n)}$ that fulfill

$$\sum_{m=1}^{N} \sum_{d=1}^{D} \psi_m^{(n)} \mathrm{I} \left[ X_{m,d} \neq X'_{i,j} \right] + \sum_{i=1}^{N} \sum_{j=1}^{N} \Psi_m^{(n)} \mathrm{I} \left[ A_{m,d} \neq A'_{i,j} \right] = 0$$
$$\implies f_n(\boldsymbol{X}, \boldsymbol{A}) = f_n(\boldsymbol{X}', \boldsymbol{A}'). \tag{126}$$

for any perturbed graph $\mathcal{G}' = (\boldsymbol{X}', \boldsymbol{A}')$. Eq. 126 expresses that the prediction of output $f_n$ remains unchanged if all inputs in its receptive field remain unchanged. Conversely, it expresses that perturbations outside the receptive field can be ignored. Unlike in our work, Schuchardt et al. (2021) describe their base certificates as sets in adversarial budget space. That is, some certification procedure is applied to each output $f_n$ to obtain a set

$$\mathbb{K}^{(n)} \subseteq [r_{\boldsymbol{X}}^+] \times [r_{\boldsymbol{X}}^-] \times [r_{\boldsymbol{A}}^+] \times [r_{\boldsymbol{X}}^-] \tag{127}$$

with $[k] = \{0, \dots, k\}$. If $\begin{bmatrix} c_{\boldsymbol{X}}^+ & c_{\boldsymbol{X}}^- & c_{\boldsymbol{A}}^+ & c_{\boldsymbol{A}}^- \end{bmatrix}^T \in \mathbb{K}^{(n)}$, then prediction $y_n$ is robust to any perturbed input with exactly $c_{\boldsymbol{X}}^+$ attribute additions, $c_{\boldsymbol{X}}^-$ attribute deletions, $c_{\boldsymbol{A}}^+$ edge additions and $c_{\boldsymbol{A}}^-$ edge deletions. A more detailed explanation can be found in Section 3 of (Schuchardt et al., 2021). Note that the base certificates only depend on the number of perturbations, not their location in the input. Only by combining them using the receptive field indicators from Eq. 126 can one obtain a collective certificate that is better than the naïve collective certificate (i.e. counting how many predictions are certifiably robust to the collective threat model). The resulting collective certificate

is

$$\min_{\boldsymbol{b}^+,\boldsymbol{b}^+,\boldsymbol{B}^+,\boldsymbol{B}^-} \sum_{n \in \mathbb{T}} \mathrm{I}\left[\left[\left(\boldsymbol{\psi}^{(n)}\right)^T \boldsymbol{b}_{\boldsymbol{X}}^+ \quad \left(\boldsymbol{\psi}^{(n)}\right)^T \boldsymbol{b}_{\boldsymbol{X}}^- \quad \sum_{i,j} \Psi_{i,j}^{(n)} \boldsymbol{B}_{i,j}^+ \quad \sum_{i,j} \Psi_{i,j}^{(n)} \boldsymbol{B}_{i,j}^-\right]^T \in \mathbb{K}^{(n)}\right]$$
(128)

$$\text{s.t.} \quad \sum_{m=1}^{N} b_m^+ \le r_{\boldsymbol{X}}^+, \quad \sum_{m=1}^{N} b_m^- \le r_{\boldsymbol{X}}^-, \quad \sum_{i=1}^{N}\sum_{j=1}^{N} B_{i,j}^+ \le r_{\boldsymbol{A}}^+, \quad \sum_{i=1}^{N}\sum_{j=1}^{N} B_{i,j}^- \le r_{\boldsymbol{A}}^-, \quad (129)$$

$$\boldsymbol{b}^+, \boldsymbol{b}^- \in \mathbb{N}_0^N \quad \boldsymbol{B}^+, \boldsymbol{B}^- \in \mathbb{N}_0^{N \times N}. \tag{130}$$

The variables defined in Eq. 130 model how the adversary allocates their adversarial budget, i.e. how many attributes are perturbed per node and which edges are modified. Eq. 129 ensures that this allocation in compliant with the collective threat model. Finally, in Eq. 128 the indicator vector and matrix $\boldsymbol{\psi}^{(n)}$ and $\boldsymbol{\Psi}^{(n)}$ are used to mask out any allocated perturbation budget that falls outside the receptive field of $f_n$ before evaluating its base certificate.

To solve the optimization problem, Schuchardt et al. (2021) replace each of the indicator functions with binary variables and include additional constraints to ensure that they have value 1 i.f.f. the indicator function would have value 1. To do so, they define one linear constraint per point separating the set of certifiable budgets $\mathbb{K}^{(n)}$ from its complement $\overline{\mathbb{K}}^{(n)}$ in adversarial budget space (the "Pareto front" discussed in Section 3 of (Schuchardt et al., 2021)).

From the above explanation, the main drawbacks of this collective certificate compared to our localized randomized smoothing approach and corresponding collective certificate should be clear. Firstly, if the classifier $f$ is not strictly local, i.e. the receptive field indicators $\psi$ and $\boldsymbol{\Psi}$ only have non-zero entries, then all base certificates are evaluated using the entire collective adversarial budget. It thus degenerates to the naïve collective certificate. Secondly, even if the model is strictly local, each of the outputs may assign varying levels of importance to different parts of its receptive field. Their method is incapable of capturing this additional soft locality. Finally, their means of evaluating the base certificates may involve evaluating a large number of linear constraints. Our method, on the other hand, only requires a single constraint per prediction. Our collective certificate can thus be more efficiently computed.

### I.2 Proof of Subsumption

In the following, we show that any robustness certificate obtained by using the collective certificate of Schuchardt et al. (2021) with sparsity-aware randomized smoothing base certificates can also be obtained by using our proposed collective certificate with an appropriately parameterized localized smoothing distribution. The fundamental idea is that, for randomly smoothed models, completely randomizing all input dimensions outside the receptive field is equivalent to masking out any perturbations outside the receptive field.

First, we derive the certificate of Schuchardt et al. (2021) for predictions obtained via sparsity-aware smoothing. Schuchardt et al. (2021) require base certificates that guarantee robustness when $\begin{bmatrix} c_{\boldsymbol{X}}^+ & c_{\boldsymbol{X}}^- & c_{\boldsymbol{A}}^+ & c_{\boldsymbol{A}}^- \end{bmatrix}^T \in \mathbb{K}^{(n)}$, where the $c$ indicate the number of added and deleted attribute and adjacency bits. That is, the certificates must only depend on the number of perturbations, not on their location. To achieve this, all entries of the attribute matrix and all entries of the adjacency matrix, respectively, must share the same distribution. For the attribute matrix, they define scalar distribution parameters $p_{\boldsymbol{X}}^+, p_{\boldsymbol{A}}^- \in [0,1]$. Given attribute matrix $\boldsymbol{X} \in \{0,1\}^{N \times D}$, they then sample random attribute matrices $\boldsymbol{Z}_{\boldsymbol{X}}$ that are distributed according to sparsity-aware smoothing distribution $\mathcal{S}\left(\boldsymbol{X}, \mathbf{1} \cdot p_{\boldsymbol{X}}^+, \mathbf{1} \cdot p_{\boldsymbol{X}}^-\right)$ (see § F.3.2), i.e.

$$\Pr[(Z_{\boldsymbol{X}})_{m,d} = 0] = \left(1 - p_{\boldsymbol{X}}^+\right)^{1-X_{m,d}} \cdot \left(p_{\boldsymbol{X}}^-\right)^{X_{m,d}},$$
$$\Pr[(Z_{\boldsymbol{X}})_{m,d} = 1] = \left(p_{\boldsymbol{X}}^+\right)^{1-X_{m,d}} \cdot \left(1 - p_{\boldsymbol{X}}^-\right)^{X_{m,d}}.$$

Given input adjacency matrix $\boldsymbol{A}$, random adjacency matrices $\boldsymbol{Z}_{\boldsymbol{A}}$ are sampled from the distribution $\mathcal{S}\left(\boldsymbol{A}, \mathbf{1} \cdot p_{\boldsymbol{A}}^+, \mathbf{1} \cdot p_{\boldsymbol{A}}^-\right)$. Applying Corollary F.6 (to the flattened and concatenated attribute and adjacency matrices) shows that smoothed prediction $y_n = f_n(\boldsymbol{X}, \boldsymbol{A})$ is robust to the perturbed graph

$(\boldsymbol{X}', \boldsymbol{A}')$ if

$$
\sum_{m=1}^{N}\sum_{d=1}^{D}\gamma_{\boldsymbol{X}}^{+}\cdot\mathrm{I}\left[X_{m,d}=0\neq X'_{m,d}\right]+\gamma_{\boldsymbol{X}}^{-}\cdot\mathrm{I}\left[X_{m,d}=1\neq X'_{m,d}\right]
$$
$$
+\sum_{i=1}^{N}\sum_{i=1}^{N}\gamma_{\boldsymbol{A}}^{+}\cdot\mathrm{I}\left[A_{i,j}=0\neq A'_{i,j}\right]+\gamma_{\boldsymbol{A}}^{-}\cdot\mathrm{I}\left[A_{i,j}=1\neq A'_{i,j}\right] \tag{131}
$$
$$
<\eta^{(n)}
$$

with $\quad \gamma_{\boldsymbol{X}}^{+} \quad = \quad \ln\left(\frac{\left(p_{\boldsymbol{X}}^{-}\right)^2}{1-p_{\boldsymbol{X}}^{+}}+\frac{\left(1-p_{\boldsymbol{X}}^{-}\right)^2}{p_{\boldsymbol{X}}^{+}}\right), \quad \gamma_{\boldsymbol{X}}^{-} \quad = \quad \ln\left(\frac{\left(1-p_{\boldsymbol{X}}^{+}\right)^2}{p_{\boldsymbol{X}}^{-}}+\frac{\left(p_{\boldsymbol{X}}^{+}\right)^2}{1-p_{\boldsymbol{X}}^{-}}\cdot\right), \quad \gamma_{\boldsymbol{A}}^{+} \quad =$
$\ln\left(\frac{\left(p_{\boldsymbol{A}}^{-}\right)^2}{1-p_{\boldsymbol{A}}^{+}}+\frac{\left(1-p_{\boldsymbol{A}}^{-}\right)^2}{p_{\boldsymbol{A}}^{+}}\right), \gamma_{\boldsymbol{A}}^{-}=\ln\left(\frac{\left(1-p_{\boldsymbol{A}}^{+}\right)^2}{p_{\boldsymbol{A}}^{-}}+\frac{\left(p_{\boldsymbol{A}}^{+}\right)^2}{1-p_{\boldsymbol{A}}^{-}}\cdot\right)$ and $\eta^{(n)}=\ln\left(1+\frac{1}{\sigma^{(n)2}}\left(\mu^{(n)}-\frac{1}{2}\right)^2\right)$,
where $\mu^{(n)}$ is the mean and $\sigma^{(n)}$ is the variance of the base classifier's output distribution, given the input smoothing distribution. Since the indicator functions for each perturbation type in Eq. 131 share the same weights, Eq. 131 can be rewritten as

$$
\gamma_{\boldsymbol{X}}^{+}c_{\boldsymbol{X}}^{+}+\gamma_{\boldsymbol{X}}^{-}c_{\boldsymbol{X}}^{-}+\gamma_{\boldsymbol{A}}^{+}c_{\boldsymbol{A}}^{+}+\gamma_{\boldsymbol{A}}^{-}c_{\boldsymbol{A}}^{-}\leq\eta^{(n)}, \tag{132}
$$

where $c_{\boldsymbol{X}}^{+}, c_{\boldsymbol{X}}^{-}, c_{\boldsymbol{A}}^{+}, c_{\boldsymbol{A}}^{-}$ are the overall number of added and deleted attribute and adjacency bits, respectively. Eq. 132 matches the notion of base certificates defined by Schuchardt et al. (2021), i.e. it corresponds to a set $\mathbb{K}^{(n)}$ in adversarial budget space for which we provably know that prediction $y_n$ is certifiably robust if $\begin{bmatrix} c_{\boldsymbol{X}}^{+} & c_{\boldsymbol{X}}^{-} & c_{\boldsymbol{A}}^{+} & c_{\boldsymbol{A}}^{-} \end{bmatrix}^T \in \mathbb{K}^{(n)}$. When we insert the base certificate Eq. 132 into objective function Eq. 128, the collective certificate of Schuchardt et al. (2021) becomes equivalent to

$$
\min_{\boldsymbol{b}^{+},\boldsymbol{b}^{+},\boldsymbol{B}^{+},\boldsymbol{B}^{-}}\sum_{n\in\mathbb{T}}\mathrm{I}\left[\gamma_{\boldsymbol{X}}^{+}\left(\boldsymbol{\psi}^{(n)}\right)^{T}\boldsymbol{b}_{\boldsymbol{X}}^{+}+\gamma_{\boldsymbol{X}}^{-}\left(\boldsymbol{\psi}^{(n)}\right)^{T}\boldsymbol{b}_{\boldsymbol{X}}^{-}\right.
$$
$$
\left.+\gamma_{\boldsymbol{A}}^{+}\sum_{i,j}\Psi_{i,j}^{(n)}\boldsymbol{B}_{i,j}^{+}+\sum_{i,j}\gamma_{\boldsymbol{A}}^{-}\Psi_{i,j}^{(n)}\boldsymbol{B}_{i,j}^{-}\leq\eta^{(n)}\right] \tag{133}
$$

$$
\text{s.t.}\quad \sum_{m=1}^{N}b_m^{+}\leq r_{\boldsymbol{X}}^{+},\quad \sum_{m=1}^{N}b_m^{-}\leq r_{\boldsymbol{X}}^{-},\quad \sum_{i=1}^{N}\sum_{j=1}^{N}B_{i,j}^{+}\leq r_{\boldsymbol{A}}^{+},\quad \sum_{i=1}^{N}\sum_{j=1}^{N}B_{i,j}^{-}\leq r_{\boldsymbol{A}}^{-}, \tag{134}
$$

$$
\boldsymbol{b}^{+},\boldsymbol{b}^{-}\in\mathbb{N}_0^{N}\quad \boldsymbol{B}^{+},\boldsymbol{B}^{-}\in\mathbb{N}_0^{N\times N}. \tag{135}
$$

Next, we show that obtaining base certificates through localized randomized smoothing with appropriately chosen parameters and using these base certificates within our proposed collective certificate (see Theorem 4.2) will result in the same optimization problem. Instead of using the same smoothing distribution for all outputs, we use different distribution parameters for each one. For the $n$'th output, we sample random attributes matrices from distribution $\mathcal{S}\left(\boldsymbol{X},\boldsymbol{\Theta}_{\boldsymbol{X}}^{+}{}^{(n)},\boldsymbol{\Theta}_{\boldsymbol{X}}^{-}{}^{(n)}\right)$ with $\boldsymbol{\Theta}_{\boldsymbol{X}}^{+}{}^{(n)},\boldsymbol{\Theta}_{\boldsymbol{X}}^{-}{}^{(n)}\in[0,1]^{N\times D}$. Note that, in order to avoid having to index flattened vectors, we overload the definition of sparsity-aware smoothing to allow for matrix-valued parameters. For example, the value $\boldsymbol{\Theta}_{\boldsymbol{X}}^{+}{}_{n,d}^{(n)}$ indicates the probability of flipping the value of input attribute $X_{n,d}$ from 0 to 1 and the value $\boldsymbol{\Theta}_{\boldsymbol{X}}^{-}{}_{n,d}^{(n)}$ indicates the probability of flipping the value of input attribute $X_{n,d}$ from 1 to 0. We choose the following values for these parameters:

$$
\boldsymbol{\Theta}_{\boldsymbol{X}}^{+}{}_{m,d}^{(n)}=\psi_m^{(n)}\cdot p_{\boldsymbol{X}}^{+}+\left(1-\psi_m^{(n)}\right)\cdot 0.5, \tag{136}
$$

$$
\boldsymbol{\Theta}_{\boldsymbol{X}}^{-}{}_{m,d}^{(n)}=\psi_m^{(n)}\cdot p_{\boldsymbol{X}}^{-}+\left(1-\psi_m^{(n)}\right)\cdot 0.5, \tag{137}
$$

where $\boldsymbol{\psi}^{(n)}$ is the receptive field indicator vector defined in Eq. 126 and $p_{\boldsymbol{X}}^{+},\cdot p_{\boldsymbol{X}}^{-}\in[0,1]$ are the same flip probabilities we used for the certificate of Schuchardt et al. (2021). Due to this parameterization, attribute bits inside the receptive field are randomized using the same distribution as in the certificate of Schuchardt et al. (2021), while attribute bits outside are set to either

0 or 1 with equal probability. Similarly, we sample random adjacency matrices from distribution $\mathcal{S}\left(\boldsymbol{A}, \boldsymbol{\Theta}_{\boldsymbol{A}}^{+\,(n)}, \boldsymbol{\Theta}_{\boldsymbol{A}}^{-\,(n)}\right)$ with $\boldsymbol{\Theta}_{\boldsymbol{A}}^{+\,(n)}, \boldsymbol{\Theta}_{\boldsymbol{A}}^{-\,(n)} \in [0,1]^{N \times D}$ and

$$\Theta_{\boldsymbol{A}\,i,j}^{+\,(n)} = \Psi_{i,j}^{(n)} \cdot p_{\boldsymbol{A}}^{+} + \left(1 - \Psi_{i,j}^{(n)}\right) \cdot 0.5, \tag{138}$$

$$\Theta_{\boldsymbol{A}\,u,j}^{-\,(n)} = \Psi_{i,j}^{(n)} \cdot p_{\boldsymbol{A}}^{-} + \left(1 - \Psi_{i,j}^{(n)}\right) \cdot 0.5, \tag{139}$$

where $\boldsymbol{\Psi}^{(n)}$ is the receptive field indicator matrix defined in Eq. 126. Note that, since we only alter the distribution of bits outside the receptive field, the smoothed prediction $y_n = f_n(\boldsymbol{X}, \boldsymbol{A})$ will be the same as the one obtained via the smoothing distribution used by Schuchardt et al. (2021). Applying Corollary F.6 (to the flattened and concatenated attribute and adjacency matrices) shows that smoothed prediction $y_n = f_n(\boldsymbol{X}, \boldsymbol{A})$ is robust to the perturbed graph $(\boldsymbol{X}', \boldsymbol{A}')$ if

$$\sum_{m=1}^{N} \sum_{d=1}^{D} \tau_{\boldsymbol{X}\,m,d}^{+} \cdot \mathrm{I}\left[X_{m,d} = 0 \neq X_{m,d}'\right] + \tau_{\boldsymbol{X}\,m,d}^{-} \cdot \mathrm{I}\left[X_{m,d} = 1 \neq X_{m,d}'\right]$$
$$+ \sum_{i=1}^{N} \sum_{j=1}^{N} \tau_{\boldsymbol{A}\,i,j}^{+} \cdot \mathrm{I}\left[A_{i,j} = 0 \neq A_{i,j}'\right] + \tau_{\boldsymbol{A}\,i,j}^{-} \cdot \mathrm{I}\left[A_{i,j} = 1 \neq A_{i,j}'\right] \tag{140}$$
$$< \eta^{(n)}.$$

Because we only changed the distribution outside the receptive field, the scalar $\eta^{(n)}$, which depends on the output distribution's mean and variance $\mu$ and $\sigma$ will be the same as the one obtained via the smoothing scheme used by Schuchardt et al. (2021) et al. Due to Corollary F.6 and the definition of our smoothing distribution parameters in Eqs. (136) to (139), the scalars $\tau_{\boldsymbol{X}\,m,d}^{+}, \tau_{\boldsymbol{X}\,m,d}^{-}, \tau_{\boldsymbol{A}\,i,j}^{+}, \tau_{\boldsymbol{A}\,i,j}^{-}$ have the following values:

$$\tau_{\boldsymbol{X}\,m,d}^{+} = \psi_m^{(n)} \cdot \gamma_{\boldsymbol{X}}^{+} + \left(1 - \psi_m^{(n)}\right) \cdot 2 \cdot \ln\left(\frac{(1-0.5)^2}{0.5} + \frac{0.5^2}{1-0.5}\right) \tag{141}$$

$$\tau_{\boldsymbol{X}\,m,d}^{-} = \psi_m^{(n)} \cdot \gamma_{\boldsymbol{X}}^{-} + \left(1 - \psi_m^{(n)}\right) \cdot 2 \cdot \ln\left(\frac{(1-0.5)^2}{0.5} + \frac{0.5^2}{1-0.5}\right) \tag{142}$$

$$\tau_{\boldsymbol{A}\,i,j}^{-} = \Psi_{i,j}^{(n)} \cdot \gamma_{\boldsymbol{A}}^{+} + \left(1 - \Psi_{i,j}^{(n)}\right) \cdot 2 \cdot \ln\left(\frac{(1-0.5)^2}{0.5} + \frac{0.5^2}{1-0.5}\right) \tag{143}$$

$$\tau_{\boldsymbol{A}\,i,j}^{-} = \Psi_{i,j}^{(n)} \cdot \gamma_{\boldsymbol{A}}^{-} + \left(1 - \Psi_{i,j}^{(n)}\right) \cdot 2 \cdot \ln\left(\frac{(1-0.5)^2}{0.5} + \frac{0.5^2}{1-0.5}\right), \tag{144}$$

where the $\gamma$ are the same weights as those of the base certificate Eq. 131 of Schuchardt et al. (2021). Inserting the above values of $\tau$ into the base certificate Eq. 140 and using the fact that $\ln\left(\frac{(1-0.5)^2}{0.5} + \frac{0.5^2}{1-0.5}\right) = \ln(1) = 0$ results in

$$\sum_{m=1}^{N} \sum_{d=1}^{D} \psi_m^{(n)} \cdot \gamma_{\boldsymbol{X}}^{+} \cdot \mathrm{I}\left[X_{m,d} = 0 \neq X_{m,d}'\right] + \psi_m^{(n)} \cdot \gamma_{\boldsymbol{X}}^{-} \cdot \mathrm{I}\left[X_{m,d} = 1 \neq X_{m,d}'\right]$$
$$+ \sum_{i=1}^{N} \sum_{j=1}^{N} \Psi_{i,j}^{(n)} \cdot \gamma_{\boldsymbol{A}}^{-} \cdot \mathrm{I}\left[A_{i,j} = 0 \neq A_{i,j}'\right] + \Psi_{i,j}^{(n)} \cdot \gamma_{\boldsymbol{A}}^{-} \cdot \mathrm{I}\left[A_{i,j} = 1 \neq A_{i,j}'\right] \tag{145}$$
$$< \eta^{(n)}.$$

While our collective certificate derived in § 4 only considers one perturbation type, we have already discussed how to certify robustness to perturbation models with multiple perturbation types in § F.3.2: We use a different budget variable per input dimension and perturbation type. Furthermore, the attribute bits of each node share the same noise level. Therefore, we can use the method discussed in § E.3, i.e. use a single budget variable per node instead of using one per node and attribute.

Modelling our collective problem in this way, using Eq. 145 as our base certificates and rewriting the first two sums using inner products results in the optimization problem

$$\min_{\boldsymbol{b}^+,\boldsymbol{b}^+,\boldsymbol{B}^+,\boldsymbol{B}^-} -\sum_{n\in\mathbb{T}} \mathrm{I}\left[\gamma_{\boldsymbol{X}}^+ \left(\boldsymbol{\psi}^{(n)}\right)^T \boldsymbol{b}_{\boldsymbol{X}}^+ + \gamma_{\boldsymbol{X}}^- \left(\boldsymbol{\psi}^{(n)}\right)^T \boldsymbol{b}_{\boldsymbol{X}}^-\right.$$

$$\left. +\gamma_{\boldsymbol{A}}^+ \sum_{i,j} \Psi_{i,j}^{(n)} \boldsymbol{B}_{i,j}^+ + \sum_{i,j} \gamma_{\boldsymbol{A}}^- \Psi_{i,j}^{(n)} \boldsymbol{B}_{i,j}^- \le \eta^{(n)}\right] \quad (146)$$

$$\text{s.t.} \quad \sum_{m=1}^N b_m^+ \le r_{\boldsymbol{X}}^+, \quad \sum_{m=1}^N b_m^- \le r_{\boldsymbol{X}}^-, \quad \sum_{i=1}^N\sum_{j=1}^N B^+{}_{i,j} \le r_{\boldsymbol{A}}^+, \quad \sum_{i=1}^N\sum_{j=1}^N B^-{}_{i,j} \le r_{\boldsymbol{A}}^-, \quad (147)$$

$$\boldsymbol{b}^+, \boldsymbol{b}^- \in \mathbb{N}_0^N \quad \boldsymbol{B}^+, \boldsymbol{B}^- \in \mathbb{N}_0^{N\times N}. \quad (148)$$

This optimization problem is identical to that of Schuchardt et al. (2021) from Eqs. (133) to (135). The only difference is in how these problems would be mapped to a mixed-integer linear program. We would directly model the indicator functions in the objective using a single linear constraint. Schuchardt et al. (2021) would use multiple linear constraints, each corresponding to one point in the adversarial budget space.

To summarize: For randomly smoothed models, masking out perturbations using a-priori knowledge about a model's strict locality is equivalent to completely randomizing (here: flipping bits with probability $50\%$) parts of the input. While Schuchardt et al. (2021) only derived their certificate for binary data, it can also be applied to strictly local models for continuous data. Considering our certificates for Gaussian (Proposition F.1) and uniform (Proposition F.2) smoothing, where the base certificate weights are $\frac{1}{\sigma^2}$ and $\frac{1}{\lambda}$, respectively, it is possible to perform the same masking operation as Schuchardt et al. (2021) by using $\sigma \to \infty$ and $\lambda \to \infty$.

Finally, it should be noted that the certificate by Schuchardt et al. (2021) allows for additional constraints, e.g. on the adversarial budget per node or the number of nodes controlled by the adversary. As all of them can be modelled using linear constraints on the budget variables (see Section C of their paper), they can be just as easily integrated into our mixed-integer linear programming certificate.

