# OpenReview forum: "Localized Randomized Smoothing for Collective Robustness Certification"
_ICLR.cc/2023/Conference — ICLR 2023 notable top 25%_

### Official Review · Reviewer_7Fc5 · 2022-10-25

**Confidence:** 4
**Correctness:** 4
**Technical Novelty And Significance:** 3
**Empirical Novelty And Significance:** Not applicable
**Recommendation:** 8

**Clarity, Quality, Novelty And Reproducibility:**

The paper provides a novel approach to solving the collective certification problem, and the proposed solution achieves much better results than the current state-of-the-art. The paper is also very well-written and easy to follow, and the authors also provide extensive, detailed, reproducible experiments.

**Strength And Weaknesses:**

Strengths
- The proposed method extends the idea of using a strict locality condition to produce certificates to exploiting soft locality to produce a general certification method.
- The authors propose a general framework to efficiently combine local certificates as long as they satisfy the correct interface.
- The Variance-constrained certification approach provides a scalable way to do anisotropic $\ell_0$ certification, which could be independent interest.
- The experiments provided in the paper are detailed and give a comprehensive view of the benefits of the proposed algorithm.

**Summary Of The Paper:**

This paper proposes a framework, Localized Smoothing, to solve the problem of collective certification. The proposed method does not rely on any assumption for the validity, but it is formulated to be able to exploit any homophily/soft locality property in the multiple labeling task. The paper's main idea is to use region-specific smoothing distribution that puts more noise on less relevant pixels and less on relevant pixels. The paper also provides experimental evidence showing the algorithm's superior performance on image segmentation and node classification tasks.

**Summary Of The Review:**

The proposed method, Localized Smoothing, provides a general framework for collective robustness certification. The theorems proved in the paper show the theoretical benefit of using the proposed method over the existing state-of-the-art, which is also reflected in the experiments. Moreover, the Variance-constrained certification approach defined in the paper could be of independent interest. In general, I think the ideas in the paper, as well as the final algorithm, would be of interest to the ML community.

---

> ### Author Response · Authors · 2022-11-14
> **Rebuttal Reviewr 3 (7Fc5)**
>
> We are glad to hear that you are satisfied with our method, the experimental evaluation and the writing of the paper.
>
> If you have any further questions during the discussion period, please let us know.

---

### Official Review · Reviewer_jKFf · 2022-10-25

**Confidence:** 4
**Correctness:** 4
**Technical Novelty And Significance:** 3
**Empirical Novelty And Significance:** 3
**Recommendation:** 8

**Clarity, Quality, Novelty And Reproducibility:**


As stated above the paper is quite well written and presents a theoretically novel algorithm (with strong empirical evaluation).
However, one aspect that I did not find to be discussed in the paper is the inference procedure:

At inference time (when we are just interested in the result, not the certificate) RS requires to determine the top class (for each output) under the smoothing distribution.
Can you comment on the inference procedure in your case? In particular, whether we can solve for the same smoothing distribution when at inference time given a (potentially perturbed) as used during certification time and then use this to determine the output class?

In terms of reproducibility I believe that access to the code will be instrumental. While the authors promise to release the code on OpenReview in Section 9, I can not see it at the time I am writing this review. As I assume the cause of this is either an OpenReview setting or miscommunication by the review guidelines, I am giving the Authors the benefit of the doubt.






A minor formatting issue: Table 1 (Page 30) is too wide for the page.




**Strength And Weaknesses:**

**Strengths**:
- The paper is well-written and easy to follow. Arguments in the main paper are thorughly backed-up by derivation and explanation in the appendix.
- Algorithmically interesting approach for a relevant problem.
- Mathematical justifications are thorough and appear to be correct seem correct.

**Weaknesses**:
- Some aspects of inference are unclear (see below).



**Summary Of The Paper:**

The paper proposes a novel certification mechanism for certification in the multiple-output setting (e.g., image segmentation) based on randomized smoothing (RS), referred to as a collective certificate.
RS typically is used to create a robust-by-construction classifier in the (single-output) classification setting, by evaluating a base model multiple times and reporting the majority output.

The key idea of this work is to utilize perturbation distributions for RS that match the locality patterns of the model outputs (e.g., use anisotropic distributions that capture locality). Prior works have either ignored this locality aspect or only treated strict locality imposed by the receptive field of convolutions.
This, in turn, requires determining the appropriate anisotropic smoothing distirbutions for all local regions, which the authors propose to solve a mixed-interger linear program (MILP). To avoid the high cost of the MILP, discretizations, and simplfications (larger regions) as well as a relation to linear programming are considered.

The approach is thoroughly evaluated on image segmentation and node classification. The results are good if the locallity assumptions truly hold (and deteriored as expected if not).

**Summary Of The Review:**

This is a well-written paper proposing a new smoothing approach for multi-output settings.
My main concern is how inference would be handled soundly.

---

> ### Author Response · Authors · 2022-11-14
> **Rebuttal Reviewer 2 (jKFf)**
>
> Thank you for your thorough review!
> We are glad to hear that you also find collective certification to be a relevant problem and were satisfied with the quality of the writing.
>
> Concerning your comments on clarity, quality, novelty and reproducibility:
>
> ### Can you comment on the inference procedure in your case? In particular, whether we can solve for the same smoothing distribution when at inference time [...]?
> Before answering your question, we would like to address a potential misunderstanding: We do not use the MILP to solve for an optimal smoothing distribution. Instead, we use the MILP to combine the many per-output certificates into a single, stronger collective certificate. These per-output certificates are obtained via randomized smoothing with different *fixed* anisotropic smoothing distributions, which are chosen based on our assumptions about the model's locality (e.g. homophily in graphs).
>
> To answer your question:
> Because the distributions are fixed, inference is very similar to the usual randomized smoothing procedure:
> * For each output, we use $N_1$ samples to get an estimate of the majority class.
> * We then use $N_2$ samples to verify whether this is (with high probability) the actual smoothed prediction or whether we have to abstain.
>
> The only difference is that we have to adjust the confidence levels using Holm or Bonferroni correction to avoid the multiple comparisons problem.
>
> **Based on your comment, we have updated Section 5 to more explicitly reference Appendix G, in which the inference procedures for both our method and the baselines are discussed.**
>
> In practice, our method could be combined with methods that try to find optimal smoothing distributions, like the one proposed by Eiras et al. [1].
> However, working with such methods requires much additional care (see Súkenı́k et al. (2022) [2]) and exposition. We wanted to focus on the improved accuracy-robustness-tradeoff offered by localized smoothing, instead of just finding the optimal distribution.
>
> ### In terms of reproducibility I believe that access to the code will be instrumental.
> We have now posted a link to the implementation (including experiment configuration files) in a separate comment above.
>
> The submission guidelines suggested to "make a comment directed to the reviewers and area chairs and put a link to an anonymous repository."
> We were sadly unaware that this would only be possible after the start of the discussion period.
>
> ### Table 1 (Page 30) is too wide for the page.
>
> Thank you for pointing this out. We have fixed the formatting by moving some of the content into the table caption.
>
> ---
> We hope that we addressed all your questions and comments to your satisfaction. If you have any further questions during the discussion period, please let us know.
>
> ### References
> [1] ANCER: Anisotropic Certification via Sample-wise Volume Maximization; Francisco Eiras, Motasem Alfarra, M. Pawan Kumar, Philip H. S. Torr, Puneet K. Dokania, Bernard Ghanem, Adel Bibi; TMLR 08/2022; https://openreview.net/pdf?id=7j0GI6tPYi
> [2] Intriguing Properties of Input-Dependent Randomized Smoothing; Peter Súkenı́k, Aleksei Kuvshinov, Stephan Günnemann; ICML 2022; https://proceedings.mlr.press/v162/sukeni-k22a.html

---

> > ### Comment · Reviewer_jKFf · 2022-11-18
> > **Reply**
> >
> > Thank you for your reply and clearing up the inference procedure.
> > With this, my doubts about the soundness of the method are cleared up and I'll increase my score accordingly.

---

### Official Review · Reviewer_gYEN · 2022-10-27

**Confidence:** 4
**Correctness:** 3
**Technical Novelty And Significance:** 3
**Empirical Novelty And Significance:** 3
**Recommendation:** 6

**Clarity, Quality, Novelty And Reproducibility:**

This work is clear in some aspects such as the motivation to the problem and the proposed method. However, and as dictated in the weaknesses, there are several parts of this paper that need improvements.

**Strength And Weaknesses:**

Strengths:

- The problem that this paper studies is important.

- The experiments presented in this work provide improvements over previous approaches supporting the main claims.

Weaknesses:

- The scale of the experiments in this work is rather small. Certifying only 50 images from Pascal and CityScapes might be not enough to draw conclusions on. I would suggest extending the results on larger amount of samples. Also, some fine-grained experimental analysis on instances containing different number of categories could also be insightful.

- The second contribution of this work claims an efficient anisotropic randomized smoothing certificate. However, this problem was previously explored in the literature, as dictated in the related work, by Eiras et.al. Efficiency analysis between different approaches would strengthen this contribution. Further and since the experiments are done on few number of samples, how does the proposed method compare to ANCER from Eiras et.al when combined with other baselines?

- The writing of the paper along with the presented notation need improvements in some parts of this work. For example, small paragraphs discussing both Theorem 4.2 and 5.1 would ease the reading of this work.
Further, in theorem 4.2, the prediction is referred to as $y_n$ while in the third paragraph of section 4 it is $\mathbf y$.


**Summary Of The Paper:**

This paper proposes localized randomized smoothing to improve collective certificate.
It formulates the problem as a solution to a linear program after smoothing different parts of a given input image differently.
Experiments were carried on two standard segmentation benchmarks and one node classification dataset showing the effectiveness of the proposed approach.

**Summary Of The Review:**

This work has several merits of this work including the motivation, theoretical analysis, and supporting experiments. However, there are few concerns that I hope to be addressed in the discussion period. This includes the claim about the second contribution (efficient anisotropic certification), limited experimental analysis, and the writing of this paper.

---

> ### Author Response · Authors · 2022-11-14
> **Rebuttal Reviewer 1 (gYEN)**
>
> We first want to thank the reviewer for their thorough revision of our work and we are happy to hear that they also find collective robustness certification to be an important problem.
>
> In the following, let us first address your multi-part question on our variance-constrained anisotropic smoothing certificate, before discussing the experimental evaluation and how we used your feedback to improve the writing of the paper.
>
> ### "[...] this work claims an efficient anisotropic randomized smoothing certificate. However, this problem was previously explored in the literature, as dictated in the related work, by Eiras et al."
> You are absolutely right that Eiras et al. (2022) [1] -- as well as Fischer et al. (2020) [2] -- already proposed anisotropic randomized smoothing certificates. We point this out in Sections 2 and 5.
>
> However, they use uniform or Gaussian noise to prove robustness of $l_1$ or $l_2$ perturbations of *continuous* data. For these threat models, we actually use their proposed certificates as "base certificates" in our certification pipeline. We are not trying to compete with them.
>
> We specifically propose the variance-constrained certificate for efficient anisotropic smoothing of  *binary* (graph) data, which was not discussed in prior work.
>
> Based on your comment, we have updated Section 5 to stress that the variance-constrained certificate is specifically meant for binary data and it being applicable to continuous data is just an interesting insight.
>
> ### "Efficiency analysis between [the] different approaches would strengthen this contribution."
> While the variance-constrained certificate is specifically proposed for binary data and not necessarily meant to outperform existing certificates for continuous data, we agree that such a comparison could be interesting. For Gaussian smoothing with covariance matrix $\Sigma$, the l.h.s. of Eq. 6 has a closed-form expression. It is $exp((x'-x) \Sigma^{-1} (x'-x))$, i.e. it depends on the Mahalanobis-distance, which must also be computed in the certificate of Eiras et al. The r.h.s. of Eq. 6 can be evaluated in constant time.
>
> We have added a section comparing the variance-constrained certificate for anisotropic Gaussian smoothing and ANCER (Eiras et al. (2022) [1]) to the appendix (see Appendix F.3.3).
>
> ### In case you are also interested in how effective the different approaches are:
> Recall that Neyman-Pearson-based certificates like (Cohen  et al. (2019) [3]) and (Eiras et al. (2022) [1]) certify based on the probability $p$ of predicting a specific class. Our variance-constrained certificate uses the expected value $\mu$ and variance $\zeta$ of a softmax score.
>
> For the following comparison, we assume a Gaussian smoothing distribution with covariance $\Sigma$. We vary parameters $p$ and $\mu$ in $[0.5, 1.0]$. For each $\mu$, we calculate the largest possible variance $\zeta$ (using the Bhatia–Davis inequality), which will give us the weakest possible variance-constrained certificate (see Eq. 6).
> For both certificates, we then determine the certifiable radius, i.e. the largest $r$ such that the certificate guarantees robustness within $\{x' \mid \sqrt{(x' - x)^T\Sigma^{-1}(x'-x)} \leq r\}$.
>
> [This figure (https://figshare.com/articles/figure/Localized_Smoothing_Rebuttal/21534618?file=38171511)](https://figshare.com/articles/figure/Localized_Smoothing_Rebuttal/21534618?file=38171511)  shows the difference in certifiable radius, with the dashed line indicating parameters for which both certificates are identical.
> The Neyman-Pearson-based certificates are stronger when $p$ is large, i.e. almost all samples from the smoothing distribution are correctly classified, but not necessarily with high confidence.
> The variance-constrained certificate is stronger when $p$ is smaller and $\mu$ is larger, i.e. some samples are misclassified, but the correctly classified ones have high confidence.
>
> Again, the above figure shows the worst case for the variance-constrained certificate.
> If the variance $\zeta$ of the softmax scores is small, our method can certify much larger radii, as shown in [this figure (https://figshare.com/articles/figure/Localized_Smoothing_Rebuttal/21534618?file=38172108)](https://figshare.com/articles/figure/Localized_Smoothing_Rebuttal/21534618?file=38172108).
>
> We have also included this comparison in Appendix F.3.3.

---

> > ### Author Response · Authors · 2022-11-14
> > **Rebuttal Reviewer 1 (gYEN) - Part 2**
> >
> > ### "How does the proposed method compare to ANCER from Eiras et al. when combined with other baselines?""
> > Concerning SegCertify*: Recall that this baseline is identical to the naive collective certificate, i.e. certifying each output $f_n$ independently and then counting how many are certifiably robust.
> > As discussed in Section 3.2, **our collective certificate is always at least as strong when using the same smoothing distribution** (see also Eq. 1).
> >
> > On top of that, using **SegCertify with ANCER will usually lead to a lower certified accuracy than SegCertify with isotropic smoothing**.
> > When smoothing output $g_n$ using $D$-dimensional Gaussian noise with standard deviations $\sigma_1,\dots, \sigma_D$, ANCER yields a certified ellipsoid
> > $\\{x' \mid \sqrt{\sum_{d=1}^D(x'_d - x_d) \mathbin{/} \sigma_d^2} < r \\}$
> > , i.e. the output is least robust in parts of the input smoothed with $\sigma_\mathrm{min} = \min_d \sigma_d$.
> > An adversary performing $l_p$-norm perturbations can just allocate all of their adversarial budget to these least robust dimensions.
> > Thus, one might as well smooth the entire input with $\sigma_\mathrm{min}$. One would get the same type of robustness guarantee using less noise, which should usually lead to better predictions and (due to more consistent predictions) a larger $r$.
> > [This figure (https://figshare.com/articles/figure/Localized_Smoothing_Rebuttal/21534618?file=38206389)](https://figshare.com/articles/figure/Localized_Smoothing_Rebuttal/21534618?file=38206389) also illustrates this effect on prediction quality for the Cityscapes segmentation dataset.
> >
> > **The same argument applies to CenterSmoothing** and to $l_1$ perturbations with anisotropic uniform noise. The adversary can allocate all of their budget to the least robust dimension, so one might as well use isotropic noise.
> >
> > Using **anisotropic noise for certifying collective robustness under $l_p$-norm perturbations
> > is only beneficial for our method**, which explicitly models that the outputs have different levels of robustness in different parts of the input and that the adversary has to attack all of them simultaneously.
> >
> > ### "I would suggest extending the results on larger amount of samples."
> > We agree that using even more images from PascalVOC or CityScapes would be preferable.
> > We plan to include results for larger subsets of the two certificates in the camera ready version.
> >
> > We would however like to point out that image segmentation is not our sole focus, but that we also consider the Cora ML and Citeseer datasets typically used in the graph robustness domain.
> >
> > More importantly, note that our experiments are significantly more compute-intensive than those in other randomized smoothing literature. We evaluate three randomized smoothing certificates (ours, SegCertify and center smoothing), with each one having different accuracy-robustness trade-offs for different smoothing distributions parameters. To show that one is better than the others, we have to demonstrate Pareto-optimality by repeating our experiments for a large number of different parameter values (see, e.g., Fig. 2).
> >
> > Aside from that, robustness certification (even with randomized smoothing), is an inherently hard and expensive problem. Prior work (which does not evaluate Pareto-optimality) also uses small subsets of the typically used benchmark datasets. For instance, the seminal paper of Cohen et al. (2019) [3] used a small subset of ImageNet. The baseline SegCertify (Fischer et al. (2021) [4]) used 100 images from Cityscapes.
> >
> >
> > ### "For example, small paragraphs discussing both Theorem 4.2 and 5.1 would ease the reading of this work [...]"
> > Thank you for your feedback. We have added a paragraph that provides an intuitive explanation of the certificate from Theorem 5.1. We have also expanded the discusion of  Theorem 4.2. In particular, we motivate why our collective certificate from Eq. 3 can be expressed as a MILP and point out parallels between the two formulations.
> >
> > ### "In Theorem 4.2, the prediction is referred to as y_n while in the third paragraph of section 4 it is y.""
> > Thank you for pointing out this typo, we have corrected it.
> >
> > -----------
> >
> > We hope that we addressed all your questions and comments to your satisfaction. If you have any further questions during the discussion period, please let us know.
> >
> > ### References
> > [1] ANCER: Anisotropic Certification via Sample-wise Volume Maximization; Eiras et al.; TMLR 08/2022; https://openreview.net/pdf?id=7j0GI6tPYi
> > [2] Certified Defense to Image Transformations via Randomized Smoothing; Fischer et al.; NeurIPS 2020; https://proceedings.neurips.cc/paper/2020/hash/5fb37d5bbdbbae16dea2f3104d7f9439-Abstract.html
> > [3] Certified Adversarial Robustness via Randomized Smoothing; Cohen et al.; ICML 2019; https://proceedings.mlr.press/v97/cohen19c.html
> > [4] Scalable Certified Segmentation via Randomized Smoothing; Fischer et al.; https://proceedings.mlr.press/v139/fischer21a.html

---

> > > ### Comment · Reviewer_gYEN · 2022-11-16
> > > **Thank you**
> > >
> > > I would like to thank the authors for their response. I applaud the authors on the efforts put in the response and for incorporating the changes in the paper. Therefore, I increase my score from 5 to 6.
> > > The main reason behind not putting a higher score is the experimental evaluation. I would appreciate more analysis on larger scale experimental setups such as increasing the number of certified samples and runtime comparisons.
> > > At last, while indeed ANCER optimizes a diagonal covariance matrix per input sample, the theoretical analysis covered the general non-diagonal $\Sigma$.

---

### Author Response · Authors · 2022-11-14
**Changelog**

Based on the reviewers' valuable feedback, we have made multiple improvements to our submission.
The changes are highlighted in blue in our updated manuscript.

### Main changes
* Stress that we are not competing with existing anisotropic smoothing certificates for continuous data in Section 5
* Discuss variance-constrained certification for Gaussian smoothing in Appendix F.3.3.
* Expand explanation and motivation of Theorem 4.2
* Provide intuitive explanation of Theorem 5.1
* Reference Appendix G on Monte Carlo inference and certification during our discussion of base certificates in Section 5


### Minor changes
* Add footnote that the variance-constrained certificate depends on Rényi-divergence of smoothing distributions to Section 5
* Replace $y_n$ with $\mathbf{y}$ in Theorem 4.2
* Replace $g_n$ with arbitrary function $g$ in Theorem 5.1
* Reduce width of Table 1
* Make space for new explanations by removing a few linebreaks and less important words / remarks.
* Update reference for Eiras et al. (2022), now that their paper has been accepted for publication

---

### Decision · Program_Chairs · 2023-01-20

**Decision:**

Accept: notable-top-25%

**Justification For Why Not Higher Score:**

The paper could be accepted as an oral potentially. My hesitations are that 1) the initial versions of the paper lacked some writing clarity, and while the revisions addressed some concerns, it can be improved further, 2) The certification requires solving a MILP, which may present scalability issues.

**Justification For Why Not Lower Score:**

The authors present a sound and novel technique that improves randomized smoothing certificates, and is justified both theoretically and empirically.

**Metareview: Summary, Strengths And Weaknesses:**

The authors extend the randomized smoothing approach to anisotripic distributions, showing how this extension enables computation of stronger robustness certificates for tasks beyond classification like image segmentation

Strengths:
1. The method proposed is sound and exploits a natural idea of anisotropic noise to obtain tighter certificates.
2. The experiments convincingly demonstrate the effectiveness of the technique.

Weaknesses:
1. No major weaknesses, minor ones were clearly addressed during the rebuttal phase.

Hence, I recommend acceptance.


**Note From Pc:**

if the above contains the word "oral" or "spotlight" please see: "oral" presentation means -> notable-top-5% and "spotlight" means -> notable-top-25%. As stated in our emails, we are disassociating presentation type from AC recommendations

**Summary Of Ac-Reviewer Meeting:**

No meeting